# Ultrametric Cluster Hierarchies: I Want 'em All!

**Andrew Draganov**[*1,2]    **Pascal Weber**[*3,4,5]
**Rasmus Jørgensen**[1]    **Anna Beer**[3]    **Claudia Plant**[3,5]    **Ira Assent**[1,2]

[1]*Department of Computer Science, Aarhus University*, Aarhus, Denmark
[2]*IAS-8: Data Analytics and Machine Learning, Forschungszentrum Jülich*, Germany
[3]*Faculty of Computer Science, University of Vienna*, Vienna, Austria
[4]*UniVie Doctoral School Computer Science, University of Vienna*, Vienna, Austria
[5]*Data Science @ Uni Vienna, University of Vienna*, Vienna, Austria

draganovandrew@gmail.com    pascal.weber@univie.ac.at

## Abstract

Hierarchical clustering is a powerful tool for exploratory data analysis, organizing data into a tree of clusterings from which a partition can be chosen. This paper generalizes these ideas by proving that, for any reasonable hierarchy, one can optimally solve any center-based clustering objective over it (such as $k$-means). Moreover, these solutions can be found exceedingly quickly and are *themselves* necessarily hierarchical. Thus, given a cluster tree, we show that one can quickly access a plethora of new, equally meaningful hierarchies. Just as in standard hierarchical clustering, one can then choose any desired partition from these new hierarchies. We conclude by verifying the utility of our proposed techniques across datasets, hierarchies, and partitioning schemes.[†]

## 1   Introduction

Hierarchical clustering is a fundamental technique for exploratory data analysis [42, 77]. The key idea is that having a tree of clusterings over a given dataset allows users to choose any partition from this hierarchy that best suits the users' needs. These notions go beyond standard agglomerative clustering algorithms to also include density-based clustering techniques like DBSCAN [24] and HDBSCAN [12]. Even non-hierarchical clustering algorithms like $k$-means can be solved efficiently by leveraging hierarchical representations [17, 18, 55].

The central underlying concept of hierarchical clustering methods is that they can be modeled using ultrametrics: distances which satisfy the strong triangle inequality $d(x, z) \leq \max(d(x, y), d(y, z))$ for all $x, y, z$. Put simply, a set of points can only satisfy this inequality if they are arranged in nested, hierarchical structures [4]. Originally described for phylogenetic and agglomerative clustering tasks [57, 65], the depth of this equivalence between hierarchical clustering and ultrametrics has inspired multiple subfields of unsupervised learning theory [3, 19, 22].

**Our results.**   We prove an elegant, previously unknown property of ultrametrics: all standard center-based clustering tasks (i.e., $k$-means, $k$-median, $k$-center) can be solved optimally in any ultrametric. The key insight is that, in this setting, these center-based clustering tasks can be reduced to sorting. Consequently, our algorithm is remarkably efficient: given an ultrametric on $n$ points, finding the full set of optimal solutions (i.e., for all $k \in [n]$) requires only $\texttt{Sort}(n)$ time – the time to sort $n$ values. Our results therefore improve on recent work in both runtime and generality [6, 18, 44].

39th Conference on Neural Information Processing Systems (NeurIPS 2025).

---

[*]First authors with equal contribution in alphabetical order.

[†]https://github.com/pasiweber/SHiP-framework/ – Implementation and experiments
https://pypi.org/project/SHiP-framework/ – Python interface package

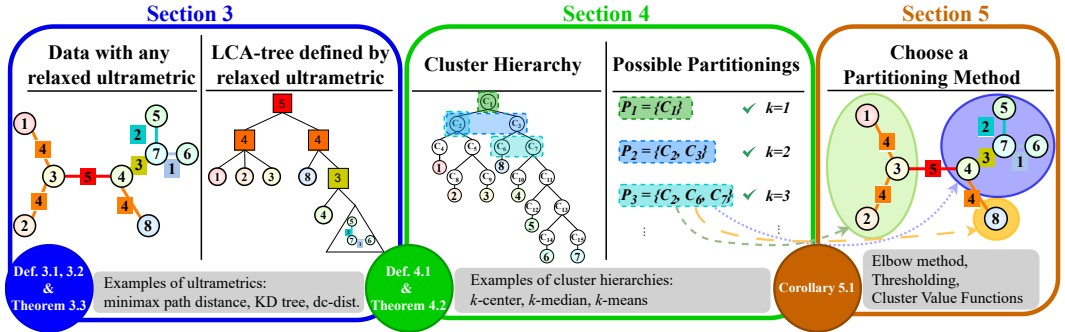

Figure 1: Overview of our proposed SHiP clustering framework in which we (1) fit an ultrametric, (2) choose a center-based hierarchy on the ultrametric, and (3) extract a partition from the hierarchy.

Moreover, these center-based partitions are *themselves* necessarily hierarchical: the set of optimal center-based clustering solutions in an ultrametric form a cluster-tree.

Building on these theoretical insights, we introduce the SHiP (**S**imilarity-**Hi**erarchy-**P**artitioning) clustering framework. While traditional approaches merely extract flat partitions from predefined hierarchies, SHiP enables discovering entirely *new* hierarchical relationships via center-based clustering, substantially increasing the expressiveness of hierarchical clustering. Users can then efficiently extract any desired partition from these novel hierarchies. We depict this process in Figure 1, where we specify which sections of our paper describe the SHiP framework's various components.

Figure 2 provides a motivating example of the SHiP framework's practical value. When HDBSCAN fails to find the desired partition, the typical workflow would involve trying entirely different algorithms like Euclidean $k$-means or Ward clustering [77], repeating the costly data-fitting step while still potentially failing to find the correct partition. SHiP eliminates this redundancy: having already fit the dc-dist ultrametric [6] during HDBSCAN's execution, we can instantly

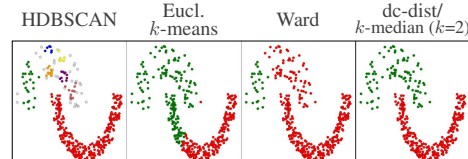

Figure 2: Clusterings on the 2-moons dataset with varying densities.

explore alternative cluster structures through different hierarchies. In this case, the correct partition is found in the $k$-median hierarchy when $k = 2$. This exemplifies SHiP's core advantage: once an ultrametric is computed, numerous clustering perspectives become available essentially for free, enabling rapid dataset exploration. Our extensive experiments confirm this is not a cherry-picked example; many SHiP-derived combinations consistently produce novel, high-quality clusterings that compete with or outperform state-of-the-art algorithms with minimal additional computational cost.

## 2   Related Work

Our primary theoretical result is that all center-based clustering tasks in ultrametrics can be reduced to sorting. The novelty and simplicity of this is particularly surprising as each of these topics—ultrametrics, hierarchical clustering, and center-based clustering—has received significant attention:

**Ultrametrics.** Due to the strong triangle inequality, ultrametrics naturally encode hierarchical relationships [4]. Consequently, ultrametrics arise in numerous applications like biology [52], number theory [34], or physics [62]. In computer science, they allow for simple, parallelizable solutions [38] and extensive research has explored low-distortion embeddings of metrics into ultrametrics [4, 19]. This is often done via hierarchically well-separated trees (HSTs), data structures which model the original distances via tree path-distances [3, 39]. Notable examples include KD trees [8], Cover trees [9], and HSTs which obtain optimal distortion but are slower to compute [25, 79]. These are all used to accelerate machine learning algorithms [53, 73], computer graphics [27], and nearest-neighbor search [63, 69]. Our framework enables the first comprehensive evaluation of HSTs across clustering tasks, revealing that they consistently fail to preserve underlying cluster structures regardless of

hierarchy or partitioning method. Our findings challenge the commonly assumed suitability of HSTs for accelerating ML tasks, especially in precision-sensitive settings.

A separate line of work has studied clustering algorithms in trees. Here, it is known that, for a fixed value of $k$, $k$-center can be solved optimally in $O(n)$ time [54, 75, 10]. Similarly, $k$-median can be solved optimally in $O(n \log^{k+2} n)$ time [71, 7]. There are two primary differences between this work and our setting. First, in our setting, the centers may only be placed on leaves in the tree. Second, we require that all distances are ultrametric, whereas distances over edge-weighted trees are not necessarily ultrametric. Consequently, our results show that under these additional constraints, all center-based clustering tasks in trees reduce to the same underlying problem.

**Hierarchical Clustering.** Hierarchical clustering methods represent data at multiple scales of granularity in a single structure. These techniques typically produce dendrograms—tree structures that encode nested partitions of the data—from which users can extract flat clusterings (partitions) at different resolution levels [12, 56]. Part of hierarchical clustering's appeal is that, given a hierarchy, the partitions can often be extracted in $O(n)$ time [43]. Thus, hierarchical clustering is particularly valuable in exploratory data analysis, where the number of clusters is often unknown a priori [51, 64].

Traditional hierarchical clustering algorithms (such as single- and complete-linkage) merge clusters according to distance calculations which are often ultrametric in nature [57]. For instance, the single-linkage algorithm corresponds to an ultrametric over the dataset's minimum spanning tree (MST), where the *single-link distance* (also known as the minimax path distance) between two points is given by the weight of the largest edge in the MST path between them [26]. Recent variants of hierarchical clustering have moved away from procedural merging rules in favor of finding ultrametrics which minimize an objective function with respect to the data [1, 15, 16, 19, 22].

**Center-Based Clustering.** Center-based clustering is one of the most thoroughly researched paradigms in unsupervised learning. Formulations such as $k$-means, $k$-median and $k$-center are NP-hard in arbitrary metric spaces [21, 31], leading to a rich literature on approximation algorithms with provable speed/accuracy guarantees [2, 28, 55]. Within this, several papers have leveraged HST ultrametrics for faster center-based clustering [17, 23, 44].

Two recent works are particularly relevant to our approach. Beer et al. [6] showed that $k$-center can be solved optimally in the density-connectivity ultrametric (dc-dist) in $O(n^2)$ time. This ultrametric is a generalization of the single-link distance and is the backbone behind the DBSCAN and HDBSCAN clustering algorithms. The algorithm in Beer et al. is essentially equivalent to algorithm 1 in Cohen-Addad et al. [18]. Cohen-Addad et al. went further by providing a second algorithm which optimally solves $k$-median in a 2-HST ultrametric in $O(n \log^2(\Delta + n))$ time, where $\Delta$ is the tree-depth of the HST they construct. Both papers observed that the resulting solutions are necessarily hierarchical. Our results can be interpreted as a generalization of algorithm 2 in Cohen-Addad et al. Specifically, we show that their algorithm 2 holds for any ultrametric and reduces to $k$-center (which itself reduces to sorting).

In this sense, our approach yields several advantages over the prior techniques. First, rather than designing algorithms for specific center-based clustering tasks on specific ultrametrics, we provide a general theoretical framework that handles *all* center-based clustering objectives optimally in *all* ultrametrics. Second, our runtime improves over the state-of-the-art and we prove it to be tight. Lastly, our algorithms are relatively simple to abstract, allowing us to evaluate our SHiP framework across many ultrametrics and datasets. Such an experimental ablation was previously absent from the literature.

## 3    Ultrametrics and Tree Representations

We begin by formally introducing the data structure that we require for our proof techniques. Namely, the upcoming results hold over a generalization of the standard ultrametric:

**Definition 3.1.** Let $L$ be a set. Then $d : L \times L \rightarrow \mathbb{R}_{\geq 0}$ is a *relaxed ultrametric* over $L$ if, for all $\ell_i, \ell_j, \ell_k \in L$, the following conditions are satisfied:

$$(1)\ d(\ell_i, \ell_j) = d(\ell_j, \ell_i) \geq 0 \quad \text{and} \quad (2)\ d(\ell_i, \ell_k) \leq \max(d(\ell_i, \ell_j), d(\ell_j, \ell_k)).$$

Note that the standard ultrametric is a restriction that additionally requires $d(\ell_i, \ell_i) = 0$. Thus, not all relaxed ultrametrics are distances as $d(\ell_i, \ell_i) > 0$ is allowed. Still, we use the word "distance" for readability. We represent relaxed ultrametric relationships via the following data structure:

**Definition 3.2.** A *lowest-common-ancestor tree* (*LCA-tree*) is a rooted tree $T$ such that every node $\eta \in T$ has value $d(\eta) \geq 0$ associated with it. We write $\eta_i \preceq \eta_j$ to indicate that $\eta_j$ lies on the path from $\eta_i$ to the root and $\eta_i \vee \eta_j$ to refer to the LCA of $\eta_i$ and $\eta_j$. We say that the *LCA-distance* between two leaves $\ell_i, \ell_j \in T$ is given by $d(\ell_i \vee \ell_j)$.

An LCA-tree is not necessarily binary: if three or more subtrees are all equidistant, they can all be children of the same node. While similar data structures already exist for standard ultrametrics [4, 35], the following theorem (proof in A.2) states that it can also encode all relaxed ultrametrics:[1]

**Theorem 3.3.** *Let $(L, d')$ be a finite relaxed ultrametric space. Then there exists LCA-tree $T$ with LCA-distance $d$ and a bijection $f : L \leftrightarrow \text{leaves}(T)$ such that, for all $\ell_i, \ell_j \in L$, $d'(\ell_i, \ell_j) = d(f(\ell_i) \vee f(\ell_j))$.*

This is visualized by the first box of Figure 1. It shows a minimum spanning tree (MST) over some data on the left. In this MST, the single-link ultrametric is given by the weight of the largest edge in the path between two nodes [26]. For example, the single-link distance between nodes ① and ④ is ⑤. The right-hand side of the first box of Figure 1 then stores these distances in an LCA-tree (i.e., the LCA of nodes ① and ④ has value 5). We prove in Appendix A.2 that all LCA-trees are relaxed ultrametrics as long as they satisfy the following conditions:

**Corollary 3.4.** *Let $T$ be an LCA-tree. For any leaf $\ell \in T$, let $p(\ell) = [\ell, \eta_a, \ldots, \eta_b, r(T)]$ be the path from $\ell$ to the root of the tree $r(T)$. Then the LCA-distances on $T$ form a relaxed ultrametric if and only if, for all $\ell \in \text{leaves}(T)$ and $\eta_i, \eta_j \in p(\ell)$, the following conditions are satisfied:*

$$(1) \quad d(\ell) \geq 0 \quad and \quad (2) \quad \eta_i \preceq \eta_j \implies d(\eta_i) \leq d(\eta_j).$$

Corollary 3.4 describes a key property: if a tree's node values grow along paths from the leaves to the root, then the corresponding LCA-distances are a relaxed ultrametric. This is the natural representation of a hierarchy: since the subtrees grow in size as we go towards the root, the values corresponding to those subtrees also grow. Going forward, we assume that a relaxed ultrametric is given in its LCA-tree form, satisfying the properties in Corollary 3.4.

## 4 Center-based Clustering in Ultrametrics

Our main theoretical result is that center-based clustering can be solved optimally in a relaxed ultrametric, that these solutions are hierarchical, and that they can all be found in $\text{Sort}(n)$ time.[2] We define the $(k, z)$-*clustering* and the $k$-*center clustering* objectives over an LCA-tree $T$ as finding the set of centers $\mathbf{C} \subseteq \text{leaves}(T)$ with $|\mathbf{C}| = k$ which minimize, respectively,

$$\underbrace{\text{Cost}_z(T, \mathbf{C}) = \sum_{\ell \in \text{leaves}(T)} \min_{c \in \mathbf{C}} d(\ell, c)^z}_{(k,z)\text{-clustering objective}}, \qquad \underbrace{\text{Cost}_\infty(T, \mathbf{C}) = \max_{\ell \in \text{leaves}(T)} \min_{c \in \mathbf{C}} d(\ell, c)}_{k\text{-center clustering objective}}.$$

Note that $(k, z)$-clustering gives the well-known $k$-median and $k$-means tasks for $z = 1$ and $z = 2$.

We now define what it means for a clustering to be hierarchical. Given a set of points $L$, we define a *cluster $C$* as any subset of $L$. We then define a *partition* $\mathcal{P}_k = \{C_1, \ldots, C_k\}$ as any *non-overlapping* set of $k$ clusters, i.e., for all $C_i, C_j \in \mathcal{P}$, $C_i \cap C_j = \emptyset$. Then:

**Definition 4.1.** (Lin et al. [48]) A *cluster hierarchy* $\mathcal{H} = \{\mathcal{P}_1, \ldots, \mathcal{P}_n\}$ is a set of partitions where

for $k = 1$:          $\mathcal{P}_1 \subseteq L$, and

for $1 < k \leq n$:     $\mathcal{P}_k = (\mathcal{P}_{k-1} \setminus C_i) \cup \{C_j, C_l\}$, such that (a) $C_i = C_j \cup C_l \in \mathcal{P}_{k-1}$ with $C_j \cap C_l = \emptyset$, (b) $C_j \neq C_i$ and $C_l \neq C_i$, and (c) $i \neq j \neq l$.

---

[1]Although standard ultrametrics are often encoded via shortest-path distances in a tree [61], *relaxed* ultrametrics cannot be represented in this way due to the possibility of having $d(\ell_i, \ell_i) > 0$.

[2]Although sorting is traditionally an $O(n \log n)$ operation, it often only requires $O(n \log \log n)$ time [30].

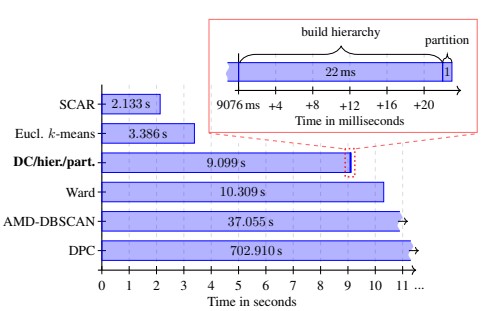

Figure 3a: Runtimes of competitors and our framework's components on the letterrec dataset. Compared to computing the dc-dist ultrametric, finding the $k$-means hierarchy and extracting the elbow partition requires negligible time.

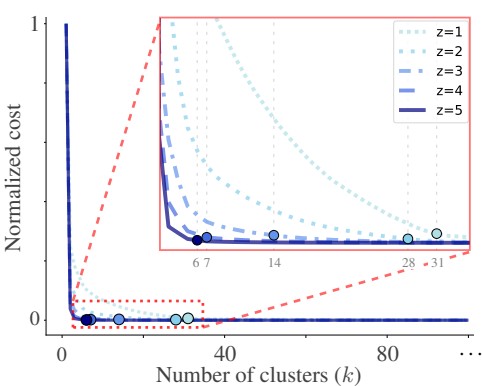

Figure 3b: Example elbow plots for the $z = 1, 2, 3, 4$, and 5 settings under the dc-dist ultrametric on the D31 synthetic dataset. Elbow locations (circles) are determined using all $k \in [1, n]$ with $n = 3100$.

In short, each partition $\mathcal{P}_k$ is obtained by splitting one cluster from $\mathcal{P}_{k-1}$ in two, implying that all cluster hierarchies can be represented as a rooted tree. We depict this in Figure 1 (middle), where the hierarchy is obtained by subdividing the clusters. We can now state our primary theoretical result:

**Theorem 4.2.** *Let $(L, d)$ be a finite relaxed ultrametric space represented over an LCA-tree $T$. Let $n = |L|$ and $z \in \mathbb{N}_{>0}$. Then, for both the $k$-center and $(k, z)$-clustering objectives on $T$, there exists an algorithm which finds the optimal solutions $\{\mathbf{C}_1, \ldots, \mathbf{C}_n\}$ for all $k \in [n]$ in $\texttt{Sort}(n)$ time. Furthermore, the respective partitions $\mathcal{H} = \{\mathcal{P}_1, \ldots, \mathcal{P}_n\}$ obtained by assigning all leaves in $T$ to their closest center satisfy Definition 4.1.*

**Proof Overview.** In words, Theorem 4.2 states that given any LCA-tree over a relaxed ultrametric, it takes $\texttt{Sort}(n)$ time to find the full set of optimal solutions for $k$-means, $k$-median or $k$-center across all values of $k$ and that these solutions are themselves hierarchical. The $k$-center part of this result is an extension of the farthest-first traversal algorithm in which one places each subsequent center on the point that is farthest from the current set of centers [31, 32]. In essence, Appendix A.3 shows that under the strong triangle inequality, the farthest-first traversal algorithm (a) becomes optimal and (b) reduces to sorting the distances in the ultrametric. We then show in Appendix A.4 that the $(k, z)$-clustering objectives in an ultrametric can be reduced to the $k$-center one in a relaxed ultrametric. That is, given an LCA-tree $T$ on which to do $(k, z)$-clustering, there exists a *new* LCA-tree $T'$ such that an optimal $k$-center solution on $T'$ *is* the optimal $(k, z)$-clustering solution on $T$. Interestingly, even if we are solving $(k, z)$-clustering in a standard ultrametric, it reduces to $k$-center in a relaxed one. This is why we require the notion of relaxed ultrametrics for our proofs.

A key insight which underpins this reduction is that these center-based cluster hierarchies *themselves* constitute relaxed ultrametrics. That is, let $\mathcal{H}$ be a hierarchy of optimal $k$-center or $(k, z)$-clustering solutions from Theorem 4.2. For each cluster $C$ in this hierarchy, consider the cost $d_{cost}(C)$ of assigning all of its points to a single optimal center. As we traverse the tree towards the root, these costs will necessarily grow (see Lemma A.13). Thus, by Corollary 3.4, the hierarchy $\mathcal{H}$ with LCA-distances $d_{cost}$ *must* be a relaxed ultrametric. Indeed, any binary LCA-tree satisfying the properties in Corollary 3.4 *is* its own optimal $k$-center hierarchy.

Finally, the runtime bottleneck for $k$-center lies in sorting the $O(n)$ unique distances in the ultrametric in order to apply the farthest-first traversal algorithm. This is also the bottleneck for $(k, z)$-clustering, as the reduction to $k$-center only requires $O(n)$ time. We verify that this runtime is tight: one cannot find the optimal centers for all values of $k$ in faster than $\texttt{Sort}(n)$ time (Lemma A.7). This speed is depicted in Figure 3a: although it takes several seconds to fit the density-connectivity ultrametric, it only takes *milliseconds* to construct the $k$-median hierarchy.

# 5 Choosing a Partition

Theorem 4.2 gives us a hierarchy of optimal center-based clustering solutions for all values of $k$ at once. However, what if we are interested in the "best" clustering from this hierarchy? To this end, there are several techniques for extracting additional partitions from these hierarchies in $O(n)$ time.

## 5.1 Feasibility of the Elbow Method

One of the most common methods for choosing a "best" clustering is the elbow method [72]. Here, one is given a set of values of $k$, $\{k_1, k_2, \ldots, k_f\}$, and a set of corresponding partitions $\mathcal{P} = \{\mathcal{P}_1, \mathcal{P}_2, \ldots, \mathcal{P}_f\}$. Each partition incurs a cost $\mathcal{L}_i = \sum_{C \in \mathcal{P}_i} \mathcal{L}(C)$ with respect to the clustering objective. This gives us a plot of costs over the different values of $k$. Informally, the elbow method chooses the partition $\mathcal{P}_i$ whose cost $\mathcal{L}_i$ looks to be at a sharp point in this curve.

Due to the NP-hardness of $k$-means clustering, standard elbow plots can only compute approximate solutions, traditionally done sequentially for each $k$ [67]. However, the elbow method is surprisingly viable in the ultrametric setting: Theorem 4.2 yields optimal clusterings for all $k$ values at once, eliminating both computational overhead and approximation errors. Moreover, we show that the relaxed ultrametric's elbow plot for $(k, z)$-clustering is guaranteed to be convex:

**Corollary 5.1.** *Let $\mathcal{P}$ and $\mathcal{L}$ correspond to the $n$ partitions and losses obtained in accordance with Theorem 4.2 for the $(k, z)$-clustering objective. Let $\Delta_i = \mathcal{L}_{i+1} - \mathcal{L}_i$. Then either $\Delta_i < \Delta_{i+1} \leq 0$ or $\Delta_i = \Delta_{i+1} = 0$ for all $i \in [n-1]$.*

The idea here is that $\Delta_i$ represents the elbow plot's first derivative at $k = i$. Thus, Corollary 5.1 states that the elbow plot's slope is steepest at $k = 1$ and monotonically levels out to 0 as $k \to n$. We prove this in Appendix B.1, where we also specify how we determine the index of the elbow. We depict relaxed ultrametric elbow plots for $(k, z)$-clustering with $z = 1, 2, 3, 4, 5$ in Figure 3b. An alternative plot with the $x$-axis going to $n$ can be found in Figure 7 in the Appendix.

## 5.2 Additional Partitioning Techniques

Beyond the elbow method, there are several other standard approaches for selecting partitions and augmenting the hierarchy which can be applied to our hierarchies in $O(n)$ time. These are standard in the literature and we discuss them in more depth in Appendix B.

**Thresholding the Tree.** One can threshold the cluster hierarchy at a user-defined value $\varepsilon$, as is done in single-linkage clustering or DBSCAN [6]. This involves labeling internal nodes of the tree by their costs and returning all clusters with costs below $\varepsilon$, thus extracting partitions based on a similarity threshold rather than a specific $k$ value. This naturally extends to the $(k, z)$ setting: simply return the non-overlapping nodes in the $(k, z)$-clustering hierarchy below some cost threshold.

**Cluster Value Functions.** Rather than selecting based on costs, one can also assign a new value function to clusters and then choose the set of non-overlapping clusters that maximizes the sum of these values. For example, HDBSCAN uses the *stability* objective to measure how well clusters persist across the largest range of threshold values. Similarly, techniques in hierarchical segmentation define an energy function over the clusters [43, 56]. We show in Appendix B.4 that, given any reasonable such function, one can find its maximizing partition in $O(n)$ time via depth-first search.

**Handling Noise Points.** For applications requiring noise handling, subtrees consisting of outlier points can be pruned from the hierarchy without compromising the underlying ultrametric properties. For instance, with a minimum-cluster-size parameter $\mu$, clusters smaller than $\mu$ can be removed in $O(n)$ time, ensuring all clusters in the final partition meet size requirements [12, 47]. In practice, HDBSCAN's stability objective function is applied after noise points have been removed in this way.

## 5.3 Integrating Multiple Partitioning Methods

A key advantage of our approach is that, because the hierarchies and partitions can be extracted so quickly (as seen in Figure 3a), multiple hierarchies and partitions can be integrated with negligible runtime impact. We therefore introduce the *Median-of-Elbows* (MoE) algorithm which we find to

work well in practice. After fitting an ultrametric, MoE computes $(k, z)$-clustering hierarchies for $z = 1, 2, 3, 4, 5$ and applies the elbow method to each hierarchy. It then selects the median $k$ value from this set, representing the clustering cardinality that remains stable across different distance penalty settings. One can then use this $k$ value to extract the clustering from any hierarchy. Figure 3b visualizes this process on the D31 dataset, where MoE selects $k = 14$.

## 6 From Theory to Practice

Sections 3-5 introduce the components of our SHiP (Similarity-Hierarchy-Partitioning) clustering framework: fitting an ultrametric to capture the data similarities, choosing a hierarchy, and partitioning the data. Table 1 outlines various [ultrametric/hierarchy/partition] combinations. For example, [DC/$k$-median/thresholding] means that we are using the thresholding partition technique on the dc-dist's $k$-median hierarchy. In the following, we demonstrate that the SHiP framework includes diverse, effective clustering strategies and en-

Table 1: Example ultrametric, hierarchy, and partitioning options. Our SHiP framework allows one to pick any ultrametric, pair it with any clustering hierarchy, and extract any partition.

| Ultrametrics | Hierarchies | Partitioning Methods |
|---|---|---|
| DC tree | $k$-center | Ground Truth $k$ (GT) |
| Single Linkage tree | $k$-median | User-specified $k$ |
| Complete Linkage tree | $k$-means | Elbow |
| Cover tree | . . . | Median of Elbows (MoE) |
| KD tree | | Thresholding |
| HST-DPO | | Stability |
| | | . . . |

ables quickly finding the various data groupings that can be extracted from an ultrametric. This is particularly valuable for exploratory data analysis, where quickly switching between clustering methods and results is beneficial.

### 6.1 Ultrametrics

We now formally introduce the two families of ultrametrics which we use to evaluate our framework: HSTs and the density-connectivity distance. We note, however, that one can also apply our techniques to nearly the full suite of agglomerative clustering hierarchies [57], which we leave for future work.

**Hierarchically Well-Separated Trees (HSTs).** HSTs recursively subdivide metric spaces into trees such that each internal node is equidistant to all its descendant leaves. The distances in the metric space are then approximated using the tree's path distance. We use three types of HST:

1. **Cover trees** [9] define nested (potentially non-convex) cells so that points $p$ in cell $L_{i-1}$ and $q \in L_i$ (where $L_i \subset L_{i-1}$) satisfy $d(p, q) < 2^i$.
2. **KD trees** [8] recursively subdivide the space into axis-aligned hypercubes.
3. **HST-DPOs** [79] achieve optimal distance distortion by subdividing the space using metric balls.

Figure 4a demonstrates how the same hierarchy+partition ($k$-median with ground truth $k$) behaves across different HSTs (Cover tree, KD tree, HST-DPO), revealing how the underlying tree structure shapes clustering outcomes. Namely, Cover trees construct non-convex regions of similarity, KD trees produce axis-aligned clusterings, while HST-DPOs use spherical divisions. The first two require $O(n \log n)$ construction time (assuming constant dimensionality), while HST-DPOs achieve optimal distortion at the expense of $O(n^2)$ runtime. Although HSTs traditionally utilize the tree's path distances, this can be transformed to an LCA-tree representation in $O(n)$ time by assigning $d(\eta) = 2 \cdot d_{HST}(\eta, \ell)$ for internal node $\eta$ with any descendant leaf $\ell$. Going forward, we use Cover trees as our default HST, with KD trees and HST-DPOs extensively compared in Appendix E.

**Density-Connectivity Distance (dc-dist).** The second ultrametric we use, the dc-dist, forms the foundation of density-connected clustering algorithms such as DBSCAN and HDBSCAN. In short, it is the single-linkage distance over the pairwise *mutual-reachabilities* between the points:

**Definition 6.1** (Mutual reachability [24]). Let $(L, d')$ be a metric space, $x$ and $y$ be any two points in $L$, and $\mu \in \mathbb{Z}_{>0}$. Let $\kappa_\mu(x)$ be the distance from $x$ to its $\mu$-th closest neighbor in $L$. Then the *mutual reachability* between $x$ and $y$ is $m_\mu(x, y) = \max(d'(x, y), \kappa_\mu(x), \kappa_\mu(y))$.

**Definition 6.2** (dc-dist [6]). Let $(L, d')$ be a metric space, $x$ and $y$ be any two points in $L$, and $\mu \in \mathbb{Z}_{>0}$. Let $T$ be a minimum spanning tree over $L$'s pairwise mutual reachabilities. Let $p(x, y)$ be

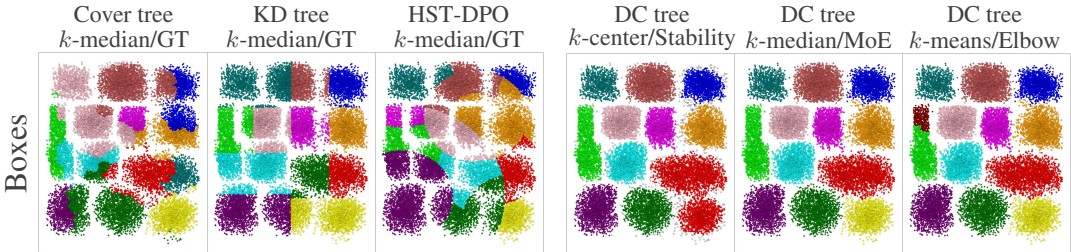

(a) Comparison of HST ultrametrics using the $k$-median hierarchy and the ground-truth value of $k$.

(b) Comparison of hierarchy/partition combinations on the dc-dist ultrametric.

Figure 4: Visualizations of the ultrametrics and hierarchy/partition combinations on the Boxes dataset.

the path in $T$ from $x$ to $y$ given by edges $\{e_i, \ldots, e_j\}$ in $T$. Lastly, let $|e|$ be the weight of any edge $e$ in $T$. Then the *dc-dist* between $x$ and $y$ is defined as

$$d_{dc}(x,y) = \max_{e \in p(x,y)} |e| \text{ if } x \neq y; \text{ else } m_\mu(x,x).$$

We note that Beer et al. [6] introduced a strictly ultrametric variant of the dc-dist where the distance of a point to itself was hardcoded to be 0. However, removing this condition as we do in Definition 6.2 simply makes the dc-dist a relaxed ultrametric (proof in B.5):

**Proposition 6.3.** *Let $(L, d')$ be a metric space. Then, the dc-dist over $L$ is a relaxed ultrametric.*

We refer to the dc-dist's LCA-tree representation as the DC tree. Note that for $\mu = 1$, the dc-dist is simply the single-link distance. We default to $\mu = 5$ for all experiments. Within our SHiP framework, we can *exactly* reproduce DBSCAN*[3] through [DC tree/$k$-center/thresholding] and HDBSCAN via [DC tree/$k$-center/stability].

### 6.2 Hierarchies and Partitioning Methods

Having fit an ultrametric, the SHiP framework allows users to explore multiple alternative groupings of the data through its many hierarchies and partitions. Each hierarchy represents a different perspective on how points in the ultrametric space should be organized, essentially defining a search space of possible clusterings. Partitioning methods then extract specific clusterings from these hierarchies, highlighting different structural characteristics. Our experiments focus on three illustrative hierarchy/partition combinations: $k$-center/stability, $k$-median/MoE and $k$-means/elbow.

Figure 4b offers an illustrative example of how these hierarchies and partitions interact. For example, the $k$-center hierarchy merges clusters based on distances to their centers, regardless of point count. When combined with the stability objective, this finds clusters which are well-separated over a wide range of thresholds. Consequently, in the Boxes dataset, $k$-center/stability merges the two red boxes since they share a density-connected path (and removes some outliers as noise).

In contrast, the $k$-median and $k$-means hierarchies strike a balance between the distances to the centers and the cluster cardinalities. For example, under the dc-dist, the $k$-median and $k$-means hierarchies evaluate how "spread out" a cluster is: the cost is low when there are a few points which are densely connected (have small gaps between dense regions). This explains why $k$-median/MoE and $k$-means/elbow separate the two red boxes in the last two columns of Figure 4b that $k$-center merges: they contained too many points with too large a gap between them. Figure 8 in the Appendix shows an example on the D31 dataset with more pronounced differences between the methods.

## 7 Experiments

We now verify the practical utility of our SHiP clustering framework by showcasing its speed and competitive outputs across ultrametrics and datasets. Table 4 in the Appendix gives an overview

---

[3]DBSCAN* is a variant of DBSCAN which treats points on the borders of clusters as noise [12].

Table 2: Runtimes of our SHiP framework's components (first three column groups) and competitors (last column group) in (minutes+)seconds (*s*), or milliseconds (*ms*). Computation times of the ultrametrics are comparable to the runtimes of the competitor algorithms. I.e., building the DC tree is on par with other density-based methods (highlighted in blue). Computation of the cluster hierarchies and partitioning methods then takes only *milliseconds*. Full version: Table 5 in the Appendix.

| | Dataset | ultrametric | | hier. | partitioning | | | competitors | | | | | |
| | | Cover tree | DC tree | $k$-means | Stab. | MoE | Elbow | $k$-means $(k = \text{GT})$ | $k$-means $(k = 500)$ | SCAR | Ward | AMD-DBSCAN | DPC |
|---|---|---|---|---|---|---|---|---|---|---|---|---|---|
| Tabular Data | Boxes | 0.059 *s* | 14.239 *s* | 33 *ms* | 3 *ms* | 134 *ms* | 2 *ms* | 0.114 *s* | 1.217 *s* | 1.481 *s* | 10.808 *s* | 1+04.670 *s* | 12+10.358 *s* |
| | D31 | 0.004 *s* | 0.327 *s* | 4 *ms* | 0 *ms* | 15 *ms* | 0 *ms* | 0.289 *s* | 0.509 *s* | 0.620 *s* | 0.153 *s* | 0.878 *s* | 12.816 *s* |
| | airway | 0.027 *s* | 4.997 *s* | 21 *ms* | 1 *ms* | 85 *ms* | 1 *ms* | 0.122 *s* | 0.392 *s* | 0.651 *s* | 5.243 *s* | 16.822 *s* | 6+46.726 *s* |
| | lactate | 0.161 *s* | 47.731 *s* | 68 *ms* | 6 *ms* | 275 *ms* | 4 *ms* | 0.126 *s* | 1.997 *s* | 2.612 *s* | 59.992 *s* | 7+43.293 *s* | 69+34.012 *s* |
| | HAR | 11.053 *s* | 23.144 *s* | 14 *ms* | 1 *ms* | 56 *ms* | 0 *ms* | 2.248 *s* | 8.478 *s* | 0.658 *s* | 18.448 *s* | 30.129 *s* | 3+45.745 *s* |
| | letterrec. | 0.322 *s* | 9.076 *s* | 22 *ms* | 2 *ms* | 87 *ms* | 1 *ms* | 3.386 *s* | 3.521 *s* | 2.133 *s* | 10.309 *s* | 37.055 *s* | 11+42.910 *s* |
| | PenDigits | 0.067 *s* | 2.566 *s* | 13 *ms* | 1 *ms* | 53 *ms* | 0 *ms* | 1.001 *s* | 2.725 *s* | 0.490 *s* | 3.281 *s* | 9.254 *s* | 3+21.409 *s* |
| Image Data | COIL20 | 3.658 *s* | 14.817 *s* | 1 *ms* | 0 *ms* | 4 *ms* | 0 *ms* | 1.637 *s* | 13.524 *s* | 0.310 *s* | 8.941 *s* | 2.378 *s* | 11.710 *s* |
| | COIL100 | 3+52.397 *s* | 14+50.661 *s* | 10 *ms* | 0 *ms* | 39 *ms* | 0 *ms* | 32.122 *s* | 1+59.225 *s* | 7.964 *s* | 12+04.037 *s* | 2+58.177 *s* | 14+18.780 *s* |
| | cmu_faces | 0.046 *s* | 0.238 *s* | 1 *ms* | 0 *ms* | 2 *ms* | 0 *ms* | 0.206 *s* | 0.661 *s* | 0.116 *s* | 0.064 *s* | 0.346 *s* | 0.515 *s* |
| | OptDigits | 0.335 *s* | 1.420 *s* | 6 *ms* | 0 *ms* | 24 *ms* | 0 *ms* | 0.502 *s* | 1.154 *s* | 0.290 *s* | 0.974 *s* | 3.073 *s* | 44.270 *s* |
| | USPS | 2.924 *s* | 8.670 *s* | 11 *ms* | 1 *ms* | 45 *ms* | 0 *ms* | 2.709 *s* | 8.493 *s* | 1.433 *s* | 6.092 *s* | 29.259 *s* | 1+48.607 *s* |
| | MNIST | 16+24.220 *s* | 37+05.095 *s* | 120 *ms* | 15 *ms* | 491 *ms* | 8 *ms* | 4.320 *s* | 3+29.969 *s* | 15.352 *s* | 19+02.498 *s* | 17+21.183 *s* | - |

of the datasets we use. We evaluate the clustering quality with the adjusted rand index (ARI) [36], treating points labeled as noise as singleton clusters. NMI [70], AMI [74] and correlation coefficient [29] results can be found in Appendix E. Both our runtime and accuracy[4] tables report the mean over 10 runs. All experiments were performed on 2x Intel 6326 with 16 cores each and 512GB RAM.

## 7.1 Runtimes

In practice, clustering is an exploratory data mining task that requires several runs of different methods or using different hyperparameter settings. Especially when done sequentially, this requires substantial computational time and resources. In contrast, our SHiP clustering framework requires just a single upfront ultrametric computation, after which we can generate a myriad of different clustering solutions with negligible additional runtime. Table 2 and Figure 3a highlight this efficiency. The first column group in Table 2 shows that computing our ultrametrics (Cover tree and DC tree) has a runtime comparable to that of the corresponding standard clustering algorithms (last column group). Specifically, we show on the right (under competitors) the time required for Euclidean $k$-means with ground-truth $k$, and Euclidean $k$-means with $k = 500$,[5] Ward agglomerative clustering [77], an adaptive multi-density DBSCAN version (AMD-DBSCAN) [76], density peaks clustering (DPC) [66], and an accelerated version of spectral clustering (SCAR) [33]. However, once the ultrametric has been computed, generating different hierarchies (column group 2) and partitioning methods (column group 3) requires only *milliseconds*. Thus, users can explore many different clustering solutions in essentially the same time it takes to run a traditional clustering algorithm.

## 7.2 Clustering Quality

This efficiency in switching between clusterings is particularly valuable given the results shown in Table 3. Here, we study the quality of $k$-center/stability, $k$-median/MoE, and $k$-means/elbow on the DC- and Cover tree ultrametrics and compare them against the competitor algorithms. Importantly, $k$-means, Ward, and SCAR are all given the ground-truth $k$ to maximize their competitiveness. In comparison, even in the more realistic setting where the true number of clusters is unknown, we can achieve better clustering performance than they do using the DC tree. However, there is no best hierarchy/partitioning combination. Although $k$-means/elbow performs best in many cases, it is not universally superior to the other combinations; different pairings excel on different datasets. This underscores the value of rapidly switching between different hierarchies and also partitioning methods.

Notably, Cover tree combinations consistently perform worse than Euclidean $k$-means, with comparable results for KD trees and HST-DPOs in Appendix E. This raises the question of whether HSTs are

---

[4]For non-deterministic algorithms, we also provide the standard deviation for the clustering accuracy.

[5]Note that the SHiP clustering framework simultaneously obtains the optimal solutions for every value of $k$.

always well-suited for representing the underlying cluster structure in data, and under what conditions their use can effectively accelerate machine learning pipelines [18, 44, 53, 73].

If the ground truth number of clusters is given to our framework, $k$-median and $k$-means over the dc-dist (essentially density-connected $k$-median and $k$-means) achieve even slightly better results (see Table 6; Appendix). Furthermore, we point out that Ward agglomerative clustering is ultrametric in nature and, therefore, falls under the umbrella of our framework [57].

Table 3: ARI values for the SHiP framework on the DC and Cover tree ultrametrics (resp. first/second column group). ARI values for competitor algorithms are in the third column group. Euclidean $k$-means, SCAR, and Ward are given the ground-truth $k$ value. Full version: Table 6 in the Appendix.

| | Dataset | DC tree | | | Cover tree | | | competitors | | | | |
| | | $k$-center Stability | $k$-median MoE | $k$-means Elbow | $k$-center Stability | $k$-median MoE | $k$-means Elbow | Eucl. $k$-means | SCAR | Ward | AMD-DBSCAN | DPC |
|---|---|---|---|---|---|---|---|---|---|---|---|---|
| Tabular Data | Boxes | 90.1 | **99.3** | 97.9 | 2.6 | $42.1 \pm 4.7$ | $24.2 \pm 1.6$ | $93.5 \pm 4.3$ | $0.1 \pm 0.1$ | 95.8 | 63.9 | 25.9 |
| | D31 | 79.7 | 42.7 | 82.9 | $46.5 \pm 1.8$ | $62.0 \pm 5.4$ | $67.7 \pm 3.2$ | $\mathbf{92.0 \pm 2.7}$ | $41.7 \pm 5.4$ | **92.0** | 86.4 | 18.5 |
| | airway | 38.0 | **65.9** | 58.8 | 0.8 | $18.2 \pm 2.4$ | $12.0 \pm 1.4$ | $39.9 \pm 2.0$ | $-0.9 \pm 0.5$ | 43.7 | 31.7 | 65.1 |
| | lactate | 41.0 | 41.0 | 67.5 | 0.1 | $4.1 \pm 0.6$ | $1.7 \pm 0.2$ | $28.6 \pm 1.1$ | $1.5 \pm 1.0$ | 27.7 | **71.5** | 0.0 |
| | HAR | 30.0 | 46.9 | 52.8 | $14.7 \pm 8.8$ | $14.2 \pm 4.7$ | $9.6 \pm 2.2$ | $46.0 \pm 4.5$ | $5.5 \pm 3.2$ | 49.1 | 0.0 | 33.2 |
| | letterrec. | 12.1 | 16.6 | 17.9 | $5.8 \pm 0.2$ | $7.2 \pm 0.6$ | $6.2 \pm 0.3$ | $12.9 \pm 0.6$ | $0.4 \pm 0.1$ | $14.7 \pm 0.9$ | 7.9 | 0.0 |
| | PenDigits | 66.4 | 73.1 | 75.4 | $8.0 \pm 0.8$ | $12.0 \pm 0.6$ | $8.9 \pm 0.5$ | $55.3 \pm 3.2$ | $0.9 \pm 0.3$ | 55.2 | 55.6 | $28.8 \pm 1.1$ |
| Image Data | COIL20 | **81.2** | 72.8 | 72.6 | $46.4 \pm 4.4$ | $46.6 \pm 2.1$ | $47.7 \pm 2.0$ | $58.2 \pm 2.8$ | $33.5 \pm 2.0$ | 68.6 | 39.2 | $35.9 \pm 0.1$ |
| | COIL100 | **80.1** | 66.8 | 70.0 | $44.6 \pm 4.2$ | $46.6 \pm 1.5$ | $50.1 \pm 1.2$ | $56.1 \pm 1.4$ | $16.7 \pm 0.8$ | 61.4 | 14.2 | 0.2 |
| | cmu_faces | 60.2 | 56.6 | **66.5** | $8.6 \pm 3.1$ | $37.1 \pm 4.1$ | $34.2 \pm 2.1$ | $53.2 \pm 4.7$ | $38.5 \pm 2.9$ | 61.6 | 0.7 | 0.6 |
| | OptDigits | 55.3 | **77.0** | 77.0 | $40.9 \pm 3.5$ | $20.9 \pm 2.3$ | $18.1 \pm 2.4$ | $61.3 \pm 6.6$ | $14.4 \pm 4.1$ | $74.6 \pm 2.4$ | 63.2 | 0.0 |
| | USPS | 33.7 | 29.3 | 29.3 | $12.0 \pm 1.7$ | $8.7 \pm 1.0$ | $11.2 \pm 1.5$ | $52.3 \pm 1.7$ | $2.9 \pm 0.9$ | **63.9** | 0.0 | 21.0 |
| | MNIST | 19.7 | 41.7 | 46.0 | $11.1 \pm 1.7$ | $5.4 \pm 0.6$ | $5.4 \pm 0.6$ | $36.9 \pm 1.0$ | $1.3 \pm 0.4$ | 52.7 | 0.0 | - |

## 8    Conclusions and Limitations

This paper proposed and evaluated a generalization of standard hierarchical clustering. Our theoretical contributions suggest new algorithms for hierarchical clustering under, for example, the Dasgupta objective [22], as well as connections to spectral clustering [49]. We leave both to future work. Furthermore, although our approach enables $\texttt{Sort}(n)$-time access to novel clustering hierarchies after fitting an ultrametric, the initial ultrametric computation remains the performance bottleneck. A key limitation is that the clustering quality depends significantly on this choice of ultrametric. I.e., we find that the dc-dist consistently produces high-quality clusterings across different hierarchy/partition combinations while HSTs produce subpar results. We therefore suggest that future work focuses on developing fast algorithms for fitting ultrametrics which effectively model the data.

## Acknowledgements

We thank Chris Schwiegelshohn, who provided advice and insights when developing the theoretical framework surrounding Theorem A.8.

We gratefully acknowledge financial support from the Vienna Science and Technology Fund (WWTF-ICT19-041) and the Austrian funding agency for business-oriented research, development, and innovation (FFG-903641[6]). Andrew Draganov is partially supported by the Independent Research Fund Denmark (DFF) under a Sapere Aude Research Leader grant No 1051-00106B and by project W2/W3-108 Impuls und Vernetzungsfonds der Helmholtz-Gemeinschaft. We also acknowledge funding from Danish Pioneer Centre for AI[7], DNRF grant number P1.

---

[6]https://projekte.ffg.at/projekt/4814676
[7]https://aicentre.dk

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

# A  Proofs for Sections 3 and 4

In this section, we prove Theorem 3.3, Corollary 3.4, and Theorem 4.2 from the main body of the paper. We restate each here:

**Theorem 3.3.** *Let $(L, d')$ be a finite relaxed ultrametric space. Then there exists LCA-tree $T$ with LCA-distance $d$ and a bijection $f : L \leftrightarrow leaves(T)$ such that, for all $\ell_i, \ell_j \in L$, $d'(\ell_i, \ell_j) = d(f(\ell_i) \vee f(\ell_j))$.*

**Corollary 3.4.** *Let $T$ be an LCA-tree. For any leaf $\ell \in T$, let $p(\ell) = [\ell, \eta_a, \ldots, \eta_b, r(T)]$ be the path from $\ell$ to the root of the tree $r(T)$. Then the LCA-distances on $T$ form a relaxed ultrametric if and only if, for all $\ell \in leaves(T)$ and $\eta_i, \eta_j \in p(\ell)$, the following conditions are satisfied:*

$$(1) \quad d(\ell) \geq 0 \quad and \quad (2) \quad \eta_i \preceq \eta_j \implies d(\eta_i) \leq d(\eta_j).$$

**Theorem 4.2.** *Let $(L, d)$ be a finite relaxed ultrametric space represented over an LCA-tree $T$. Let $n = |L|$ and $z \in \mathbb{N}_{>0}$. Then, for both the $k$-center and $(k, z)$-clustering objectives on $T$, there exists an algorithm which finds the optimal solutions $\{\mathbf{C}_1, \ldots, \mathbf{C}_n\}$ for all $k \in [n]$ in $\mathtt{Sort}(n)$ time. Furthermore, the respective partitions $\mathcal{H} = \{\mathcal{P}_1, \ldots, \mathcal{P}_n\}$ obtained by assigning all leaves in $T$ to their closest center satisfy Definition 4.1.*

## A.1  Overview of Ideas

We begin with a more thorough proof outline to the one that appears in the main body of the paper.

First, we will define a relaxation of ultrametrics and show a few immediate properties of these spaces. Their key property is that there is essentially an equivalence relation induced by any relaxed ultrametric: for any distance value $d$, the sets of points that are within distance $d$ of each other partition the space. This is a generalization of the ideas in Carlsson et al. [14].

We then show that all relaxed ultrametrics can be represented as lowest-common-ancestor-trees (LCA-trees). These are rooted trees where every node has a value associated with it. The distance between two leaves in an LCA-tree is the value in their lowest common ancestor. Furthermore, all LCA-trees are (relaxed) ultrametrics if the values are monotonically non-decreasing along the path from any leaf to the root (Corollary 3.4). As a result, there is a bijective relationship between LCA-trees and ultrametrics (Theorem 3.3). We proceed by only considering ultrametrics in the LCA-tree data structure.

We now consider the center-based clustering objectives over these LCA-trees. Here, a centers are placed on leaves of the tree and the point-to-center distances are given by the LCA-distances. For $k$-center, we will see that the well-known strategy of farthest-first traversal [31] (which achieves a 2-approximation in Euclidean space) is actually optimal in ultrametrics. This algorithm works by placing the first center on a random leaf and then, for each subsequent center, placing it on the node which induces the highest cost. The intuition here is that the farthest-first traversal's standard 2-approximation is a direct consequence of the triangle inequality, so changing this to the strong triangle inequality resolves the approximation constant.

We now reduce the algorithm of farthest-first traversal in an ultrametric to sorting the nodes in the LCA-tree. By placing this first center on any leaf, every other leaf in the tree gets mapped to it. Thus, the largest point-center distance corresponds to two leaves whose LCA is the root of the tree. We must therefore place our subsequent centers on leaves until there is no point-center distance which goes through the root of the LCA-tree. This is done by placing centers in subtrees in which there are no placed centers. After the root has been handled, we will place centers in accordance with the root's children, focusing on the one which has largest value first. By Corollary 3.4, it must be the case that the LCA-tree's values grow as we approach the root. Thus, the runtime being $\mathtt{Sort}(n)$ comes from the fact that the farthest-first traversal algorithm "fills in" the tree from the root down. As a result, we will solve $k$-center in an ultrametric by sorting the distances from largest to smallest and placing the corresponding centers.

Given this, we will conclude the proof by showing how to reduce the general problem of $(k, z)$-clustering (such as $k$-means) to this $k$-center algorithm. We will consider these through the lens of the *cost-decreases* of placing $k$-means or $k$-median centers in an ultrametric. That is, if we have an optimal solution for some value of $k$, then the optimal solution for $k + 1$ centers will have a lower cost. Thus, there is a cost-decrease associated with placing each optimal center. Our key observation

is that these cost-decreases *themselves* satisfy the strong triangle inequality. We will also show that greedily choosing centers that maximize these cost-decreases gives optimal $k$-means and $k$-median solutions in an ultrametric. Putting the pieces together, these results imply that we can apply the $k$-center algorithm in the LCA-tree of cost-decreases to get optimal $k$-means and $k$-median solutions.

**Notation.**  We use $T$ to represent an arbitrary rooted tree with root $r$. We use $\eta$ to represent arbitrary internal nodes and $\ell$ to represent arbitrary leaves. We also assume that every internal node $\eta$ in the tree is equipped with a value, which we write by $d(\eta)$. We use the notation $\eta_i \preceq \eta_j$ to indicate that $\eta_j$ lies on the path from $\eta_i$ to $r$. We use children$(\eta)$ and parent$(\eta)$ to indicate the direct children and parent of a node. Lastly, we use the notation from [22] with $\ell_i \vee \ell_j$ denoting the lowest common ancestor (LCA) of a set of nodes/leaves and $T[\eta]$ denoting the subtree rooted at $\eta$. Thus, $T[\ell_i \vee \ell_j]$ is the smallest subtree containing both $\ell_i$ and $\ell_j$.

For notation on clustering, we define $(k, z)$-clustering as finding the set of centers $\mathbf{C} \in T$ with $|\mathbf{C}| = k$ which minimize

$$\text{Cost}_z(T, \mathbf{C}) = \sum_{\ell \in \text{leaves}(T)} \min_{c \in \mathbf{C}} d(\ell, c)^z.$$

This corresponds to $k$-median and $k$-means for $z = 1$ and $z = 2$, respectively. We also define $k$-center clustering as finding the $\mathbf{C}$ which minimizes

$$\text{Cost}_\infty(T, \mathbf{C}) = \max_{\ell \in \text{leaves}(T)} \min_{c \in \mathbf{C}} d(\ell, c).$$

## A.2  Ultrametrics and LCA-Trees

In this section, we introduce relaxed ultrametrics and prove Theorem 3.3 and Corollary 3.4.

Throughout the literature, the stand-out candidate for a hierarchical (dis-)similarity measure is the *ultrametric*. We will, however require a looser definition, which we refer to as a *relaxed ultrametric*:

**Definition 3.1.** Let $L$ be a set. Then $d : L \times L \to \mathbb{R}_{\geq 0}$ is a *relaxed ultrametric* over $L$ if, for all $\ell_i, \ell_j, \ell_k \in L$, the following conditions are satisfied:

$$(1)\ d(\ell_i, \ell_j) = d(\ell_j, \ell_i) \geq 0 \quad \text{and} \quad (2)\ d(\ell_i, \ell_k) \leq \max(d(\ell_i, \ell_j), d(\ell_j, \ell_k)).$$

We will prove our results over these relaxed ultrametrics. We note that the set of ultrametrics is a *subset* of the set of relaxed ultrametrics. That is, strict ultrametrics have the additional condition that the distance between two points is $0$ if and only if they are the same point. Thus, our theoretical results immediately apply to all ultrametrics.

The *strong triangle inequality* in Definition 3.1 is what allows ultrametrics to capture hierarchical relationships. Specifically, the strong triangle inequality implies that any three points in an ultrametric space must form an isosceles triangle with an angle less than 60 degrees:

**Fact A.1.** *Let $d$ be a dissimilarity measure on a space $L$, which satisfies the strong triangle inequality. Then for any $\ell_i, \ell_j, \ell_k \in L$, one of the following holds:*

1. $d(\ell_i, \ell_j) \leq d(\ell_i, \ell_k) = d(\ell_j, \ell_k)$

2. $d(\ell_i, \ell_k) \leq d(\ell_i, \ell_j) = d(\ell_j, \ell_k)$

3. $d(\ell_j, \ell_k) \leq d(\ell_i, \ell_j) = d(\ell_i, \ell_k)$

*Proof.* Assume that all three are unequal, so WLOG $d(\ell_i, \ell_j) < d(\ell_i, \ell_k) < d(\ell_j, \ell_k)$. Then the strong triangle inequality does not hold, since $d(\ell_j, \ell_k) \not\leq \max(d(\ell_i, \ell_j), d(\ell_i, \ell_k))$.

Similarly, assume that the singleton edge is longer than the two others, i.e. $d(\ell_i, \ell_k) = d(\ell_j, \ell_k) < d(\ell_i, \ell_j)$. This also breaks the strong triangle inequality, since $d(\ell_i, \ell_j) \not\leq \max(d(\ell_i, \ell_j), d(\ell_j, \ell_k))$. $\qquad \square$

As a result, for any three points in a relaxed ultrametric space, knowing two of the pairwise distances is sufficient to give the ordering of all three. For example, if two of the distances in a triangle are equal, then the third must be equal to this distance or smaller.

This naturally extends to groups of more than 3 points. Consider the case where we have $n$ points $\{x_1, x_2, \ldots, x_n\}$ so that for any $x_i, x_j$, we have ultrametric distance $d(x_i, x_j) < \varepsilon$ for any small $\varepsilon > 0$. Now let $y$ be a point with $d(x_1, y) = \delta$ with $\delta \gg \varepsilon$. Since the $x$'s are all close to one another, Fact A.1 implies that $d(x_i, y)$ must also equal $\delta$ for all $i$. This has the following fundamental consequences:

**Theorem 3.3.** *Let $(L, d')$ be a finite relaxed ultrametric space. Then there exists LCA-tree $T$ with LCA-distance $d$ and a bijection $f : L \leftrightarrow leaves(T)$ such that, for all $\ell_i, \ell_j \in L$, $d'(\ell_i, \ell_j) = d\left(f(\ell_i) \vee f(\ell_j)\right)$.*

We first put this in context before giving its proof. Theorem 3.3 states that any ultrametric can be represented over the leaves of a tree, with the property that the distance between two leaves is uniquely determined by the value in their LCA. We note that variants of this theorem have been given elsewhere [45, 57, 58]; nonetheless, the results later in this section require the form given above. We refer to this data structure as an LCA-tree and the "distances" in this tree as LCA-distances.

*Proof.* We use Fact A.1 to design an algorithm to construct the tree $T$ by repeatedly splitting the relaxed ultrametric over its largest distance $d_{max}$. First, let us see that $d_{max}$ induces a partition over $L$. Let $\ell_i \in L$ be any point and let $L_{\ell_i} = \{\ell_j : d(\ell_i, \ell_j) < d_{max}\} \cup \{\ell_i\}$ be the set consisting of $\ell_i$ and all the points closer to it than $d_{max}$. Now let $\ell_k \in L \setminus L_{\ell_i}$ be any other point not in $L_{\ell_i}$. By definition $d(\ell_i, \ell_k) = d_{max}$. Furthermore, by Fact A.1, $d(\ell_j, \ell_k) = d_{max}$ for all $\ell_j \in L_{\ell_i}$. Thus, $d_{max}$ induces a partition on a relaxed ultrametric where, across any two points in distinct clusters, the distance is necessarily $d_{max}$.

We now use this idea to devise an algorithm that embeds the ultrametric in an LCA-tree. This algorithm receives a relaxed ultrametric space $(L, d)$ as input. It begins by instantiating a (initially empty) root node $r$ which will serve as the root of the LCA-tree. Let $d_{max}$ be the largest distance in $(L, d)$ and let $\mathcal{P}$ be the partition induced by $d_{max}$. Assign $d(r) = d_{max}$. For each group in the partition, make a node and assign it as a child of the root $r$.

We apply this construction recursively for each of the children. The base case occurs when $L$ has either one or two points. If there is only one point, $x_i \in L$, we simply return a leaf $\ell_i$. This leaf is given weight $d(\ell_i) = d(\ell_i, \ell_i)$ and we define $f(x_i) = \ell_i$. If $L$ has two points, $x_i$ and $x_j \in L$, then we create two leaves $\ell_i$ and $\ell_j$ as children of the input node. The mapping is arbitrarily defined as $f(x_i) = \ell_i$ and $f(x_j) = \ell_j$. We assign the input node with weight $d(\eta) = d(x_i, x_j)$ and give the leaves weights $d(\ell_i) = d(\ell_i, \ell_i)$ and $d(\ell_j) = d(\ell_j, \ell_j)$.

We inductively verify that this construction produces a valid LCA-tree and that all distances in the relaxed ultrametric are preserved by the LCA-distances. In the base case, the space $(L, d)$ has either one or two points. We respectively represent these as a singleton root node or a rooted tree with two children. In both cases, the value of the root is the distance between the two points and the distance of each point to itself is the value in the leaves. Thus, in the base case, all pairwise distances in $L$ are preserved via the LCA-distances.

In the inductive step, assume that $(L, d)$ has more than two points, that the maximal distance is $d_{max}$, and that all smaller distances are represented via distinct trees. By the above logic, the distance between any two nodes in separate trees must be $d_{max}$. Since the construction described above assigns the value $d_{max}$ to the root and assigns the existing trees to it as children, this new node is a parent to the already-existing trees. Thus, their internal LCA-trees and LCA-distances are not affected. However, the LCA between any nodes in separate subtrees has value $d_{max}$. Therefore, the entire ultrametric $(L, d)$ is preserved.

$\square$

The following corollary gives the sufficient conditions for an LCA-tree to correspond to a relaxed ultrametric:

**Corollary 3.4.** *Let $T$ be an LCA-tree. For any leaf $\ell \in T$, let $p(\ell) = [\ell, \eta_a, \ldots, \eta_b, r(T)]$ be the path from $\ell$ to the root of the tree $r(T)$. Then the LCA-distances on $T$ form a relaxed ultrametric if and*

*only if, for all* $\ell \in leaves(T)$ *and* $\eta_i, \eta_j \in p(\ell)$, *the following conditions are satisfied:*

$$(1) \quad d(\ell) \geq 0 \quad and \quad (2) \quad \eta_i \preceq \eta_j \implies d(\eta_i) \leq d(\eta_j).$$

*Proof.* We first assume that we have an LCA-tree satisfying the conditions and show that it must be a relaxed ultrametric. First, note that the LCA-distances in the tree satisfy symmetry by the definition of LCA: $d(\ell_i, \ell_j) = d(\text{LCA}(\ell_i, \ell_j)) = d(\ell_j, \ell_i)$. Furthermore, conditions (1) and (2) together ensure the non-negativity conditions required for a relaxed ultrametric. Therefore, it remains to show the strong triangle inequality.

Let $\ell_i, \ell_j$ and $\ell_k$ be three leaves in the LCA-tree. If they all have the same LCA (i.e., $\ell_i \vee \ell_j = \ell_j \vee \ell_k = \ell_i \vee \ell_k$), then the leaves are equidistant in the LCA-tree, and the strong triangle inequality is satisfied. Thus, assume WLOG that $\ell_i \vee \ell_j \preceq \ell_i \vee \ell_k$. This implies that $\ell_i \vee \ell_k = \ell_j \vee \ell_k$. Consequently, the strong triangle inequality is satisfied:

$$\ell_i \vee \ell_j \preceq \ell_i \vee \ell_k = \ell_j \vee \ell_k \underset{\substack{\uparrow \\ \text{By Assumption}}}{\implies} d(\ell_i \vee \ell_j) \leq d(\ell_i \vee \ell_k) = d(\ell_j \vee \ell_k).$$

We now prove the other direction of the if-and-only-if: we assume an LCA-tree whose LCA-distances are a relaxed ultrametric and show that it must satisfy conditions 1 and 2. By the non-negativity of the ultrametric, the condition $d(\ell) \geq 0$ is satisfied for all leaves $\ell$. Regarding the second condition, assume for the sake of contradiction that it is not satisfied. Then we must have nodes $\eta_i$ and $\eta_j$ with $\eta_i \preceq \eta_j$ and $d(\eta_i) > d(\eta_j)$. We now choose leaves $\ell_i, \ell'_i$ and $\ell_j$ so that $\ell_i \vee \ell'_i = \eta_i$ and $\ell_i \vee \ell_j = \ell'_i \vee \ell_j = \eta_j$. Then we have $d(\ell_i, \ell_j) = d(\ell'_i, \ell_j) < d(\ell_i, \ell'_i)$.

This is in violation of Fact A.1 and therefore contradicts our assumption that the LCA-distances are ultrametric.

$\square$

As a result of Corollary 3.4, if we wish to show that a tree's LCA-distances satisfy the strong triangle inequality, we need to show that the tree's values are non-decreasing on paths from the leaves to the root. Going forward, every discussion of ultrametrics will implicitly be through their LCA-tree representation.

### A.3 $k$-Center in Ultrametrics

#### A.3.1 Structure of a Center-Based Solution

We first describe how cluster memberships are defined in an LCA-tree. Recall that centers are placed on leaves and a cluster is the set of points which is closest to a center. Since many of the center-to-leaf relationships are equidistant in an LCA-tree, we use the "marking" procedure from [18] to define a consistent notion of cluster attribution.

Let $\mathbf{C} = [c_1, \ldots c_k]$ be $k$ arbitrarily ordered centers that correspond to distinct leaves in the LCA-tree. We obtain the cluster memberships $C_i = \{\ell \in T : c_i = \arg\min_{c \in \mathbf{C}} d(\ell, c)\}$ by adding the centers in the given order and, for each center placed, marking the nodes in the tree from the corresponding leaf to its lowest unmarked ancestor. Thus, if we place center $c_i$ in a leaf node, we go up the tree towards the root and mark every node with "$C_i$" until we hit a previously marked node. Leaves are then assigned to clusters by finding their lowest marked ancestor. An example is shown in Figure 5a, where we first place a center in subtree $T_1$ and mark every node on the path from the center to the root with $C_1$. We then add the second center in subtree $T_2$ and mark along its path to the root until we reach an already-marked node. We therefore have that the leaves in $T_1 \cup T_3$ belong to cluster $C_1$ and the leaves in $T_2$ belong to cluster $C_2$.

Put simply, *if a node $\eta$ is marked, then there is a center in $T[\eta]$*. We note that this attribution of leaves to centers applies in both the $k$-center and the $(k, z)$-clustering settings and is independent of the algorithm used to obtain the centers.

Interestingly, the number of optimal center-based clustering solutions in any LCA-tree is necessarily exponential in $k$. This is shown in Figure 5b: the set of distances from leaves in the tree to $c_i$ does not change regardless of whether we place $c_i$ on $\ell_i, \ell_j$ or $\ell_k$. When describing optimal solutions,

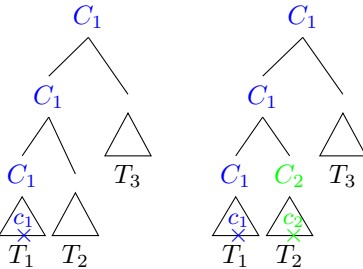 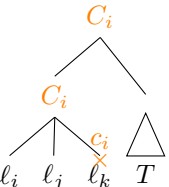

(a) A visualization of the marking procedure. We first place center $c_1$ and mark every node from it to the root. We then place center $c_2$ and mark every node from it to its lowest unmarked ancestor.

(b) The cost of the clustering does not depend on which leaf the center $c_i$ is assigned to.

Figure 5: A visualization of how leaves get assigned to centers in the LCA-tree.

we consider them equivalent up to such swaps in the last layer of the tree. We discuss heuristics for choosing the "best" of these optimal solutions in Appendix C.1.2.

We lastly formalize the notion of *hierarchical* clusters:

**Definition 4.1.** (Lin et al. [48]) A *cluster hierarchy* $\mathcal{H} = \{\mathcal{P}_1, \ldots, \mathcal{P}_n\}$ is a set of partitions where

for $k = 1$ :      $\mathcal{P}_1 \subseteq L$, and

for $1 < k \leq n$ :      $\mathcal{P}_k = (\mathcal{P}_{k-1} \setminus C_i) \cup \{C_j, C_l\}$, such that (a) $C_i = C_j \cup C_l \in \mathcal{P}_{k-1}$ with $C_j \cap C_l = \emptyset$, (b) $C_j \neq C_i$ and $C_l \neq C_i$, and (c) $i \neq j \neq l$.

In short, a hierarchical set of solutions means that the clustering in $\mathcal{P}_k$ is the same as the one at $\mathcal{P}_{k-1}$ except that a single cluster was split apart. We say a cluster $C_i \in \mathcal{H}$ if $\exists \, \mathcal{P}_j \in \mathcal{H}$ with $C_i \in \mathcal{P}_j$.

### A.3.2    $k$-Center in LCA-trees

We now develop our center-based clustering algorithms in LCA-trees, starting with the $k$-center clustering task. Recall that the $k$-center objective requires finding $k$ centers that minimize

$$\text{Cost}_\infty(T, \mathbf{C}) = \max_{\ell \in T} \min_{c \in \mathbf{C}} \text{Dist}(\ell, c).$$

Using the marking construction, we can succinctly state the cost of any $k$-center solution in an LCA-tree:

**Fact A.2.** *Let $\mathbf{C}$ be a set of centers in an LCA-tree $T$. Let $\eta$ be the most expensive marked node in $T$ with at least one unmarked child. Then $\text{Cost}_\infty(T, \mathbf{C}) = d(\eta)$.*

*Proof.* We will show a bijection between the leaf-to-center distances induced by solution $\mathbf{C}$ and the nodes which are marked and have an unmarked child. Since the cost of a $k$-center solution is the maximum over all such leaf-to-center distances, it will therefore be the maximum-valued such node.

First, we show that all leaf-to-center distances are contained in nodes which are both (a) marked and (b) have an unmarked child. Let $c$ be any center in $\mathbf{C}$ and let $\ell$ be any leaf that is closest to center $c$, i.e., $c = \arg\min_{c' \in \mathbf{C}} d(\ell, c')$. Let $\eta = c \vee \ell$. Then condition (a) is true of $\eta$ by definition of the marking procedure: notice that $c \vee \ell_j$ is marked for all leaves $\ell_j \in T$. To see why (b) is true, consider that if every child of $\eta$ is marked, then $c$ could not have been the closest center to $\ell_i$ (by Corollary 3.4).

It similarly holds that all nodes which are marked and have an unmarked child correspond to a leaf-to-center assignment. I.e., any such node $\eta$ is marked, implying there is a center in $T[\eta]$. $\eta$ also has an unmarked child. Consequently, there is a leaf $\ell \in T[\eta]$ whose LCA with the center-set is $\eta$.

Thus, the $k$-center cost is equal to the maximum leaf-to-center distance. As shown above, this in turn is equivalent to the maximum value among nodes which are marked and have an unmarked child. $\quad\square$

**Farthest-First Traversal.** Although the $k$-center task is NP-hard in the general metric setting, Fact A.2 allows us to solve it optimally (and almost trivially) in an LCA-tree using the farthest-first traversal algorithm [32, 28, 31].

We implement farthest-first traversal in an LCA-tree as follows. We assign the first center to a random leaf in the tree. For each subsequent center, we find the node whose value is equal to the cost of the solution (per Fact A.2). This node must have at least one unmarked child. We therefore choose one of these unmarked children at random and place the next center at a random leaf in it. We do this iteratively until we have placed $n$ centers. This is formalized in Algorithm 1. By Corollary 3.4, this algorithm will only place a center for the subtree at node $\eta$ if it has already placed a center for the subtree at parent$(\eta)$.

---

**Algorithm 1** `LCA-tree Farthest First Traversal`

---

**Input:** LCA-tree $T$

1: Place first center on random leaf and mark $T$ accordingly
2: **while** unmarked node exists **do**
3:     $\eta_i$ = highest value marked node with at least one unmarked child
4:     **while** $\eta_i$ has unmarked child $\eta_j$ **do**
5:         Place a center on random leaf in $T[\eta_j]$ and mark $T$ accordingly
6:     **end while**
7: **end while**
8: Return centers

---

**Lemma A.3.** *Let $T$ be an LCA-tree satisfying the conditions in Corollary 3.4 and let $k \in [n]$. Then the farthest-first traversal algorithm finds the optimal $k$-center solution in $T$.*

*Proof.* Let $k$ be any value in $[n-1]$ (if $k = n$, there is only one solution which is trivially optimal). Assume for contradiction that the 'greedy' $k$-center solution $\mathbf{C}_g$ which is obtained by farthest-first traversal is *not* optimal. Then there must be another clustering $\mathbf{C}_o$ of $k$ centers that is *actually* optimal, i.e. $\text{Cost}_\infty(T, \mathbf{C}_o) < \text{Cost}_\infty(T, \mathbf{C}_g)$. The two solutions must differ by at least one center placement and, consequently, the sets of nodes which these solutions mark must also differ. Let $\eta$ be the maximum value node which is marked in $\mathbf{C}_g$ but not in $\mathbf{C}_o$.

Since $\mathbf{C}_o$ did not mark $\eta$, we have $\text{Cost}_\infty(T, \mathbf{C}_o) \geq d(\text{parent}(\eta)) \geq d(\eta)$ by Fact A.2 and Corollary 3.4. However, $\mathbf{C}_g$ *has* marked $\eta$. Furthermore, for all nodes $\eta' \in T$ with $d(\eta') > d(\eta)$, Algorithm 1 has necessarily placed a center in $T[\eta']$ (otherwise it would not have reached the state where it placed a center on $T[\eta]$). Therefore, we must have $\text{Cost}_\infty(T, \mathbf{C}_g) \leq d(\eta)$. This gives the desired contradiction. $\square$

Interestingly, this correctness proof does not depend on *which* leaf gets chosen as the center in a subtree – just that any leaf is chosen. Also, the solutions induced by this clustering are necessarily hierarchical:

**Corollary A.4.** *Let $\{\mathbf{C}_1, \ldots, \mathbf{C}_n\}$ be the sets of $k$-center solutions obtained by Algorithm 1. Let the respective partitions be $\mathcal{H} = \{\mathcal{P}_1, \ldots, \mathcal{P}_n\}$ obtained by assigning all leaves in $T$ to their closest center in each solution. Then $\mathcal{H}$ satisfies Definition 4.1.*

*Proof.* For $k = 1$, all leaves belong to a single center. Now assume we have the farthest-first traversal solution $\mathbf{C}_k$ for some $1 < k \leq n - 1$ and are placing the next center $c_{k+1}$. This center will be placed in the subtree belonging to an unmarked node $\eta$ whose parent is marked. Before placing $c_{k+1}$, all of $T[\eta]$ belonged to a single center in $\mathbf{C}_k$ per the marking procedure. After placing $c_{k+1}$, all of $T[\eta]$ belongs to it. Thus, one cluster which was present in the $\mathbf{C}_k$ solution was split into two clusters. $\square$

Unfortunately, the naive farthest-first algorithm may take $O(n^2)$ time to obtain all clusterings for $k \in \{1, \ldots, n\}$ since, when placing center $c_i$, we may have to search through $O(n)$ nodes to find the one with the next-highest cost. However, we can improve this to $\texttt{Sort}(n)$ time – the time it takes to sort a list of the $O(n)$ values in the LCA-tree – by noting that the nodes' values grow as we go up the tree. Thus, it is sufficient to sort the internal nodes by these costs and then place their corresponding centers in that order. The following lemma formalizes this intuition:

**Lemma A.5.** *There exists an algorithm that performs farthest-first traversal in an LCA-tree in* $\mathtt{Sort}(n)$ *time.*

*Proof.* Before providing the algorithm, we first give an overview of the ideas which facilitate it. Assume we have placed $k - 1$ centers and are placing the $k$-th one. If the cost is induced by node $\eta$, then this means that at least one of $\eta$'s children is marked and at least one is unmarked. Let $\eta_m$ be the marked one and $\eta_u$ be the unmarked one. We must therefore place a center in $T[\eta_u]$ in order to shrink the cost induced by $d(\eta)$. We refer to the leaf where this center will be placed as $\eta$'s *corresponding center*.

Thus, our algorithm must separate the nodes in the tree into two components. *Either* a node gets marked when a center is placed in its parent's subtree *or* it will have a center placed in its subtree. This partitioning is accomplished by Algorithm 2: at a given node $\eta$, Algorithm 3 assigns $\eta$'s corresponding center as the corresponding center of its left-most child. For $\eta$'s remaining children, we store their value in a global dictionary. Algorithm 2 finally returns the dictionary of nodes in the tree and their values. This requires a single depth-first search and therefore Algorithm 2 runs in $O(n)$ time.

Given the output of Algorithm 2, we have a list of all the nodes whose corresponding centers must be placed. We now notice that, when placing the $k$-th center in accordance with the cost $d(\eta)$, we have *necessarily* placed the centers with respect to $\eta$'s parent (per Corollary 3.4 and Fact A.2). Thus, we can simply sort these nodes by their costs and place the corresponding centers in this order. This is done by Algorithm 3 and runs in $\mathtt{Sort}(n)$ time. $\qquad\square$

---

**Algorithm 2** CorrespondingCenters

**Input:** node $\eta$ in an LCA-tree; dict Costs mapping nodes to distances; dict $c$ mapping nodes to corresponding centers

1:  **if** $\eta$ is leaf **then**
2:      $c[\eta] = \eta$
3:      **Return**
4:  **end if**

5:  ChildCount $= 0$
6:  **for** $\eta' \in$ children$(\eta)$ **do**
7:      CorrespondingCenters$(\eta', \mathtt{Costs})$
8:      **if** ChildCount $= 0$ **then**
9:          $c[\eta] = c[\eta']$
10:     **else**
11:         $\mathtt{Costs}[\eta'] = d(\eta)$
12:     **end if**
13:     ChildCount $+= 1$
14: **end for**
15: **Return**

---

**Algorithm 3** Ultrametric-kCenter

**Input:** LCA-tree $T$

1:  Costs, $c$ = { }, { }
2:  CorrespondingCenters$(T.\mathrm{root}, \mathtt{Costs}, c)$ // assume pass-by-reference
3:  Costs = OrderedDict(Costs) // sorted from largest to smallest
4:  **for** $\eta \in$ Costs **do**
5:      Place center at $c[\eta]$
6:  **end for**

---

In this sense, the hierarchy of optimal $k$-center solutions is essentially isomorphic to the LCA-tree. I.e. in the $k$-center hierarchy, the costs are precisely equal to the values of the nodes and we place centers which correspond to nodes ordered by their value. Thus, given a binary LCA-tree, its $k$-center hierarchy is necessarily equivalent to it.

Together, Lemma A.3, Corollary A.4 and Lemma A.5 prove the following statement about $k$-center in LCA-trees.

**Theorem A.6.** *Let $T$ be an LCA-tree satisfying the conditions in Corollary 3.4 and $n$ be the number of leaves in $T$. Then there exists an algorithm which finds the optimal $k$-center solutions $\{\mathbf{C}_1, \ldots, \mathbf{C}_n\}$ for all $k \in \mathbb{N}_n$ in $\mathtt{Sort}(n)$ time. Furthermore, the respective partitions $\mathcal{H} = \{\mathcal{P}_1, \ldots, \mathcal{P}_n\}$ obtained by assigning all leaves in $T$ to their closest center satisfy Definition 4.1.*

We lastly note that this runtime is tight:

**Lemma A.7.** *Let $T$ be an LCA-tree satisfying the conditions in Corollary 3.4. Then there is a worst-case instance on which one cannot find all the optimal $k$-center solutions on $T$ for $k \in \{1, \ldots, n\}$ in faster than $\mathtt{Sort}(n)$ time.*

*Proof.* Consider a rooted tree that is complete and balanced: every leaf is at the same depth, and every internal node has two children. Let the leaves all be at depth $w$, so that there are $2^w$ leaves. Starting at depth $w$, assign the leaves' unique values from 1 to $2^w$. Then, for the nodes at depth $w - 1$, assign them unique values from $2^w + 1$ to $2^w + 2^{w-1}$. Continue this process until we reach the root node, to which we assign value $2^{w+1}$. As a result, we visit the tree's nodes one-by-one from the lowest level to the root and maintain a counter of the number of visited nodes. Each node is assigned the value of the counter when it is visited.

Labeling the nodes by these values necessarily gives us an LCA-tree: all values are non-negative, and values are non-decreasing along paths from the leaves to the root. Furthermore, all internal nodes at depth $i$ have distinct values. Suppose we are now performing $k$-center and have placed centers for all the nodes at depth $w - 1$ but have not yet for the nodes at depth $w$. There are, therefore, $O(n)$ available nodes on which to place centers. Of these, only one has the maximum leaf-to-center distance and therefore induces the cost.

Thus, we cannot place the remaining $O(n)$ centers faster than the time it takes to sort the remaining leaves' parents by their costs. □

## A.4 $(k, z)$-clustering in LCA-trees

The result for $k$-center essentially boils down to the speed and optimality of the classic 2-approximation when applied in an ultrametric. We now turn to the more interesting result that the optimal $(k, z)$-clustering solutions behave very similarly to the optimal $k$-center ones:

**Theorem A.8.** *Let $T$ be an LCA-tree satisfying the conditions in Corollary 3.4, $n$ be the number of leaves in $T$, and $z$ be any positive integer. Then there exists an algorithm which finds the optimal $(k, z)$-clustering solutions $\{\mathbf{C}_1, \ldots, \mathbf{C}_n\}$ for all $k \in \mathbb{N}_n$ in $\mathtt{Sort}(n)$ time. Furthermore, the respective partitions $\mathcal{H} = \{\mathcal{P}_1, \ldots, \mathcal{P}_n\}$ obtained by assigning all leaves in $T$ to their closest center satisfy Definition 4.1. are hierarchical.*

### A.4.1 Optimal $(k, z)$ Centers in Subtrees.

We start by considering what happens when we have a $(k - 1, z)$-clustering solution in an LCA-tree and place the $k$-th optimal center. Placing this center will create a trail of markings from the leaf up until the first marked node along the path to the root. The key insight which we will utilize is that this center is *necessarily* optimal everywhere along this trail:

**Lemma A.9.** *Let $c_z = OPT_{1,z}(T)$ be an optimal $(1, z)$-clustering solution for LCA-tree $T$. Then for every subtree $T' \subset T$ such that $c_z \in T'$, $c_z$ is an optimal $(1, z)$-clustering solution for $T'$.*

*Proof.* We show this inductively. The base case is the trivial LCA-tree of one node where, inherently, the only choice of the center is optimal, and there are no subtrees. For the inductive case, consider LCA-tree $T$ whose root has $k$ children, such that each child has an optimal center placed within it. Our goal is to show that if we were to have one center for all of $T$, the optimum would be one of the $k$ centers in its subtrees.

If we only had one center to place for all of $T$, that center must be in one of its subtrees. WLOG, let the optimal center for $T$ be in the subtree $T_1$. Thus, our cost for one center is necessarily of the form $\text{cost}_1(T) = \text{cost}_1(T_1) + \sum_{i=2}^{k} |T_i| \cdot d(\text{root}(T))^z$, where $|T_i|$ is the number of leaves in

the $i$-th subtree of $T$. By the inductive hypothesis, we already had an optimal center for subtree $T_1$, implying that $\text{cost}_1(T_1)$ is minimized by choosing the optimal center in $T_1$. The other term $\sum_{i=2}^{k} |T_i| \cdot d(\text{root}(T))^z$ does not depend on where in $T_1$ the center is placed. Therefore, the optimal center from $T_1$ remains optimal for $T$.

$\square$

The key thing to note about Lemma A.9 is that it must also apply to all internal nodes of an LCA-tree. I.e., suppose we place an optimal center with respect to the unmarked subtree $T' \subset T$. Since $T'$ is unmarked, Lemma A.9 holds for $T'$. Consequently, *every internal node in the LCA-tree has an optimal center (leaf) associated with it.* We give this its own definition:

**Definition A.10.** Let $T$ be an LCA-tree satisfying the conditions in Corollary 3.4, let $\eta$ be a node in $T$, and let $z$ be a positive integer. Then $\eta$'s *corresponding $z$-center* is $c_z(\eta) = OPT_{1,z}(T[\eta])$.

Lemma A.9 is the key property of $(k, z)$-clustering in ultrametric spaces that makes the entire proof go through. To illustrate its effectiveness, consider the Euclidean 1-means setting on a dataset of two clearly separated clusters. The Euclidean mean falls *between* the two clusters. In contrast, Lemma A.9 says that, in the ultrametric setting, the optimal 1-means solution is not only located within one of the two cluster, but that it is optimal for the cluster in which it is located. In practice, this means that after placing an optimal center, we can essentially forget about it – it is guaranteed to be optimal for any set of points it serves.

**Overview of Proof for Theorem A.8.** We now give a simple blueprint illustrating how we use this for fast, optimal $(k, z)$-clustering. Assume we have placed the first center and have left a set of nodes unmarked. For each such unmarked node, we can determine its optimal center as well as how much the subtree's $(k, z)$-clustering cost would decrease by placing this center. Curiously, an application of Lemma A.9 shows that these cost-decreases themselves satisfy the strong triangle inequality. Another application of Lemma A.9 then allows us to show that greedily choosing the maximum cost-decrease gives an optimal $(k, z)$-clustering solution in an LCA-tree. As a result, we can apply a variant of the $k$-center algorithm to the LCA-tree of cost decreases.

**Notation.** We require additional notation to simplify the presentation. Let us define the cost of a node as

$$\text{NodeCost}(\eta, z) = |T[\eta]| \cdot d(\text{parent}(\eta))^z,$$

where $|T[\eta]|$ is the number of leaves in the subtree rooted at $\eta$. This represents the cost contributed by $\eta$'s leaves when $\eta$ is unmarked, but its parent is marked. In essence, this is the cost of $T[\eta]$ in a $(k, z)$-clustering solution if there is no center in $T[\eta]$. Similarly, we define the cost *decrease* at $\eta$ as

$$\text{CostDecrease}(\eta, z) = \text{NodeCost}(\eta, z) - \text{Cost}(T[\eta], c_z(\eta)),$$

where $\text{Cost}(T[\eta], c_z(\eta)) = \sum_{\ell \in \text{leaves}(T[\eta])} d(\ell, c_z(\eta))^z$. Here, $c_z(\eta)$ is $\eta$'s corresponding z-center. We define the cost-decrease of the root node to be infinite.

In essence, the cost-decrease quantifies how much placing an optimal center in $T[\eta]$ would decrease the subtree's total cost. Importantly, the cost-decrease of a node $\eta$ assumes that $\text{parent}(\eta)$ – the node directly above $\eta$ – is marked. We will see in Lemma A.13 that this is a reasonable assumption: every useful center we will place in the $(k, z)$-clustering setting will always have a marked parent.

### A.4.2 Proving Theorem A.8.

First, we see that all of the corresponding $z$-centers can be found in $O(n)$ time on an LCA-tree:

**Lemma A.11.** *Let $T$ be an LCA-tree satisfying the conditions in Corollary 3.4. Then there exists an algorithm which, for all $\eta \in T$, finds $c_z(\eta)$ and $\text{Cost}(T[\eta], c_z(\eta))$ in $O(n)$ time. Furthermore, this algorithm stores the cost-decrease for all nodes $\eta'$ for which $c_z(\eta') \neq c_z(\text{parent}(\eta'))$.*

*Proof.* This is accomplished by Algorithm 4, which is essentially a depth-first implementation of Lemma A.9's proof.

---

**Algorithm 4** Corresponding-z-Centers

---

**Input:** node $\eta$ in an LCA-tree; dict Costs mapping nodes to their ; dict CostDecreases mapping nodes to their cost decrease; dict $c_z$ mapping nodes to their corresponding-z-centers

1: **if** $\eta$ is leaf **then**
2:      $c_z[\eta] = \eta$
3:      Costs$[\eta] = d(\eta)$
4:      **Return**
5: **end if**

6: **if** $\eta$ is root **then**
7:      ParentDist $= d(\eta) + 1$
8:      CostDecreases$[\eta] = \infty$
9: **else**
10:      ParentDist $= d(\text{parent}(\eta))$
11: **end if**

12: SumOfCosts $= \sum_{\eta' \in \text{children}(\eta)} |T[\eta']| \cdot d(\eta)$
13: CostIfChosen $= \{\eta'\text{: } 0 \text{ for } \eta' \in \text{children}(\eta)\}$
14: ChildCostDecreases $= \{\eta'\text{: } 0 \text{ for } \eta' \in \text{children}(\eta)\}$
15: **for** $\eta' \in \text{children}(\eta)$ **do**
16:      Corresponding-z-Centers($\eta'$, Costs, CostDecreases, $c_z$)
17:      CostIfChosen$[\eta'] = $ SumOfCosts $- |T[\eta']| \cdot d(\eta) + $ Costs$[\eta']$
18:      ChildCostDecreases$[\eta'] = |T[\eta]| \cdot$ParentDist - CostIfChosen$[\eta']$
19: **end for**

20: ChosenChild $= \arg\max(\text{ChildCostDecreases})$
21: $c_z[\eta] = c_z[\text{ChosenChild}]$
22: Costs$[\eta] = $ CostIfChosen[ChosenChild]
23: **for** $\eta' \in \text{children}(\eta)$ such that $\eta'$ is not ChosenChild **do**
24:      CostDecreases$[\eta'] = |T[\eta']| \cdot d(\eta)$
25: **end for**
26: **Return**

---

We prove by induction that Algorithm 4 finds the costs for all internal nodes. In the base case, our current node $\eta$ is a leaf. Thus, $\eta$'s corresponding $z$-center is $\eta$ and the cost of $\eta$ to this center is simply $d(\eta)$.

We now essentially reuse the logic from Lemma A.9 to prove the inductive step. We begin the inductive step with a node $\eta$ along with the costs and corresponding $z$-centers of each of $\eta$'s children. That is, for all $\eta' \in \text{children}(\eta)$, we have access to both $c_z(\eta)$ and $\text{Cost}(T[\eta'], c_z(\eta'))$. We now seek the optimal center for $\eta$ and what the cost would be in $T[\eta]$ after placing this center. By Lemma A.9, we know that the optimal center for $\eta$ is one of its children's corresponding $z$-centers. I.e., $c_z(\eta)$ must be equal to $c_z(\eta')$ for one of the children $\eta'$. We, therefore, test what the cost would be if we placed the center at each of the children and chose the minimum. By Lemma A.9, this center must be optimal. We therefore record the cost of placing this center in $\eta$, concluding the correctness proof.

Since this is done by depth-first search, the algorithm runs in $O(n)$ time. $\qquad\square$

Rather than thinking about corresponding $z$-centers as those which minimize the cost, we will instead think of them through the equivalent notion of the centers which *maximize* the cost-decrease. The next lemma shows the peculiar property that these cost-decreases themselves form a relaxed ultrametric:

**Lemma A.12.** *Let $T$ be an ultrametric LCA-tree which satisfies the conditions in Corollary 3.4 and let $T'$ be the LCA-tree obtained by replacing all values in $T$ by the cost-decreases. I.e., for all $\eta \in T'$, $d(\eta) = CostDecrease(\eta, z)$. Then, the LCA-distances over $T'$ also satisfy the conditions in Corollary 3.4.*

*Proof.* By Corollary 3.4, showing that $T'$ satisfies the strong triangle inequality simply requires verifying that the cost-decreases are non-negative and monotonically non-decreasing along any leaf-root path in $T'$. We first show that they are monotonically non-decreasing.

Let $\eta$ be an unmarked node with $h$ children whose parent is marked. We now place the optimal center $c_z$ in $T[\eta]$. WLOG, this center must land in one of $\eta$'s children's subtrees. Call this child $\eta_c$, implying that $c_z(\eta) = c_z(\eta_c)$. We now show that the cost-decrease of $\eta$ is greater than or equal to the cost decrease of $\eta_c$ by separating $\text{CostDecrease}(\eta, z)$ into a sum of terms, of which $\eta_c$ is a subset:

$$\text{CostDecrease}(\eta, z) = \text{NodeCost}(\eta, z) - \text{Cost}(T[\eta], c_z(\eta))$$

$$= \left( |T[\eta_c]| \cdot d(\text{parent}(\eta))^z + \sum_{\substack{\eta' \in \text{children}(\eta) \\ \eta' \neq \eta_c}} |T[\eta']| \cdot d(\text{parent}(\eta))^z \right)$$

$$- \text{Cost}(T[\eta], c_z(\eta))$$

$$= \left( |T[\eta_c]| \cdot d(\text{parent}(\eta))^z + \sum_{\substack{\eta' \in \text{children}(\eta) \\ \eta' \neq \eta_c}} |T[\eta']| \cdot d(\text{parent}(\eta))^z \right)$$

$$- \left( \text{Cost}(T[\eta_c], c_z(\eta)) + \sum_{\substack{\eta' \in \text{children}(\eta) \\ \eta' \neq \eta_c}} |T[\eta']| d(\eta)^z \right)$$

$$\geq \left( \text{NodeCost}(\eta_c, z) + \sum_{\substack{\eta' \in \text{children}(\eta) \\ \eta' \neq \eta_c}} |T[\eta']| \cdot d(\text{parent}(\eta))^z \right) -$$

$$\left( \text{Cost}(T[\eta_c], c_z(\eta)) + \sum_{\substack{\eta' \in \text{children}(\eta) \\ \eta' \neq \eta_c}} |T[\eta']| d(\eta)^z \right)$$

$$= (\text{NodeCost}(\eta_c, z) - \text{Cost}(T[\eta_c], c_z(\eta)))$$

$$+ \sum_{\substack{\eta' \in \text{children}(\eta) \\ \eta' \neq \eta_c}} |T[\eta']| \cdot (d(\text{parent}(\eta))^z - d(\eta)^z)$$

$$= \text{CostDecrease}(\eta_c, z) + \sum_{\substack{\eta' \in \text{children}(\eta) \\ \eta' \neq \eta_c}} |T[\eta']| \cdot (d(\text{parent}(\eta))^z - d(\eta)^z)$$

$$\geq \text{CostDecrease}(\eta_c, z),$$

where both inequalities are due to $d(\text{parent}(\eta)) \geq d(\eta)$. It must also be the case that $\text{CostDecrease}(\eta) \geq \text{CostDecrease}(\eta')$, where $\eta' \neq \eta_c$ is any other child of $\eta$. This is by transitivity, since $\text{CostDecrease}(\eta_c) \geq \text{CostDecrease}(\eta')$ by Lemma A.9. Thus, the cost-decreases are monotonically non-decreasing along paths from the leaves to the root.

It remains to be shown that the cost-decreases are necessarily non-negative. For this, we rely on the fact that the original distances in $T$ are non-negative. Since we already know that they are monotonically non-decreasing, we must only show that the cost-decrease of placing a center at a leaf is non-negative. To this end, let $\ell$ be any leaf. By Corollary 3.4. Then $\text{CostDecrease}(\ell, z) = \text{NodeCost}(\ell, z) - \text{Cost}(T[\ell], c_z(\ell))$. However, $\text{Cost}(T[\ell], c_z(\ell)) = d(\ell)$ while $\text{NodeCost}(\ell, z) \geq d(\text{parent}(\ell))$. Thus, $\text{CostDecrease}(\ell, z) \geq d(\text{parent}(\ell)) - d(\ell) \geq 0$.

$\square$

We note that the cost-decrease at a leaf $\ell$ is not necessarily 0, meaning that the LCA-tree of cost decreases is *not* an ultrametric (but still a relaxed ultrametric). This is why we required the definition of relaxed ultrametrics rather than standard ones.

The next lemma shows that not only do these cost-decreases satisfy the conditions in Corollary 3.4, but they are also monotonically *increasing* in all relevant settings. In other words, if $T[\eta_1]$ has non-zero cost, then placing a center there decreases this cost by a non-zero amount.

**Lemma A.13.** *Let $T$ be an ultrametric LCA-tree satisfying the conditions in Corollary 3.4 and let $z$ be a positive integer. For any leaf $\ell \in T$, let $p(\ell) = [\ell, \eta_i, \ldots, \eta_j, r(T)]$ be the path from $\ell$ to the root. Then for all $\eta_1, \eta_2 \in p(\ell)$ such that $d(\eta_1) > 0$, $\eta_1 \preceq \eta_2 \implies CostDecrease(\eta_1) < CostDecrease(\eta_2)$.*

*Proof.* By Lemma A.12, we know that the cost decreases are monotonically non-decreasing along paths to the root. Thus, it remains to show that the cost-decrease at a node $\eta_1$ is non-zero if $d(\eta_1) > 0$. To do this, we first decompose the cost-decrease at $\eta$:

$$\text{CostDecrease}(\eta_1) = \text{NodeCost}(\eta_1) - \text{Cost}(T[\eta_1], c_z(\eta_1))$$
$$= |T[\eta_1]| \cdot d(\text{parent}(\eta_1)) - \text{Cost}(T[\eta_1], c_z(\eta_1))$$
$$= d(\text{parent}(\eta_1)) \cdot \left( \sum_{\eta' \in \text{children}(\eta_1)} |T[\eta']| \right) - \text{Cost}(T[\eta_1], c_z(\eta_1)).$$

Now, let $\eta_c$ be the child of $\eta_1$ containing $c_z(\eta_1)$. Much as in the proof of Lemma A.12, we rewrite the above in terms of the costs in $\eta_c$ and the costs in the other children:

$$\text{CostDecrease}(\eta_1) = d(\text{parent}(\eta_1)) \cdot \left( |T[\eta_c]| + \sum_{\substack{\eta' \in \text{children}(\eta_1) \\ \eta' \neq \eta_c}} |T[\eta']| \right)$$

$$- \text{Cost}(T[\eta_c], c_z(\eta_1)) - d(\eta_1) \cdot \left( \sum_{\substack{\eta' \in \text{children}(\eta_1) \\ \eta' \neq \eta_c}} |T[\eta']| \right)$$

$$= \left( \sum_{\substack{\eta' \in \text{children}(\eta_1) \\ \eta' \neq \eta_c}} |T[\eta']| \right) \cdot (d(\text{parent}(\eta_1)) - d(\eta_1))$$
$$+ d(\text{parent}(\eta_1)) \cdot |T[\eta_c]| - \text{Cost}(T[\eta_c], c_z(\eta_1))$$
$$\geq d(\text{parent}(\eta_1)) \cdot |T[\eta_c]| - \text{Cost}(T[\eta_c], c_z(\eta_1))$$
$$\geq d(\text{parent}(\eta_1)) \cdot |T[\eta_c]|$$
$$\geq d(\eta_1) \cdot |T[\eta_c]|$$
$$> 0$$

where the first inequality is by the fact that $d(\eta_1) \leq d(\text{parent}(\eta_1))$, the second is due to the fact that the costs are strictly non-negative, the third is by the fact that $d(\text{parent}(\eta_1)) > d(\eta_1)$, and the fourth inequality is by the assumption that $d(\eta_1) > 0$.

$\square$

Lemma A.13 allows us to address the fact that the cost-decrease at a node $\eta$ was defined under the assumption that $\eta$'s parent is marked. By Lemma A.13, the cost decreases are either strictly increasing along paths to the root or are $0$. Thus, if two nodes have equivalent non-zero cost-decreases, their subtrees must be disjoint.[8] As a result, when we go through the nodes sorted by their cost-decreases, it will always be the case that the next node we place will either have its parent marked or will have a cost-decrease of $0$ (in which case it is irrelevant in terms of the optimal solutions).

We finally move to our last lemma, which shows that greedily maximizing cost-decreases results in an optimal $(k, z)$-clustering over the LCA-tree.

---

[8]Formally, for two nodes $\eta_1, \eta_2 \in T$ with $\text{CostDecrease}(\eta_1) > 0$ and $\text{CostDecrease}(\eta_2) > 0$, we have that $\text{CostDecrease}(\eta_1) = \text{CostDecrease}(\eta_2) \implies T[\eta_1] \cap T[\eta_2] = \emptyset$.

**Lemma A.14.** *The optimal $(k, z)$-clustering solution over an LCA-tree can be obtained by greedily choosing the $k$ centers that, at each step, maximize the cost-decrease.*

*Proof.* We show this by contradiction. Consider that there is a solution that was obtained greedily and another, different one that is actually optimal. We now map every center in the "optimal" solution to its closest center in the "greedy" one and observe how the optimal solution's cost changes. There are two cases that can occur: either all of the greedy centers receive one optimal center each, or there is at least one greedy center that receives more than one optimal center and another that receives none.

We start with the case where each greedy center $c_g$ has one optimal center $c_o$ mapped to it. By definition, $c_o$ must be in the set of points that are assigned to $c_g$. Thus, it is either (a) in $T[c_g]$ or (b) in another subtree whose parent was marked by $c_g$. In case (a), Lemma A.9 states that the greedy algorithm must have chosen $c_o$. In case (b), by the greedy algorithm, $T[c_g]$ has greater cost minimization than $T[c_o]$. Thus, we can decrease the cost of the optimal solution by replacing $c_o$ with $c_g$, giving a contradiction. Therefore, we conclude that if one optimal center was mapped to each greedy center, then the solutions must be equivalent.

It remains to consider the case where more than one optimal center was mapped to a greedy center $c_g$. WLOG, let there be two optimal centers $c_o^1$ and $c_o^2$ that are mapped to $c_g$. By a similar argument as above, $c_g$ must be the same as one of these optimal centers, i.e. $c_g = c_o^1$. By extension, $c_o^2 \neq c_g$. Now consider the greedy center elsewhere in the tree that had no optimal center mapped to it. Call this center $c_g'$. This means that the greedy algorithm had both $T[c_o^2]$ and $T[c_g']$ available to it but chose $T[c_g']$. Thus, $\text{CostDecrease}(T[c_o^2]) < \text{CostDecrease}(T[c_g'])$. We can therefore decrease the cost of the optimal solution by replacing center $c_o^2$ with center $c_g'$. This gives the desired contradiction.

$\square$

This brings us to the primary result of this chapter, restated from before:

**Theorem A.8.** *Let $T$ be an LCA-tree satisfying the conditions in Corollary 3.4, $n$ be the number of leaves in $T$, and $z$ be any positive integer. Then there exists an algorithm which finds the optimal $(k, z)$-clustering solutions $\{\mathbf{C}_1, \ldots, \mathbf{C}_n\}$ for all $k \in \mathbb{N}_n$ in $\texttt{Sort}(n)$ time. Furthermore, the respective partitions $\mathcal{H} = \{\mathcal{P}_1, \ldots, \mathcal{P}_n\}$ obtained by assigning all leaves in $T$ to their closest center satisfy Definition 4.1. are hierarchical.*

*Proof.* We use Algorithm 4 from Lemma A.11 to find the cost-decreases for $(k, z)$-clustering in the LCA-tree. By Lemma A.12, these satisfy the strong triangle inequality. Furthermore, by Lemma A.14, greedily choosing centers that maximize the cost-decreases gives an optimal $(k, z)$-clustering.

Thus, running farthest-first traversal on the cost-decrease LCA-tree will give the partitions for $(k, z)$-clustering solutions. This guarantees that the corresponding partitions are hierarchical.

However, we must be careful about where we place the centers: while running a naive farthest-first traversal on the cost-decrease LCA-tree will give the *partitions* of the optimal $(k, z)$-clustering solutions, it will not necessarily give the optimal *centers*. This is due to the fact that the farthest-first traversal is optimal regardless of which center we pick for every subtree. Luckily, we have already addressed this. When considering the LCA-tree of cost-decreases, we have a mapping between every cost-decrease and the center that induces it. Thus, we will perform the farthest-first traversal by placing the nodes' corresponding $z$-centers. This ensures that both the partition *and* the centers align with the optimal $(k, z)$-clustering solutions.

Putting this all together, our final algorithm – Algorithm 5 – is quite simple. We first run Algorithm 4 to return a list of nodes and their cost-decreases. Importantly, Algorithm 4 returns only the cost-decreases for those nodes $\eta'$ with $c_z(\eta') \neq c_z(\text{parent}(\eta'))$. We then sort this list in $\texttt{Sort}(n)$ time. By the discussion after Lemma A.13's proof, we can safely go through this list and place the nodes' corresponding $z$-centers: the nodes with non-zero cost-decrease will always have their parent marked. The nodes with zero cost-decrease come at the end of the sorted list and do not affect the optimality of the solution. By Lemma A.9, each corresponding $z$-center is immediately optimal in every node that it marks. Thus, Algorithm 5 optimally solves the $(k, z)$-clustering objective in LCA-trees which satisfy the conditions in Corollary 3.4. The bottleneck in this algorithm remains the time to sort the $O(n)$ internal values in the LCA-tree.

**Algorithm 5** `Ultrametric-kz`

---

**Input:** LCA-tree $T$

1: Costs, CostDecreases, $c_z$ = { }, { }, { }
2: Corresponding-z-Centers($T$.root, Costs, CostDecreases, $c_z$) // pass-by-reference
3: CostDecreases = OrderedDict(CostDecreases)
4: **for** $\eta \in$ CostDecreases **do**
5:      Place center at $c_z(\eta)$
6: **end for**

---

$\square$

Together, Theorems A.6 and A.8 prove Theorem 4.2 from the main body of the paper.

# B   Further Details for Section 5 - Choosing a Partition

## B.1   Ultrametric Elbow Method

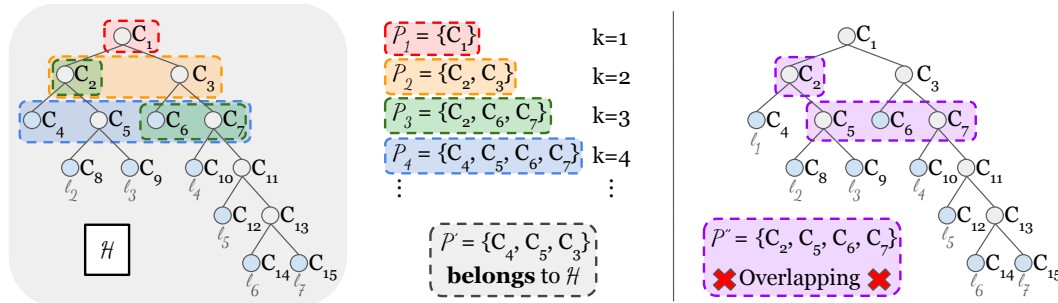

Figure 6: *Left*: an example hierarchy $\mathcal{H} = \{\mathcal{P}_1, \mathcal{P}_2, \ldots\}$; $\mathcal{P}' \notin \mathcal{H}$ is then an example partition which *belongs* to $\mathcal{H}$. *Right*: $\mathcal{P}''$ is *not* a partition since $\ell_2$ and $\ell_3$ each belong to clusters $C_2$ and $C_5$.

We begin by quickly showing Corollary 5.1, which holds as a direct consequence of Lemma A.13 in Section A.4:

**Corollary 5.1.** *Let $\mathcal{P}$ and $\mathcal{L}$ correspond to the $n$ partitions and losses obtained in accordance with Theorem 4.2 for the $(k, z)$-clustering objective. Let $\Delta_i = \mathcal{L}_{i+1} - \mathcal{L}_i$. Then either $\Delta_i < \Delta_{i+1} \leq 0$ or $\Delta_i = \Delta_{i+1} = 0$ for all $i \in [n-1]$.*

*Proof.* Lemma A.13 states that the cost-decreases associated with nodes in an LCA-tree monotonically increase along paths from any leaf to the root or are all 0. By Lemma A.14, we solve $(k, z)$-clustering using farthest-first traversal over the LCA-tree of cost-decreases. I.e., we start at the root and greedily pick the cluster that gives the maximal cost-decrease at that step. Consequently, each cost-decrease we pick must be smaller than the previous one.

Thus, let $\Delta'_k = \mathcal{L}_k - \mathcal{L}_{k+1} = -\Delta_k$. By Lemma A.13, we have $\Delta'_k > \Delta'_{k+1} \geq 0$ or $\Delta'_k = \Delta_{k+1} = 0$. Plugging in $\Delta = -\Delta'$ completes the proof. $\square$

**Choosing the elbow**   Although there are many methods for finding the elbow index, we are in the privileged setting where a single elbow exists and is clearly delineated. Inspired by Shi et al. [68], we simply define the elbow as the index where there most strongly appears to be a right angle. Namely, let $\vec{v}_i = (i, \mathcal{L}_i)$ be the $(x, y)$ position of the $i$-th point in the elbow plot. Then, for every index $k \in \{2, \ldots, n-1\}$, let $\theta_k$ be the angle induced by the vectors[9] $(\vec{v}_1 - \vec{v}_k)$ and $(\vec{v}_k - \vec{v}_n)$. We define the elbow as being at the index $k$ where $\theta_k$ is closest to 90 degrees. Since Theorem 4.2 gives us all of the partitions for $k \in \{1, \ldots, n\}$ simultaneously, this index can be found $O(n)$ time given the cluster hierarchy.

---

[9]In practice, we normalize the $k$ values and the costs to be in $[0, 1]$ so that the scales are comparable.

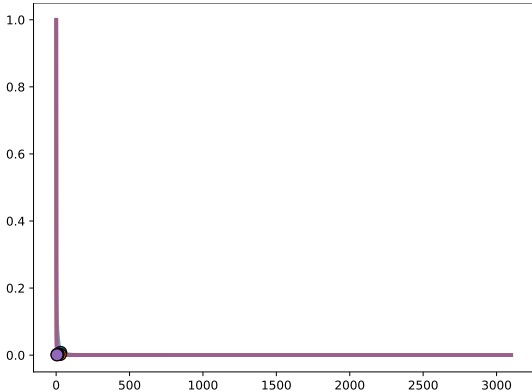

Figure 7: The same elbow plot as in Figure 3b with values of $k$ from 1 to $n$. The chosen elbows are higlighted with circles.

We note that Figure 3b only plots the elbow curves for values of $k$ up to 100. This is because plotting until $k = n$ makes the plot look in practice like a vertical line followed by a horizontal line, as seen in Figure 7. However, we use all values of $k$ from 1 to $n$ when choosing the elbow.

## B.2 Agglomerative Clustering Algorithms under our Framework

Corollary 3.4 and Theorem 4.2 imply that a large set of agglomerative clustering algorithms can be interpreted as $k$-center over various relaxed ultrametrics. As an example, consider the complete linkage algorithm [41]. Here, one starts with every point in its own cluster and recursively merges those clusters $C_i, C_j$ which have the smallest merge distance $\arg\min_{C_i,C_j} \max_{x_i \in C_i, x_j \in C_j} d(x_i, x_j)$. Importantly, with each subsequent merge, the merge distances are monotonically non-decreasing. Thus, by Corollary 3.4, labeling each cluster in the hierarchy by its merge distance gives us a relaxed ultrametric and, by Theorem 4.2, $k$-center over this ultrametric gives us the complete-linkage hierarchy. Indeed, this is true of *any* agglomerative clustering method where the merge distances progressively grow as we approach the root cluster.

## B.3 Thresholding

We quickly explain how one can partition a cluster hierarchy by thresholding. We assume that the cluster hierarchy $\mathcal{H}$ has every internal node labeled by its cluster's cost. As discussed in the main body of the paper, this constitutes a relaxed ultrametric.

Now let $\varepsilon$ be any threshold value. We can return the set of clusters that have cost less than $\varepsilon$ by depth-first search in $O(n)$ time. This is precisely what DBSCAN* does on the dc-dist relaxed ultrametric. Namely, DBSCAN* returns clusters of core points that are within $\varepsilon$ of each other under the dc-dist. Under the dc-dist's relaxed ultrametric definition, this is specifically the set of nodes with a value less than $\varepsilon$.

## B.4 Cluster Value Functions

The idea is a generalization of the excess-of-mass measure in HDBSCAN: we have a user-defined function $v(C)$ which assigns a non-negative value to each cluster in the hierarchy. I.e., $v(C) \geq 0$ for all clusters $C \in \mathcal{H}$. Then the *best* partition maximizes the sum of these values:

**Definition B.1.** Given a value function $v$, we define the **best** partition under $v$ as the partition that maximizes the sum of valuations. I.e. $\mathcal{P}_v(\mathcal{H}) = \arg\max_{\mathcal{P} \text{ belonging to } \mathcal{H}} \sum_{C \in \mathcal{P}} v(C)$.

This brings us to the following result:

**Theorem B.2.** *Let $\mathcal{H}$ be a cluster hierarchy. Let $v$ be a function such that, for all $C \in \mathcal{H}$, $v(C)$ can be obtained in $O(1)$ time. Then there exists an algorithm to find the best partition $\mathcal{P}_v(\mathcal{H})$ in $O(n)$ time.*

*Proof.* The algorithm for finding the best clustering is essentially Algorithm 3 from Campello et al. [12]. We provide our own version of it in Algorithm 6 for completeness' sake.[10] It works by depth-first search where, at each node, we simply calculate its value and compare it against the sum of the values of its children. Since calculating the value takes $O(1)$ time and there are $O(n)$ nodes in the cluster hierarchy, the algorithm therefore runs in $O(n)$ time.

We prove its correctness inductively. In the base case, we have a single leaf whose best clustering is simply the leaf itself. In the inductive case, we are given a cluster $C$ in the hierarchy and have the best clusterings of $C$'s children. We seek to find $B_v(C)$. By the non-overlapping requirement, if we include $C$ in $B_v(C)$, then we cannot include any of its children. Similarly, since the values of the children are non-negative, if we include one child, then we may as well include all of them. Thus, $B_v(C)$ is either $\{C\}$ or $\{C' : C' \in \text{children}(C)\}$.

---

**Algorithm 6** `BestClustering`

---

**Input:** node $C$ in an cluster hierarchy;
**Output:** the best clustering under this node $B_v(C)$, the value of the clustering $v(C)$;

1: **if** $C$ is leaf **then**
2:     $B_v(C) = \{C\}$
3:     **Return** $B_v(C)$
4: **end if**

5: ChildValues $= 0$
6: ChildClusterings $= \{\}$
7: **for** $C' \in \text{children}(C)$ **do**
8:     $B_v(C'), v(C') = $ `BestClustering`$(C')$
9:     ChildValues $\mathrel{+}= v(C')$
10:     ChildClusterings.append($B_v(C')$)
11: **end for**
12: **if** $v(C) > $ ChildValues **then**
13:     **Return** $\{C\}$, $v(C)$
14: **end if**
15: **Return** ChildClusterings, ChildValues

---

$\square$

**The *stability* cluster value function.** We now introduce the stability cluster value function from Campello et al. [12]. Let $C$ be any cluster in a hierarchy and let $C'$ be $C$'s parent in that hierarchy. Then the stability objective can be roughly phrased as the following:

$$v_E(C) = |C| \cdot \left( \frac{1}{\mathcal{L}(C)} - \frac{1}{\mathcal{L}(C')} \right) \tag{1}$$

We note that our description of the stability criterium differs slightly from the original function discussed in [12, 13, 53]. Namely, we omit here the notion that singleton points may fall out of the clustering as it is not easy to represent in our notation and the differences are negligible for the purposes of this discussion. Our implemented stability criterion is the original (correct) one. For a full discussion of the original function, we refer the reader to the referenced literature.

In either case, the stability value function can be interpreted as emphasizing those clusters that have a large number of points and have significantly lower costs than their parent. While we find that the stability criterion performs well in the $k$-center hierarchy, it can produce sub-par partitions when applied in the $(k, z)$-clustering hierarchies. This is because the per-point cost in the $(k, z)$-clustering task is comparable to the per-cluster cost in the $k$-center objective.

To be consistent with the literature [12], we utilize the stability cluster value function in the noisy setting. Given a user defined parameter $\mu$ representing the minimum cluster size, we first prune the tree so that all nodes with fewer than $\mu$ children are removed. We then run the stability value function over the remaining points. This selects a set of internal nodes to represent the clusters. Finally, we

---

[10]Note, we write $B_v(C)$ to refer to the best clustering of the cluster hierarchy rooted at $C$.

re-introduce the points which were pruned away. If the re-introduced points are in the sub-tree of a cluster, we assign them to the cluster. If, instead, they are not below a cluster found by the stability function, we label them as "noise".

## B.5 Further details on the dc-dist ultrametric

We begin by proving Proposition 6.3:

**Proposition 6.3.** *Let $(L, d')$ be a metric space. Then, the dc-dist over $L$ is a relaxed ultrametric.*

*Proof.* Note that it is known that the minimax distances over a space constitute an ultrametric [26]. Thus, we have that $d_{dc}(\ell_i, \ell_j) = d(\ell_j, \ell_i)$ and that $d_{dc}(\ell_i, \ell_j) \geq 0$ for all $\ell_i, \ell_j$.

To show that the dc-dist is a relaxed ultrametric, we will leverage that the only difference between it and the minimax distance is that the minimax distance of a point to itself is 0 while the dc-dist of a point to itself is its mutual reachability. Thus, we will prove that the dc-dist is a relaxed ultrametric by showing that the distance to itself is non-negative and that it inherits the strong-triangle inequality from the minimax ultrametric. Together, these imply the two properties for Corollary 3.4.

First, note that $d_{dc}(\ell_i, \ell_i) = \max(||\ell_i - \ell_i||, \kappa_\mu(\ell_i), \kappa_\mu(\ell_i)) = \max(0, \kappa_\mu(\ell_i)) = \kappa_\mu(\ell_i)$. Thus, the dc-dist of a point to itself is the distance of $\ell_i$ to its $\mu$-th nearest neighbor in the ambient metric. This is necessarily non-negative. Similarly, the mutual-reachabilities for any other pair of points are also necessarily non-negative. Because the dc-dist is the minimax distance over the pairwise mutual reachabilities and the pairwise mutual reachabilities are all non-negative, the dc-dist must also be.

Let us now show that the dc-dist satisfies the second property of Corollary 3.4. First, notice that the dc-dist of a point to itself is necessarily less than or equal to the dc-dists of that point to any other point in the set, i.e., $d_{dc}(\ell_i, \ell_i) \leq \max(d_{dc}(\ell_i, \ell_j), d_{dc}(\ell_j, \ell_i))$ for all $\ell_j$. To see this, consider the mutual reachability of $\ell_i$ to any other point $\ell_j$ is necessarily greater than $m_\mu(\ell_i, \ell_i)$. Namely, $m_\mu(\ell_i, \ell_j) = \max(d'(\ell_i, \ell_j), \kappa_\mu(\ell_i), \kappa_\mu(\ell_j)) \geq \kappa_\mu(\ell_i)$. As a result, any step in the mutual reachability MST that originates at $\ell_i$ will be at least as large as $m_\mu(\ell_i, \ell_i) = d_{dc}(\ell_i, \ell_i)$. Put simply, a point's closet point under the dc-dist is itself. Pairing this with the inheritance of the strong triangle inequality from the minimax distance gives us the second property of Corollary 3.4.

Thus, we have shown the two properties for Corollary 3.4: the dc-dists are non-negative and are non-decreasing as we traverse the DC tree from any leaf to the root. Consequently, it is a relaxed ultrametric. □

**Relationship to DBSCAN and HDBSCAN.** First, Campello et al. [12] showed that one can obtain DBSCAN* partitions using the single-linkage hierarchy over the mutual reachabilities. Rephrasing into this paper's notation, they showed that one can obtain DBSCAN* embeddings by thresholding the dc-dist's LCA-tree at a user-defined value $\varepsilon$; i.e., removing all nodes $\eta$ in the LCA-tree with $d(\eta) > \varepsilon$. Subsequently, Beer et al. [6] proved that $k$-center can be optimally solved over the dc-dist and that these partitions correspond to single-linkage over the mutual reachabilities. In this sense, one can think of $k$-center on the minimax distances as equivalent to single-linkage clustering. Lastly, Campello et al. [12] showed that, given the dc-dists's LCA-tree, one can choose a partition from it by optimizing the EoM cluster merging criterium over the pruned tree.

## C   Runtime Speedups / Efficient Operations

We now give an overview of how we implement the theory from Section A.3 and A.4 in practice in our codebase. We assume that we have already computed an ultrametric so that queries to the ultrametric require $O(1)$ time. From this, we describe how we build an LCA-tree, how to transform it to other clustering hierarchies, and how we extract partitions from it. We center this discussion on the dc-dist's dc-tree as a point of reference since we believe that this is the most common use case of our framework. However, we note that the discussion applies immediately to any relaxed ultrametric.

### C.1   Building and utilizing the hierarchy efficiently (theoretical complexities)

The overall structure of our implementation is the following:

1. **Building the hierarchy**
   (a) Compute annotations of cost-decreases over the dc-tree bottom up. These contain information that we will use for creating hierarchies.
   (b) Sort the annotations.
   (c) Create the hierarchy in a way that takes O(n) time utilizing the parent pointers in the annotations and in-place pointer updating.

2. **Get each solution in O(n)**
   (a) Annotate each node in the tree with the k that resulted in creating this node, given that the $k$-th center is chosen.
   (b) Top-down (or bottom-up) algorithm that cuts all edges in the tree going from $k$-annotation $> k$ to $k$-annotation $\leq k$. The result is all nodes above the cut.
   (c) Small edge case to deal with for nodes with $> 2$ children.

3. **Initialize smart pointer access** to the leaves in the tree so that internal nodes can get their leaves in constant time.

### C.1.1   Building the new LCA-tree of cost-decreases

How we build the tree in $O(n \cdot log(n))$.

Two main functions:

- Annotate tree
- Create hierarchy

**Annotating the tree**
'Annotate tree' creates the annotations for each internal node in the tree. Each annotation is the following: $A_k = [cost\_decrease, center, parent\_pointer, tree\_node]$.

**Idea C.1.** *Only maximal annotations for a given center will be picked with the greedy algorithm.*

By maximal annotations, we mean those highest in the tree corresponding to a specific center, i.e., the annotation for that center with the highest annotated cost-decreases. To see this, consider that only maximal annotations will be children of marked paths of other, already picked, centers in the tree.

**Idea C.2.** *The cluster corresponding to the parent center $p'$ of the maximal annotation of $p$ is exactly the cluster/set of points from which choosing $p$ will exclusively take points.*

This comes from the fact that each annotation is maximal in its corresponding subtree, which means that the maximal subtree above always will have chosen that center first. Also, any other center $p''$ chosen within that subtree of the maximal annotation of $p'$ will have taken disjoint points from that center $p'$'s cluster that $p$ cannot otherwise a contradiction and $p''$ should have been the parent annotation of $p$.

We formally define the parent center as the single center in solution $k - 1$ that contains all points of the $k$'th center, which will get assigned for $k$. We now get the following insights needed to make building the tree efficient:

1: We only need to store one annotation for each center, where we just store the highest / best annotation cost-decrease for that center. This can be updated as the annotations are computed.
2: Each annotation/center can contain a pointer to the parent center.

Having these, we now have all the information required to build the tree - for each new center, we have exactly the cost-decrease corresponding to picking it and the parent center. We can sort the list of annotations, and each k solution will correspond to the first k centers picked in that order.

**Building the tree from the annotations**
First, we sort the annotations. Picking the first annotation corresponds to the root node and cluster of all points with that center. Picking a subsequent new center will always correspond to a split in the tree, splitting up the parent center's cluster. One side of the split will be what is left over from the parent cluster, and the other side will be the new cluster for this $k$. The only caveat is that if there are multiple, some number $a$, annotations with the same cost-decrease and parent, we only get $a + 1$

nodes from this multi-split. We get a node for each of the $a$ new centers and the $a + 1$'st node for what is left over in the partition of the parent node's cluster.

Now, the key is that the cost-decreases are decreasing in the sorted order. This means that any subsequent annotations with a parent that already has been split up will always create a new split at the lowest possible node/cluster corresponding to that parent center, and all nodes above that also correspond to that parent center will never get new splits from them. This means that we can maintain for each center the current lowest/active node, which will be where any new center pointing to it should split from. This can easily be managed within the annotations with a pointer that is updated as the tree is created.

The steps are then generally the following at a given $k$'th annotation for some center $c_k$ in the traversal of the sorted list:

Case 1: If the parent node $p$ corresponding to some center $c_p$ has no offspring, add two new nodes below. One for the new annotation and $p'$ for the old center representing that nodes have been taken from it. For the new node for $c_k$, update the annotation to point to it.

Case 2: If the parent $p$ already has offspring associated with a higher cost-decrease, then add the new split nodes below $p'$ that then have to exist as that other offspring will have created that in case 1. Update the parent center's annotation to now point to $p'$ instead of $p$, as we can now be sure everything pointing to that center should never be added to $p$ but instead to $p'$ or further below at a later point. For the new node for $c_k$, update the annotation to point to it.

Case 3: The other offspring of the parent node has the same cost-decrease associated with the current annotation. This means the $p'$ has already been created, so just add a singular new node as a child of $p$ corresponding to center $c_k$. For the new node for $c_k$, update the annotation to point to it.

**Complexity** Computing the annotations is a single-pass bottom-up algorithm over the tree that does constant work in each tree node. As there are $O(n)$ nodes in the tree, this step has a worst-case complexity of $O(n)$.

Sorting the list of annotations has a worst-case complexity of $O(n \cdot log(n))$ as there are $n$ annotations to sort.

Computing the tree from the sorted annotations does a single scan over the list of annotations, with constant work in any iteration and, therefore, a worst-case complexity of $O(n)$.

Therefore, the total worst-case complexity is $O(n \cdot log(n))$.

### C.1.2   Optimizing the tree: Resolving ties better

An important observation is that due to the number of equidistant points under the relaxed ultrametric, many centers will often have tied distances to points between them. For example, consider that we have placed our first center $c_1$ and it marks every node along the path to the root. Now let there be two subtrees $T_i$ and $T_j$ which are *not* marked but whose parent node *is* marked. We now place a center $c_2$ in $T_i$. This creates a new path of markings that stops at the root of $T_i$.

Now notice that *every* node in $T_j$ is equidistant to center $c_1$ and to center $c_2$. Thus, we can leave $T_j$ assigned to $c_1$, or we can re-assign it to $c_2$, and the cost remains *unchanged*. Over the course of placing $k$ clusters, there will inevitably be many such ties, implying that there are an exponential number of optimal solutions!

The previous theory described this setting by simply utilizing a "first come, first served" principle and never re-assigning points to new clusters. However, under the dc-dist and other minimax path distances, this may result in unintuitive clusterings. To see this, consider a set of 10 copies of the same cluster which are equally spaced apart. Let there be two centers assigned to these: $c_1$ is placed in the first copy of the cluster, and $c_2$ is placed in the second copy of the cluster. The remaining 8 clusters must now be assigned to either $c_1$ or $c_2$. However, due to them being equally spaced apart, we can assign each of these 8 clusters to either center and the cost will remain the same.

Our recommendation on how to resolve this is to introduce a secondary heuristic for tie-breaking. Namely, if a subtree is equidistant to two centers under the ultrametric, assign it to the center which it is closest to in the data's original metric. I.e., if our ambient distance metric is Euclidean, then we use

the Euclidean distance as a tie-breaker. This provides the intuitive clustering that one expects when looking at groups of points.

In practice, we achieve this by giving each internal node in the tree a representative point - the Euclidean mean of its leaves. Each annotation not yet marked (and thus chosen at a later k) maintains the closest center based on the Euclidean distance to its representative, which is resolved between any center that marks its parent. At the k where this annotation itself is chosen, we have found the best annotation, and thus, the internal node that this annotation and sub-tree of nodes itself should become a subtree of. We update all annotation pointers in this way by placing the centers one by one, following the path from the center to the leaf checking distances to unmarked sub-trees, and updating the pointer if the new center is closer than other tied centers that had already been marked.

As this is just a processing step over the annotations before the tree is actually constructed, it can easily be plugged in/out based on a boolean flag. Furthermore, the choice of heuristic can also easily be changed, but here it should be kept in mind that it might influence the running time if any complex function is used.

The choice of using the mean point as a representative and comparing Euclidean distances is based on the following: under the dc-dist, one often gets one cluster which consists of multiple equidistant sub-clusters. These equidistant sub-clusters often follow snaky paths throughout the ambient space. Let us now suppose our large cluster gets split into two, implying that its sub-clusters must get assigned to one of these two new groups. Since the sub-clusters are all equidistant, they can be arbitrarily assigned to either of the two groups. By breaking ties so that the sub-clusters are closest to each other in the ambient space, we dramatically increase the chance of the sub-clusters getting re-assigned in an intuitive manner.

**Complexity**
This process requires traversing the tree from the leaf to the root for all centers placed. To construct the full hierarchy/dendrogram/tree of solutions, we do constant time calculations (scales with dimensionality) at each visit of an internal node, as it only requires computing the Euclidean distance of two points. This means that the worst-case complexity becomes $O(n^2)$ in the case of a fully unbalanced tree, but in practice and expectation, this will be much closer to $O(n \cdot log(n))$.

### C.1.3 Find solution for a given $k$

To find the solution for a given $k$ efficiently, we start out by annotating each node of the new tree with additional information. Generally, the insight is that going from $k$ to $k + 1$ corresponds to splitting up a single node/cluster into two new nodes in the tree. A slight detail is that some nodes will have multiple children, where the "split" is implicit as all the children split from the same node. We can mark at which $k$ each node is split from the parent in constructing the tree, where $k$ is just the current iteration. We simply mark all new nodes created in an annotation with that $k$.

To then recover a solution for a given $k$, all that is required is to cut all edges in the tree going from $k' \geq k$ to $k' < k$. The solution is the clusters associated with all the nodes above the cut. The only note is that if a leaf is reached and no cut is found in that path, the leaf is just part of the solution.

The correctness of this algorithm comes from the fact that any clusters below this cut only existed at a higher $k''$ than the solution for $k$ we are recovering.

**Complexity**
Marking the tree is done with a constant factor added to the complexity of the list traversal of building the tree, so $O(n)$.

Finding the cut can also be done in a single traversal of the tree top-down, simply stopping the traversal of a given branch when a cut is found and returning the nodes. So this is also $O(n)$.

Furthermore, the solution for a given $k$ can be stored, and the algorithm to find another $k$ can be "resumed" from this solution, making finding similar $k$ values very fast, almost constant.

### C.1.4 Find leaves for a given node quickly / labeling a solution

We create an array of the id's in the leaves, where each id is inserted in postfix order. This means that looking at the tree, each internal node of the tree will correspond to a continuous area of that array. If we store the bounds of those segments in each internal node, the nodes corresponding to a node can

be returned in constant time by just returning that continuous segment of data of the new array. Only a single postfix order traversal of the tree is required to set up this array and pointers in the internal node, which then takes $O(n)$.

However, if we want to recover the labels in the standard form of each label being in the order that the points were provided, a traversal of the recovered solution is required, putting the corresponding label in the right place in an output array of labels. Simply put, the label value of point $p$ has to be inserted in place $p$ of the output array of labels. This requires worst-case $O(n)$ complexity for a given $k$ solution, as it is just a linear scan with constant work for each of the $n$ points.

## D  The Algorithms

We now describe how these algorithms are implemented. We use the terminology $n$-ary dc-tree to refer to a non-binary LCA-tree storing dc-dist relationships. Again, this immediately transfers to all relaxed ultrametrics.

Algorithm 7 describes how we find the corresponding $z$-centers and the costs associated with them in practice. Algorithm 8 describes how we use this information to obtain the corresponding $(k, z)$-clustering hierarchy. Algorithm 9 shows how we extract clusterings for a specific value of $k$ from a hierarchy. Lastly, Algorithm 10 describes how we implement the tie-breaking heuristic assuming Euclidean distance.

---

**Algorithm 7** $k$-centroid-annotation

---

**Input:** An $n$-ary dc-tree $T$ over the dataset $\mathbf{X}$, power $z$, array $A$

 1: **if** $|T| = 1$ **then**
 2:      $A[T.\text{id}] = \{(T.\text{parent.dist})^z, T.\text{id}, nullPtr\}$
 3:      **return** $0, T.\text{id}$
 4: **else**
 5:      $bestCost = \infty, bestCenter = -1$
 6:      **for** $C \in T.children$ **do**
 7:          $subCost, subCenter = k\text{-centroid-annotation}\,(C, z)$
 8:          $currCost = subCost + (T.\text{dist})^z \cdot (T.\text{size} - C.\text{size})$
 9:          **if** $currCost < bestCost$ **then**
10:             $bestCost = currCost, bestCenter = currCenter$
11:          **end if**
12:      **end for**
13:      **for** $C \in T.\text{children}$ **do**
14:          $A[C.\text{id}].\text{parent} = bestCenter$
15:      **end for**
16:      $costDecrease = (T.\text{parent.dist}^z \cdot T.\text{size} - bestCost$
17:      $A[T.\text{id}] = \{costDecrease, bestCenter, nullPtr\}$
18:      **return** $bestCost, bestCenter)$
19: **end if**

---

**Algorithm 8** $k$-centroid-hierarchy

**Input:** An $n$-ary dc-tree $T$ over the dataset $\mathbf{X}$, power $z$

1: $A = k$-centroid-annotation $(T, z, [T.size])$
2: Sort($A$)
3: Root = Node(parent $= nullPtr$, cost $= -1$, id $= A[0]$.center)
4: $A[0]$.tree = Root
5: **for** $a_{cur} \in A[1]...A[n-1]$ **do**
6:      $N_{new} = $ Node(parent $= nullPtr$, cost $= -1$, id $= a_{cur}$.center)
7:      $a_{cur}$.tree $= N_{new}$
8:      $a_{par} = A[a_{cur}$.parent$], c_p = a_{par}$.tree.cost
9:      **if** $c_p \neq a_{cur}$.cost $\wedge c_p \geq 0$ **then**          $\triangleright$ Parent has added children with higher cost
10:          $N_{par} = a_{par}$.tree.children$[0]$
11:          $N_{par}$.cost $= a_{cur}$.costDecrease
12:          $N_{new}$.parent $= N_{par}$
13:          $N'_{par} = $ Node(parent $= N_{par}$, cost $= -1$, id $= N_{par}$.id)
14:          $N_{par}$.children.add($N'_{par}$)          $\triangleright$ Add same center node as first child
15:          $N_{par}$.children.$add(N_{new})$
16:          $a_{par}$.tree $= N_{par}$          $\triangleright$ Update annotation to point to lowest corresponding node
17:      **else if** $c_p < 0$ **then**          $\triangleright$ Parent has no children added
18:          $N_{new}$.parent $= a_{par}$.tree
19:          $N'_{par} = $ Node(parent $= a_{par}$.tree, cost $= -1$, id $= a_{par}$.id)
20:          $a_{par}$.tree.children.add($N'_{par}$)
21:          $a_{par}$.tree.children.add($N_{new}$)
22:      **else**          $\triangleright$ Parent has children added with same cost
23:          $a_{par}$.tree.children.add($N_{new}$)
24:          $N_{new}$.parent $= a_{par}$.tree
25:      **end if**
26: **end for**
27: **return** Root

---
**Algorithm 9** $k$-centroid-cluster
---
**Input:** An annotated $k$-centroid hierarchy-tree $T$ over the dataset $\mathbf{X}$, $searchK$
---
 1: **if** $|T| = 1$ **then**
 2:     **Output** $T$
 3: **end if**
 4: $maxK = \text{maxK}(T.\text{children}), minK = \text{minK}(T.\text{children})$
 5: **if** $maxK \leq searchK$ **then**                 ▷ Every edge should be cut
 6:     **for** $C \in T.\text{children}$ **do**
 7:         $k$-centroid-cluster$(C, searchK)$
 8:     **end for**
 9: **else**                              ▷ Only some edges should be cut
10:     $A = []$
11:     **for** $C \in T.\text{children}$ **do**
12:         **if** $C.k > searchK \vee C.\text{is\_orig\_cluster}$ **then**      ▷ Non-cut edges merge with original
13:             $A.\text{add}(C)$
14:         **else**
15:             **Output** $C$
16:         **end if**
17:     **end for**
18:     **Output** Merge$(A)$
19: **end if**
---

---
**Algorithm 10** Optimize Annotations
---
**Input:** Sorted list of annotations in decreasing order $A$, tree $T$ annotated with representatives
---
 1: **for** $a_{cur} \in A$ **do**                       ▷ For each annotation
 2:     $N = a_{cur}.leaf$
 3:     **while** $N \neq null$ **do**              ▷ Traverse from center leaf to root
 4:         $N.mark(a_{cur}.center)$
 5:         **for** $C \in N.children$ **do**
 6:             **if** $C.mark = null$ **then**      ▷ Any unmarked children of marked path
 7:                 $C.anno.bestParent = min\{C.anno.bestParent, Euclid(C.rep, Anno.center)\}$
 8:             **end if**
 9:         **end for**
10:         $N = N.parent$
11:     **end while**
12: **end for**
13: **Return** $A$                    ▷ Return list of annotations with updated pointers
---

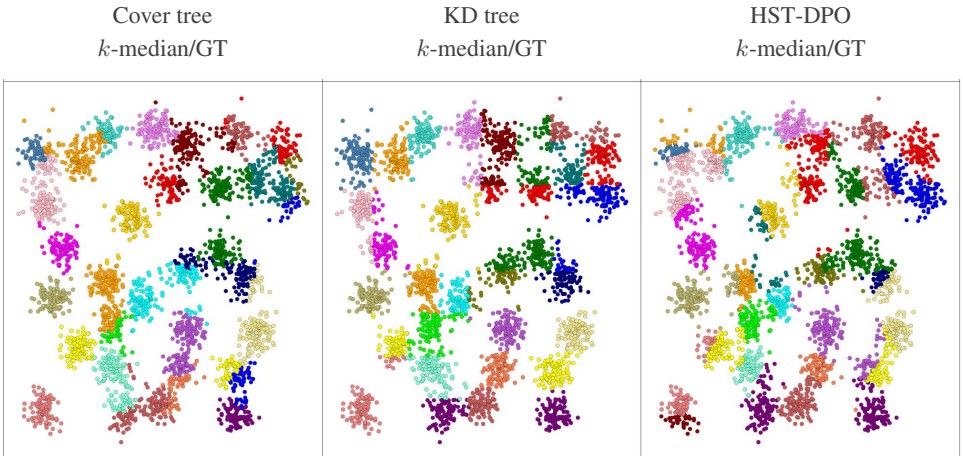

(a) Comparison of HST ultrametrics using the $k$-median hierarchy and the ground-truth value of $k$.

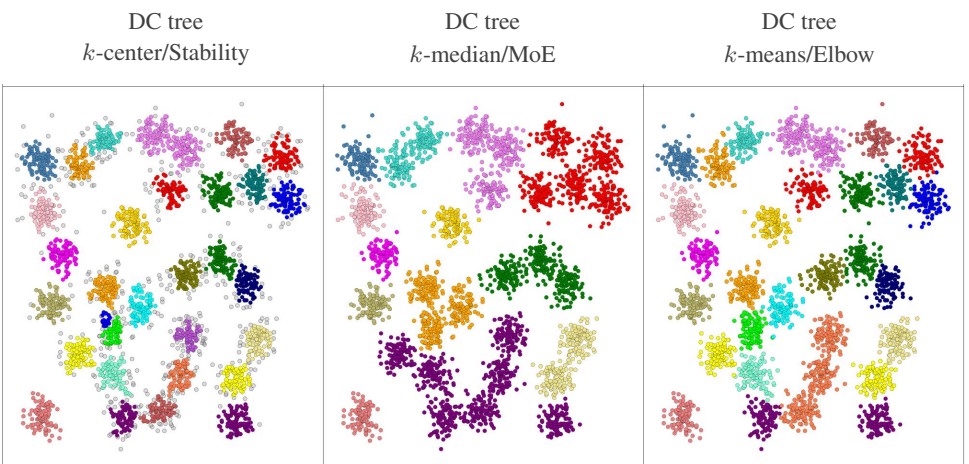

(b) Comparison of hierarchy/partition combinations on the dc-dist ultrametric.

Figure 8: A visualization of ultrametrics and hierarchy/partition combinations on the D31 dataset.

## E    Implementations of the used HSTs

We build KD trees, Cover trees using the C++ library *mlpack4* [20], and build HST-DPO trees using the C++ code from `https://github.com/yzengal/ICDE21-HST`.

## F    Used Datasets

Table 4 lists the datasets on which we validated and compared the proposed framework SHiP to other competitors.

## G    Tables – Runtimes and performance metrics

Table 5 shows the runtimes, Table 6 the ARI values, Table 7 shows the NMI values, Table 8 shows the AMI and Table 9 shows the correlation coefficients cores for all datasets.

Table 4: Dataset properties. Number of samples ($n$), dimensions ($d$), number of ground truth clusters ($k$), number of noise points (#noise), and the source.

| | | Dataset | $n$ | $d$ | $k$ | #noise | Source |
|---|---|---|---|---|---|---|---|
| **Density-based 2D-Data** | | Boxes | 21,600 | 2 | 12 | 0 | [78] |
| | Tomas Barton Benchmark | D31 | 3100 | 2 | 31 | 0 | [5] |
| | | 3-spiral | 312 | 2 | 3 | 0 | [5] |
| | | aggregation | 788 | 2 | 7 | 0 | [5] |
| | | chainlink | 1,000 | 3 | 2 | 0 | [5] |
| | | cluto-t4-8k | 8,000 | 2 | 6 | 764 | [5] |
| | | cluto-t5-8k | 8,000 | 2 | 7 | 1,153 | [5] |
| | | cluto-t7-10k | 10,000 | 2 | 9 | 792 | [5] |
| | | cluto-t8-8k | 8,000 | 2 | 8 | 323 | [5] |
| | | complex8 | 2,551 | 2 | 8 | 0 | [5] |
| | | complex9 | 3,031 | 2 | 9 | 0 | [5] |
| | | compound | 399 | 2 | 6 | 0 | [5] |
| | | dartboard1 | 1,000 | 2 | 4 | 0 | [5] |
| | | diamond9 | 3,000 | 2 | 9 | 0 | [5] |
| | | jain | 3,373 | 2 | 2 | 0 | [5] |
| | | pathbased | 299 | 2 | 3 | 0 | [5] |
| | | smile1 | 1,000 | 2 | 4 | 0 | [5] |
| | | Synth_low | 5,000 | 100 | 10 | 500 | [40] |
| | | Synth_high | 5,000 | 100 | 10 | 500 | [40] |
| **Real-World Data** | Tabular Data | Mice | 1,077 | 68 | 8 | 0 | [50] |
| | | adipose | 14,947 | 2 | 12 | 0 | [11] |
| | | airway | 14,163 | 2 | 10 | 0 | [11] |
| | | lactate | 39,825 | 2 | 6 | 0 | [11] |
| | | HAR | 10,299 | 561 | 6 | 0 | [50] |
| | | letterrec. | 20,000 | 16 | 26 | 0 | [50] |
| | | Pendigits | 10,992 | 16 | 10 | 0 | [50] |
| | Image Data | COIL20 | 1,440 | 16,384 | 20 | 0 | [60] |
| | | COIL100 | 7,200 | 49,152 | 100 | 0 | [59] |
| | | cmu_faces | 624 | 960 | 20 | 0 | [50] |
| | | Optdigits | 5,620 | 64 | 10 | 0 | [50] |
| | | USPS | 9,298 | 256 | 10 | 0 | [37] |
| | | MNIST | 70,000 | 784 | 10 | 0 | [46] |

Table 5: Runtimes of various parts of our clustering framework SHiP over ten runs, compared with standard clustering methods. Time is given in minutes, seconds, and milliseconds [min:sec.ms]. The first four columns ("build") refer to the computation time of the different ultrametric trees. The next two columns state the runtimes of computing one hierarchy ($k$-means) on the DC tree. The second column here shows the timing for the heuristic from Section C.1.2. Note that these runtimes are not correlated to the chosen ultrametric. The subsequent three columns state the runtimes of different partitioning methods, i.e., 'Stability', 'Median of Elbows (MoE)', and the 'Elbow' method on various hierarchies.

| | Dataset | build ultrametric | | | | build hierarchy | | partitioning | | | $k$-means ($k$ = GT) | $k$-means ($k$ = 500) | competitors | | | |
| | | KD tree | Cover tree | HST-DPO | DC tree | $k$-means | $k$-means (heur.) | Stability | MoE | Elbow | | | SCAR | Ward | AMD-DBSCAN | DPC |
|---|---|---|---|---|---|---|---|---|---|---|---|---|---|---|---|---|
| Density-based 2D-Data | Boxes | 00:00.013 | 00:00.059 | 00:09.679 | 00:14.239 | 00:00.033 | 00:00.498 | 00:00.005 | 00:00.134 | 00:00.002 | 00:00.114 | 00:01.217 | 00:01.481 | 00:10.808 | 01:04.670 | 12:10.358 |
| | D31 | 00:00.002 | 00:00.004 | 00:00.207 | 00:00.327 | 00:00.004 | 00:00.020 | 00:00.000 | 00:00.015 | 00:00.000 | 00:00.289 | 00:00.509 | 00:00.620 | 00:00.153 | 00:00.878 | 00:12.816 |
| | 3-spiral | 00:00.000 | 00:00.000 | 00:00.002 | 00:00.009 | 00:00.000 | 00:00.001 | 00:00.000 | 00:00.000 | 00:00.000 | 00:00.095 | 00:00.131 | 00:00.023 | 00:00.002 | 00:00.025 | 00:00.091 |
| | aggregation | 00:00.000 | 00:00.000 | 00:00.024 | 00:00.035 | 00:00.001 | 00:00.004 | 00:00.000 | 00:00.004 | 00:00.000 | 00:00.227 | 00:00.204 | 00:00.038 | 00:00.010 | 00:00.121 | 00:00.663 |
| | chainlink | 00:00.000 | 00:00.001 | 00:00.042 | 00:00.045 | 00:00.001 | 00:00.008 | 00:00.000 | 00:00.004 | 00:00.000 | 00:00.133 | 00:00.171 | 00:00.049 | 00:00.012 | 00:00.159 | 00:01.128 |
| | cluto-t4-8k | 00:00.004 | 00:00.017 | 00:01.359 | 00:01.898 | 00:00.012 | 00:00.164 | 00:00.001 | 00:00.046 | 00:00.000 | 00:01.344 | 00:01.514 | 00:00.417 | 00:01.233 | 00:06.470 | 01:32.488 |
| | cluto-t5-8k | 00:00.005 | 00:00.013 | 00:01.319 | 00:01.836 | 00:00.010 | 00:00.198 | 00:00.001 | 00:00.042 | 00:00.000 | 00:00.682 | 00:01.758 | 00:00.378 | 00:01.256 | 00:05.942 | 01:33.225 |
| | cluto-t7-10k | 00:00.006 | 00:00.019 | 00:02.100 | 00:02.848 | 00:00.014 | 00:00.252 | 00:00.002 | 00:00.058 | 00:00.000 | 00:02.532 | 00:02.052 | 00:00.455 | 00:02.058 | 00:10.670 | 02:39.063 |
| | cluto-t8-8k | 00:00.005 | 00:00.014 | 00:01.328 | 00:01.728 | 00:00.012 | 00:00.011 | 00:00.001 | 00:00.046 | 00:00.000 | 00:01.930 | 00:01.899 | 00:00.327 | 00:01.394 | 00:09.212 | 01:39.524 |
| | complex8 | 00:00.001 | 00:00.003 | 00:00.129 | 00:00.244 | 00:00.003 | 00:00.011 | 00:00.000 | 00:00.012 | 00:00.000 | 00:00.565 | 00:00.592 | 00:00.221 | 00:00.120 | 00:01.295 | 00:10.900 |
| | complex9 | 00:00.001 | 00:00.004 | 00:00.185 | 00:00.304 | 00:00.003 | 00:00.014 | 00:00.000 | 00:00.012 | 00:00.000 | 00:00.745 | 00:00.682 | 00:00.175 | 00:00.158 | 00:01.376 | 00:12.483 |
| | compound | 00:00.000 | 00:00.000 | 00:00.004 | 00:00.011 | 00:00.000 | 00:00.001 | 00:00.000 | 00:00.000 | 00:00.000 | 00:00.066 | 00:00.127 | 00:00.036 | 00:00.003 | 00:00.031 | 00:00.204 |
| | dartboard1 | 00:00.000 | 00:00.002 | 00:00.027 | 00:00.051 | 00:00.001 | 00:00.003 | 00:00.000 | 00:00.004 | 00:00.000 | 00:00.210 | 00:00.270 | 00:00.033 | 00:00.016 | 00:00.138 | 00:01.346 |
| | diamond9 | 00:00.001 | 00:00.004 | 00:00.183 | 00:00.310 | 00:00.003 | 00:00.017 | 00:00.000 | 00:00.012 | 00:00.000 | 00:00.321 | 00:00.751 | 00:00.194 | 00:00.153 | 00:01.107 | 00:12.540 |
| | jain | 00:00.000 | 00:00.000 | 00:00.004 | 00:00.006 | 00:00.000 | 00:00.001 | 00:00.000 | 00:00.000 | 00:00.000 | 00:00.005 | 00:00.045 | 00:00.024 | 00:00.002 | 00:00.028 | 00:00.101 |
| | pathbased | 00:00.000 | 00:00.000 | 00:00.002 | 00:00.006 | 00:00.000 | 00:00.001 | 00:00.000 | 00:00.000 | 00:00.000 | 00:00.013 | 00:00.039 | 00:00.023 | 00:00.001 | 00:00.024 | 00:00.061 |
| | smile1 | 00:00.000 | 00:00.002 | 00:00.024 | 00:00.036 | 00:00.001 | 00:00.005 | 00:00.000 | 00:00.004 | 00:00.000 | 00:00.321 | 00:00.319 | 00:00.048 | 00:00.017 | 00:00.139 | 00:01.392 |
| Tabular Data | Synth_low | 00:00.040 | 00:00.087 | 00:04.264 | 00:01.090 | 00:00.007 | 00:00.128 | 00:00.001 | 00:00.027 | 00:00.000 | 00:00.351 | 00:03.754 | 00:00.367 | 00:00.999 | 00:02.879 | 00:33.945 |
| | Synth_high | 00:00.034 | 00:00.030 | 00:04.075 | 00:00.878 | 00:00.006 | 00:00.117 | 00:00.001 | 00:00.024 | 00:00.000 | 00:00.782 | 00:04.199 | 00:00.455 | 00:00.982 | 00:04.573 | 00:33.785 |
| | Mice | 00:00.003 | 00:00.006 | 00:00.126 | 00:00.052 | 00:00.001 | 00:00.013 | 00:00.000 | 00:00.004 | 00:00.000 | 00:00.173 | 00:00.876 | 00:00.054 | 00:00.073 | 00:00.482 | 00:01.423 |
| | airway | 00:00.008 | 00:00.027 | 00:04.054 | 00:04.997 | 00:00.021 | 00:00.592 | 00:00.003 | 00:00.085 | 00:00.001 | 00:00.122 | 00:00.392 | 00:00.651 | 00:05.243 | 00:16.822 | 06:46.726 |
| | lactate | 00:00.025 | 00:00.161 | 00:39.433 | 00:47.731 | 00:00.068 | 00:07.184 | 00:00.012 | 00:00.275 | 00:00.004 | 00:00.126 | 00:01.997 | 00:02.612 | 00:59.992 | 07:43.293 | 69:34.012 |
| | HAR | 00:00.341 | 00:11.053 | 01:55.260 | 00:23.144 | 00:00.014 | 00:01.936 | 00:00.002 | 00:00.056 | 00:00.000 | 00:02.248 | 00:08.478 | 00:00.658 | 00:18.448 | 00:30.129 | 03:45.745 |
| | letterrec. | 00:00.024 | 00:00.322 | 00:14.396 | 00:09.076 | 00:00.022 | 00:01.103 | 00:00.002 | 00:00.087 | 00:00.001 | 00:03.386 | 00:03.521 | 00:02.133 | 00:10.309 | 00:37.055 | 11:42.910 |
| | PenDigits | 00:00.015 | 00:00.067 | 00:04.625 | 00:02.566 | 00:00.013 | 00:00.416 | 00:00.002 | 00:00.053 | 00:00.000 | 00:01.001 | 00:02.725 | 00:00.490 | 00:03.281 | 00:09.254 | 03:21.409 |
| Image Data | COIL20 | 00:01.026 | 00:03.658 | 01:04.304 | 00:14.817 | 00:00.001 | 00:00.379 | 00:00.001 | 00:00.004 | 00:00.000 | 00:01.637 | 00:13.524 | 00:00.310 | 00:08.941 | 00:02.378 | 00:11.710 |
| | COIL100 | 00:36.361 | 03:52.397 | 82:50.274 | 14:50.661 | 00:00.010 | 00:22.517 | 00:00.001 | 00:00.039 | 00:00.000 | 00:32.122 | 01:59.225 | 00:07.964 | 12:04.037 | 02:58.177 | 14:18.780 |
| | cmu_faces | 00:00.021 | 00:00.046 | 00:00.682 | 00:00.238 | 00:00.001 | 00:00.010 | 00:00.000 | 00:00.002 | 00:00.000 | 00:00.206 | 00:00.661 | 00:00.116 | 00:00.064 | 00:00.346 | 00:00.515 |
| | OptDigits | 00:00.019 | 00:00.335 | 00:03.442 | 00:01.420 | 00:00.006 | 00:00.162 | 00:00.001 | 00:00.024 | 00:00.000 | 00:00.502 | 00:01.154 | 00:00.290 | 00:00.974 | 00:03.073 | 00:44.270 |
| | USPS | 00:00.117 | 00:02.924 | 00:45.038 | 00:08.670 | 00:00.011 | 00:01.843 | 00:00.002 | 00:00.045 | 00:00.000 | 00:02.709 | 00:08.493 | 00:01.433 | 00:06.092 | 00:29.259 | 01:48.607 |
| | MNIST | 00:10.649 | 16:24.220 | 131:03.099 | 37:05.095 | 00:00.120 | 03:21.012 | 00:00.030 | 00:00.491 | 00:00.008 | 00:04.320 | 03:29.969 | 00:15.352 | 19:02.498 | 17:21.183 | - |

Table 6: ARI values for various combinations within our clustering framework SHiP over ten runs, compared with standard clustering methods. Noise points were handled as singleton clusters. This shows that the results yield high-quality clusterings when using the DC tree as an ultrametric.

| Dataset | DC tree | | | | | | HST-DPO | | | Cover tree | | | KD tree | | | competitors | | | | |
|---|---|---|---|---|---|---|---|---|---|---|---|---|---|---|---|---|---|---|---|---|
| | k-center GT | k-median GT | k-means GT | k-center Stability | k-median MoE | k-means Elbow | k-center Stability | k-median MoE | k-means Elbow | k-center Stability | k-median MoE | k-means Elbow | k-center Stability | k-median MoE | k-means Elbow | Eucl. k-means | SCAR | Ward | AMD-DBSCAN | DPC |
| **Density-based 2D-Data** | | | | | | | | | | | | | | | | | | | | |
| Boxes | 66.5 | **99.3** | **99.3** | 90.1 | **99.3** | 97.9 | 2.9±0.1 | 45.5±7.8 | 35.5±9.3 | 2.6 | 42.1±4.7 | 24.2±1.6 | 2.3 | 36.9 | 21.7 | 93.5±4.3 | 0.1±0.1 | 95.8 | 63.9 | 25.9 |
| D31 | 65.9 | **93.7** | **93.7** | 79.7 | 42.7 | 82.9 | 41.4±6.7 | 33.9±11.1 | 45.7±8.7 | 46.5±1.8 | 62.0±5.4 | 67.7±3.2 | 38.7 | 59.2 | 74.9 | 92.0±2.7 | 41.7±5.4 | 92.0 | 86.4 | 18.5 |
| 3-spiral | 55.2 | 98.1 | 98.1 | 98.6 | 98.1 | 98.1 | 6.6±1.3 | 5.8±4.2 | 6.2±3.0 | 9.6 | 14.7±1.8 | 16.3±1.9 | 9.6 | 5.9 | 9.6 | -0.6 | -0.3±0.1 | 0.0±0.1 | 30.6 | 97.4±7.8 |
| aggregation | 79.2 | **99.2** | **99.2** | 80.9 | 80.9 | 80.9 | 24.8±2.7 | 52.1±13.3 | 51.7±12.5 | 18.6±1.6 | 46.4±4.3 | 43.8±7.4 | 18.9 | 43.8 | 29.8 | 74.7±1.6 | 20.4±6.3 | 80.6±0.9 | 97.8 | 73.4 |
| chainlink | 100.0 | 100.0 | 100.0 | 100.0 | 100.0 | 100.0 | 8.0±5.1 | 12.9±5.6 | 11.1±4.7 | 6.1±0.4 | 12.1±2.9 | 8.2±0.5 | 5.2 | 7.2 | 7.0 | 9.5±0.4 | 2.4±1.5 | 28.0 | 12.4 | 8.1 |
| cluto-t4-8k | 0.4 | 79.8 | 87.0 | 87.2 | 75.5 | 61.5 | 3.4±0.3 | 32.2±6.1 | 27.6±8.4 | 2.8±0.1 | 22.9±2.1 | 19.3±3.6 | 3.1 | 23.7 | 14.1 | 45.4±2.0 | 0.6±0.5 | 49.8 | 80.1 | 43.6 |
| cluto-t5-8k | 0.1 | 74.8 | 77.9 | 86.8 | 66.7 | 65.0 | 3.9±0.3 | 50.3±6.7 | 45.2±10.1 | 4.3±0.1 | 42.0±7.5 | 29.0±6.6 | 3.2 | 37.5 | 22.3 | 74.2±0.3 | 0.3±0.2 | 73.9 | 87.4 | 42.2±2.7 |
| cluto-t7-10k | 1.0 | 68.8 | 68.8 | 88.9 | 71.4 | 36.9 | 2.8±0.1 | 25.1±4.3 | 19.4±4.4 | 2.8±0.2 | 18.9±2.4 | 13.5±0.5 | 2.3 | 17.8 | 10.9 | 33.5±1.5 | -0.1±0.9 | 41.2 | 87.8 | 31.0 |
| cluto-t8-8k | 63.4 | 79.9 | 80.1 | 63.4 | 72.4 | 65.1 | 3.8±0.3 | 26.0±4.5 | 20.5±4.0 | 3.9±0.2 | 21.3±3.7 | 15.5±0.8 | 3.0 | 23.0 | 13.4 | 35.3±1.7 | -0.1±0.2 | 33.5 | 76.9±0.1 | 30.2 |
| complex8 | 90.9 | 90.2 | 99.0 | 91.2 | 33.2 | 28.0 | 9.6±1.4 | 28.8±7.9 | 27.9±5.3 | 9.9±0.7 | 28.7±1.7 | 23.0±3.4 | 8.8 | 27.6 | 24.3 | 37.2±1.6 | 4.2±1.6 | 36.7 | 53.3 | 29.7 |
| complex9 | 93.1 | 83.3 | 83.3 | 93.1 | 93.1 | 61.4 | 8.0±0.9 | 26.7±6.8 | 24.3±4.5 | 7.5±0.7 | 25.2±2.5 | 17.8±3.4 | 7.2 | 20.5 | 16.6 | 37.6±2.2 | -1.6±1.9 | 42.2 | 81.3±0.3 | 21.2 |
| compound | 78.4 | 65.2 | 65.2 | 80.7 | 74.0 | 85.3 | 31.1±6.8 | 44.0±9.4 | 42.8±6.9 | 32.1±5.0 | 47.1±4.8 | 47.3±4.8 | 30.6 | 52.4 | 38.6 | 53.4±0.9 | 27.2±7.2 | 55.1±0.1 | 91.1 | 59.1 |
| dartboard1 | 100.0 | 100.0 | 100.0 | 100.0 | 100.0 | 100.0 | 5.3±5.3 | 7.6±8.0 | 8.7±9.1 | 12.6±0.4 | 16.2±1.9 | 17.6±1.0 | 8.9 | 14.3 | 13.9 | 2.8±9.2 | 14.8±4.0 | 2.8±1.5 | 0.0 | 0.8±0.1 |
| diamond9 | 68.6 | 99.7 | 99.7 | 83.6 | 70.4 | 45.1 | 22.6±8.2 | 41.9±12.4 | 44.4±13.1 | 14.6±0.6 | 58.8±7.7 | 45.9±4.1 | 11.0 | 44.3 | 30.0 | 98.1±5.6 | 5.2±1.4 | 59.6 | 96.6 | 15.1 |
| jain | 1.0 | 100.0 | 100.0 | 92.5 | 50.7 | 46.3 | 9.0±2.5 | 18.5±10.7 | 14.5±4.0 | 7.5±1.9 | 14.7±2.1 | 15.1±2.3 | 7.7 | 17.4 | 11.7 | 31.5±1.2 | -5.1±5.7 | 51.5 | 83.4 | 3.2±0.1 |
| pathbased | 51.6 | 54.8 | 54.8 | 58.7 | 54.8 | 58.3 | 22.9±4.5 | 29.6±7.8 | 35.0±7.0 | 21.1±5.9 | 29.1±6.8 | 26.3±10.0 | 17.7 | 24.9 | 25.4 | 46.1±0.1 | 27.6±9.3 | 48.3 | 45.7±0.8 | 38.8 |
| smile1 | 100.0 | 100.0 | 100.0 | 100.0 | 100.0 | 100.0 | 13.3±15.5 | 28.9±29.0 | 30.9±31.0 | 15.5±1.3 | 66.7±1.0 | 67.6±1.2 | 15.3 | 63.0 | 66.1 | 54.6 | 7.3±1.9 | 55.5±0.1 | 29.9 | 32.5±0.7 |
| **Tabular Data** | | | | | | | | | | | | | | | | | | | | |
| Synth_low | 94.2 | 84.0 | 84.0 | 95.4 | 35.9 | 37.2 | 34.8±16.4 | 55.1±16.6 | 54.7±14.6 | 13.2±2.3 | 55.2±8.7 | 67.9±0.8 | 12.8 | 49.2 | 80.0 | 55.9±13.2 | 1.4±1.0 | 62.5 | 32.4 | 0.0 |
| Synth_high | 94.1 | 84.6 | 84.6 | 95.4 | 23.5 | 70.5 | 27.3±20.2 | 36.4±15.0 | 53.8±10.3 | 13.0±0.4 | 24.8±1.6 | 67.6±3.4 | 8.0 | 35.0 | 46.3 | 47.9±8.2 | 2.9±0.7 | 71.7 | 0.0 | 0.0 |
| Mice | 1.1 | 26.7 | 20.0 | 0.2 | 11.6 | 11.9 | 10.7±1.6 | 10.0±2.0 | 10.8±1.4 | 9.7±2.7 | 13.5±1.9 | 11.8±1.1 | 8.6 | 11.0 | 9.9 | 14.5±1.1 | 12.7±3.1 | 16.4 | 0.0 | 3.0 |
| airway | 23.5 | 58.1 | 58.9 | 38.0 | 65.9 | 58.8 | 0.9±0.1 | 28.8±11.9 | 20.5±8.1 | 0.8 | 18.2±2.4 | 12.0±1.4 | 0.7 | 16.9 | 8.7 | 39.9±2.0 | -0.9±0.5 | 43.7 | 31.7 | 65.1 |
| lactate | 4.9 | 61.7 | 62.1 | 41.0 | 41.0 | 67.5 | 1.9±1.8 | 6.3±1.5 | 3.5±1.0 | 0.1 | 4.1±0.6 | 1.7±0.2 | 0.1 | 4.4 | 2.3 | 28.6±1.1 | 1.5±1.0 | 27.7 | 71.5 | 33.2 |
| HAR | 0.7 | 47.3 | 47.0 | 30.0 | 46.9 | 52.8 | 21.5±1.9 | 35.1±6.0 | 31.3±8.5 | 14.7±8.8 | 14.2±4.7 | 9.6±2.2 | 1.5 | 2.2 | 1.6 | 46.0±4.5 | 5.5±3.2 | 49.1 | 0.0 | -0.0 |
| letterrec. | -0.0 | 9.6 | 8.9 | 12.1 | 16.6 | 17.9 | 4.9±0.5 | 7.9±0.9 | 7.6±0.9 | 5.8±0.2 | 7.2±0.6 | 6.2±0.3 | 3.3 | 5.4 | 4.3 | 12.9±0.6 | 0.4±0.1 | 14.7±0.9 | 7.9 | -0.0 |
| PenDigits | 0.1 | 68.3 | 70.1 | 66.4 | 73.1 | 75.4 | 8.9±1.5 | 33.9±7.6 | 28.6±6.5 | 8.0±0.8 | 12.0±0.6 | 8.9±0.5 | 4.5 | 6.2 | 4.5 | 55.3±3.2 | 0.9±0.3 | 55.2 | 55.6 | 28.8±1.1 |
| **Image Data** | | | | | | | | | | | | | | | | | | | | |
| COIL20 | 58.4 | 72.8 | 75.5 | 81.2 | 72.8 | 72.6 | 39.9±2.4 | 38.4±9.0 | 44.2±4.8 | 46.4±4.4 | 46.6±2.1 | 47.7±2.0 | 22.5 | 16.9 | 18.5 | 58.2±2.8 | 33.5±2.0 | 68.6 | 39.2 | 35.9±0.1 |
| COIL100 | 19.6 | 70.2 | 69.9 | 80.1 | 66.8 | 70.0 | 40.6±4.7 | 34.3±12.6 | 43.2±6.2 | 46.6±1.5 | 50.1±1.2 |  | 23.6 | 22.5 | 24.4 | 56.1±1.4 | 16.7±0.8 | 61.4 | 14.2 | 0.2 |
| cmu_faces | 24.2 | 60.2 | 61.4 | 60.2 | 56.6 | 66.5 | 7.1±3.2 | 16.1±10.2 | 26.6±20.4 | 8.6±3.1 | 37.1±4.1 | 34.2±2.1 | 11.3 | 12.2 | 12.2 | 53.2±4.7 | 38.5±2.9 | 61.6 | 0.7 | 0.6 |
| OptDigits | 5.0 | 74.8 | 74.8 | 55.3 | 77.0 | 77.0 | 36.4±4.9 | 34.3±7.7 | 34.1±5.4 | 40.9±3.5 | 20.9±2.3 | 18.1±2.4 | 4.1 | 2.9 | 2.9 | 61.3±6.6 | 14.4±4.1 | 74.6±2.4 | 63.2 | 0.0 |
| USPS | 1.8 | 55.8 | 54.0 | 33.7 | 29.3 | 29.3 | 15.0±1.9 | 26.8±4.3 | 23.9±3.5 | 12.0±1.7 | 11.2±1.5 | 11.2±1.5 | 3.0 | 2.7 | 3.9 | 52.3±1.7 | 2.9±0.9 | 63.9 | 0.0 | 21.0 |
| MNIST | 0.3 | 50.1 | 51.0 | 19.7 | 41.7 | 46.0 | 12.3±2.8 | 7.3±2.3 | 7.6±2.0 | 11.1±1.7 | 5.4±0.6 | 5.4±0.6 | 1.4 | 0.2 | 0.3 | 36.9±1.0 | 1.3±0.4 | 52.7 | 0.0 | - |

Table 7: NMI values for various combinations within our clustering framework SHiP over ten runs, compared with standard clustering methods. Noise points were handled as singleton clusters. This shows that the results yield high-quality clusterings when using the DC tree as an ultrametric.

| Dataset | DC tree k-center GT | k-median GT | k-means GT | k-center Stability | k-median MoE | k-means Elbow | HST-DPO k-center Stability | k-median MoE | k-means Elbow | Cover tree k-center Stability | k-median MoE | k-means Elbow | KD tree k-center Stability | k-median MoE | k-means Elbow | Eucl. k-means | SCAR | competitors Ward | AMD-DBSCAN | DPC |
|---|---|---|---|---|---|---|---|---|---|---|---|---|---|---|---|---|---|---|---|---|
| **Density-based 2D-Data** | | | | | | | | | | | | | | | | | | | | |
| Boxes | 86.6 | 99.2 | 99.2 | 95.0 | 99.2 | 98.3 | 50.6 ± 0.3 | 74.0 ± 1.9 | 71.5 ± 2.7 | 51.1 ± 0.2 | 75.0 ± 1.5 | 69.1 ± 0.5 | 52.7 | 73.1 | 68.0 | 95.3 ± 1.9 | 8.8 ± 0.8 | 97.2 | 65.6 | 50.3 |
| D31 | 83.2 | 95.8 | 95.8 | 87.7 | 81.8 | 94.0 | 75.7 ± 2.4 | 69.7 ± 6.4 | 76.4 ± 4.1 | 78.6 ± 0.5 | 84.4 ± 2.1 | 86.1 ± 1.1 | 79.1 | 82.3 | 87.6 | 95.7 ± 0.9 | 81.5 ± 1.8 | 95.1 | 90.7 | 60.0 |
| 3-spiral | 71.0 | 96.7 | 96.7 | 97.6 | 96.7 | 96.7 | 24.0 ± 3.6 | 13.2 ± 8.6 | 17.0 ± 6.5 | 35.7 ± 4.1 | 39.1 ± 7.7 | 44.8 ± 2.9 | 26.7 | 17.0 | 26.7 | 0.1 | 0.2 ± 0.1 | 0.6 ± 0.1 | 45.2 | 95.9 ± 12.4 |
| aggregation | 86.6 | 98.8 | 98.8 | 88.9 | 88.9 | 88.9 | 59.6 ± 2.2 | 68.0 ± 7.7 | 71.3 ± 5.8 | 59.2 ± 1.3 | 76.0 ± 2.5 | 74.9 ± 3.2 | 62.1 | 73.8 | 67.8 | 85.0 ± 1.5 | 53.6 ± 4.5 | 91.9 ± 0.2 | 96.7 | 83.6 |
| chainlink | 100.0 | 100.0 | 100.0 | 100.0 | 100.0 | 100.0 | 22.6 ± 9.4 | 23.2 ± 10.5 | 25.2 ± 11.8 | 31.5 ± 0.8 | 38.9 ± 2.6 | 35.3 ± 0.5 | 31.3 | 27.5 | 33.5 | 7.1 ± 0.3 | 11.4 ± 2.9 | 36.6 | 34.7 | 26.6 ± 0.2 |
| cluto-t4-8k | 2.6 | 83.6 | 87.1 | 81.5 | 81.0 | 76.4 | 44.2 ± 0.3 | 51.3 ± 2.8 | 53.5 ± 2.5 | 44.3 ± 0.2 | 56.4 ± 1.0 | 56.0 ± 1.5 | 46.4 | 56.3 | 53.2 | 58.5 ± 0.4 | 10.0 ± 1.2 | 62.8 | 76.8 | 48.5 |
| cluto-t5-8k | 4.7 | 79.9 | 82.1 | 78.3 | 75.4 | 74.3 | 45.0 ± 0.4 | 65.2 ± 3.2 | 63.9 ± 3.8 | 46.8 ± 0.3 | 63.6 ± 2.9 | 59.1 ± 2.4 | 46.2 | 59.3 | 56.6 | 79.6 ± 0.2 | 10.9 ± 0.8 | 79.7 | 77.1 | 55.7 ± 3.4 |
| cluto-t7-10k | 5.0 | 80.5 | 80.4 | 83.7 | 82.4 | 69.7 | 45.8 ± 0.4 | 49.8 ± 3.1 | 52.0 ± 1.3 | 47.0 ± 0.4 | 53.7 ± 1.4 | 55.3 ± 0.7 | 47.5 | 53.7 | 53.8 | 57.2 ± 0.7 | 11.6 ± 0.9 | 64.1 | 82.6 | 43.6 |
| cluto-t8-8k | 77.3 | 85.7 | 85.6 | 79.3 | 86.0 | 80.2 | 45.9 ± 0.3 | 50.9 ± 4.5 | 52.2 ± 3.4 | 47.9 ± 0.5 | 54.0 ± 1.6 | 55.7 ± 0.6 | 47.6 | 57.7 | 54.4 | 56.5 ± 1.7 | 12.8 ± 1.2 | 55.3 | 70.0 ± 0.1 | 43.1 |
| complex8 | 93.8 | 94.4 | 98.8 | 91.3 | 70.9 | 68.3 | 50.8 ± 0.8 | 50.7 ± 8.3 | 55.7 ± 1.0 | 53.2 ± 1.1 | 58.4 ± 1.1 | 59.4 ± 1.5 | 55.1 | 59.4 | 61.6 | 59.4 ± 1.1 | 28.6 ± 2.6 | 58.1 | 65.3 ± 0.2 | 46.0 ± 0.1 |
| complex9 | 96.8 | 93.0 | 93.0 | 96.9 | 96.9 | 85.4 | 51.6 ± 0.9 | 53.1 ± 7.2 | 56.8 ± 3.3 | 52.9 ± 1.0 | 61.9 ± 2.7 | 62.2 ± 1.3 | 55.5 | 55.2 | 56.5 | 63.5 ± 1.8 | 23.6 ± 2.2 | 70.5 | 89.5 ± 0.3 | 41.1 |
| compound | 74.5 | 84.0 | 84.0 | 86.4 | 80.6 | 91.2 | 56.4 ± 3.0 | 58.3 ± 7.5 | 59.6 ± 5.0 | 62.6 ± 2.8 | 68.5 ± 2.1 | 68.8 ± 2.2 | 61.3 | 70.2 | 65.2 | 71.8 ± 0.2 | 51.6 ± 6.3 | 73.3 | 77.0 | 72.9 |
| dartboard1 | 100.0 | 100.0 | 100.0 | 100.0 | 100.0 | 100.0 | 38.5 ± 5.1 | 26.4 ± 8.3 | 30.3 ± 10.6 | 49.8 ± 0.6 | 42.5 ± 6.4 | 53.0 ± 1.8 | 42.2 | 35.2 | 37.9 | 3.5 ± 10.6 | 37.1 ± 5.8 | 5.7 ± 1.9 | 0.0 | 34.0 ± 0.1 |
| diamond9 | 86.8 | 99.6 | 99.6 | 83.3 | 85.7 | 76.7 | 58.3 ± 1.1 | 64.1 ± 8.4 | 67.9 ± 8.6 | 61.6 ± 0.5 | 80.1 ± 2.6 | 76.0 ± 1.2 | 59.5 | 70.2 | 68.4 | 99.0 ± 3.1 | 37.1 ± 3.6 | 99.6 | 96.3 | 41.7 |
| jain | 1.2 | 100.0 | 100.0 | 78.7 | 69.7 | 63.9 | 28.4 ± 2.5 | 32.5 ± 6.9 | 32.4 ± 4.4 | 30.2 ± 2.8 | 39.0 ± 3.0 | 39.0 ± 2.3 | 32.2 | 37.8 | 36.5 | 36.4 ± 0.7 | 7.4 ± 5.1 | 50.5 | 49.3 | 20.2 ± 0.2 |
| pathbased | 62.2 | 66.0 | 66.0 | 59.1 | 66.0 | 66.4 | 42.4 ± 3.7 | 39.5 ± 7.2 | 46.4 ± 6.0 | 44.2 ± 3.6 | 51.9 ± 4.2 | 50.6 ± 4.9 | 41.4 | 44.0 | 46.1 | 54.6 ± 0.1 | 37.3 ± 9.1 | 56.6 | 43.1 ± 0.7 | 43.6 |
| smile1 | 100.0 | 100.0 | 100.0 | 100.0 | 100.0 | 100.0 | 42.9 ± 9.7 | 42.5 ± 22.5 | 44.1 ± 24.2 | 50.4 ± 1.2 | 73.0 ± 1.1 | 73.7 ± 1.0 | 53.0 | 66.7 | 69.7 | 60.8 | 26.5 ± 2.8 | 61.2 ± 0.1 | 43.5 | 49.4 ± 4.2 |
| **Tabular Data** | | | | | | | | | | | | | | | | | | | | |
| Synth_low | 86.7 | 89.2 | 89.2 | 87.5 | 59.4 | 63.2 | 63.8 ± 8.1 | 72.5 ± 10.5 | 72.7 ± 10.0 | 57.9 ± 1.1 | 76.2 ± 3.7 | 79.8 ± 1.6 | 59.8 | 76.6 | 80.7 | 77.5 ± 7.0 | 24.0 ± 1.3 | 80.7 | 51.8 ± 0.5 | 0.0 |
| Synth_high | 86.6 | 87.1 | 87.1 | 87.5 | 43.2 | 81.1 | 61.6 ± 6.3 | 64.2 ± 14.2 | 70.6 ± 4.5 | 60.9 ± 0.2 | 66.3 ± 0.6 | 76.3 ± 0.7 | 58.5 | 70.0 | 69.3 | 71.4 ± 5.1 | 21.4 ± 1.1 | 84.1 | 0.0 | 0.0 |
| Mice | 21.2 | 43.0 | 39.5 | 8.0 | 25.6 | 29.5 | 29.6 ± 2.8 | 19.6 ± 3.4 | 23.4 ± 2.2 | 42.2 ± 2.5 | 36.2 ± 3.6 | 45.1 ± 2.9 | 35.5 | 28.3 | 32.5 | 25.7 ± 1.3 | 29.2 ± 2.2 | 27.9 | 0.0 | 44.9 |
| airway | 46.9 | 68.3 | 68.5 | 54.5 | 70.7 | 68.3 | 32.6 ± 0.2 | 56.4 ± 4.3 | 53.0 ± 3.8 | 33.0 ± 0.1 | 53.6 ± 1.2 | 48.9 ± 0.7 | 33.8 | 52.2 | 47.3 | 63.7 ± 1.0 | 6.7 ± 1.0 | 63.9 | 40.9 | 57.7 |
| lactate | 11.8 | 62.3 | 62.5 | 51.9 | 53.5 | 63.1 | 28.7 ± 4.5 | 38.2 ± 1.5 | 35.0 ± 1.4 | 24.2 ± 0.1 | 37.6 ± 0.8 | 33.4 ± 0.6 | 25.0 | 38.0 | 35.0 | 52.9 ± 1.6 | 3.4 ± 1.0 | 52.0 | 38.0 | 0.0 |
| HAR | 11.8 | 65.3 | 63.9 | 43.7 | 65.9 | 63.8 | 40.9 ± 1.0 | 51.9 ± 5.3 | 52.1 ± 4.2 | 39.6 ± 1.9 | 45.8 ± 1.4 | 44.3 ± 0.7 | 29.6 | 29.6 | 29.3 | 60.2 ± 2.2 | 24.6 ± 6.3 | 62.9 | 0.0 | 55.5 |
| letterrec. | 1.4 | 42.8 | 44.0 | 56.5 | 53.6 | 54.5 | 42.2 ± 2.2 | 33.0 ± 3.9 | 36.3 ± 4.1 | 51.9 ± 0.9 | 51.1 ± 1.3 | 53.2 ± 0.5 | 45.2 | 40.5 | 43.6 | 35.3 ± 0.5 | 19.4 ± 1.6 | 40.3 ± 0.9 | 49.3 | 49.7 |
| PenDigits | 7.5 | 77.4 | 78.6 | 70.3 | 81.7 | 82.8 | 50.6 ± 0.6 | 57.6 ± 2.7 | 59.3 ± 1.6 | 51.5 ± 0.5 | 57.8 ± 0.4 | 56.2 ± 0.3 | 49.1 | 50.2 | 49.6 | 68.0 ± 1.0 | 18.4 ± 1.7 | 72.7 | 59.9 | 46.0 ± 0.6 |
| **Image Data** | | | | | | | | | | | | | | | | | | | | |
| COIL20 | 80.7 | 86.9 | 88.0 | 89.1 | 86.9 | 86.8 | 68.8 ± 1.1 | 67.4 ± 8.4 | 70.9 ± 3.4 | 71.6 ± 1.5 | 75.9 ± 0.8 | 76.1 ± 0.8 | 50.0 | 51.7 | 53.1 | 77.8 ± 1.9 | 64.6 ± 0.7 | 82.6 | 65.9 | 64.5 ± 0.1 |
| COIL100 | 79.7 | 89.5 | 89.6 | 90.7 | 88.9 | 89.6 | 76.7 ± 0.9 | 73.5 ± 7.4 | 78.1 ± 3.3 | 78.3 ± 0.5 | 82.2 ± 0.3 | 82.9 ± 0.7 | 61.3 | 62.9 | 64.1 | 83.2 ± 0.6 | 73.1 ± 0.4 | 86.2 | 72.1 | 68.3 |
| cmu_faces | 71.2 | 84.0 | 85.4 | 78.9 | 81.5 | 87.3 | 31.5 ± 6.6 | 43.3 ± 15.2 | 52.2 ± 23.2 | 34.2 ± 5.1 | 78.6 ± 0.8 | 78.0 ± 0.5 | 43.4 | 44.2 | 45.0 | 75.7 ± 2.5 | 64.9 ± 1.6 | 81.5 | 22.6 | 63.5 |
| OptDigits | 29.5 | 83.0 | 83.0 | 62.3 | 84.5 | 84.5 | 61.2 ± 1.3 | 62.2 ± 2.6 | 61.9 ± 1.6 | 62.7 ± 1.5 | 58.8 ± 0.9 | 57.7 ± 0.9 | 37.6 | 38.6 | 39.3 | 72.2 ± 3.5 | 50.1 ± 4.3 | 83.4 ± 1.2 | 66.8 | 42.1 |
| USPS | 21.0 | 67.6 | 67.3 | 46.9 | 52.6 | 52.9 | 47.0 ± 0.8 | 47.2 ± 1.9 | 47.6 ± 1.9 | 47.3 ± 0.9 | 50.1 ± 0.5 | 50.7 ± 0.5 | 36.0 | 36.5 | 37.5 | 60.8 ± 1.0 | 31.2 ± 1.9 | 76.1 | 0.0 | 42.0 |
| MNIST | 14.8 | 61.6 | 62.9 | 38.2 | 58.3 | 62.6 | 39.1 ± 1.6 | 39.7 ± 0.5 | 40.0 ± 0.6 | 40.9 ± 0.8 | 41.9 ± 0.4 | 42.2 ± 0.4 | 25.6 | 25.0 | 25.0 | 49.1 ± 0.7 | 19.2 ± 2.7 | 68.2 | 0.0 | - |

Table 8: AMI values for various combinations within our clustering framework SHiP over ten runs, compared with standard clustering methods. Noise points were handled as singleton clusters. This shows that the results yield high-quality clusterings when using the DC tree as an ultrametric.

| | DC tree | | | | | | HST-DPO | | | Cover tree | | | KD tree | | | competitors | | | | |
|---|---|---|---|---|---|---|---|---|---|---|---|---|---|---|---|---|---|---|---|---|
| Dataset | k-center GT | k-median GT | k-means GT | k-center Stability | k-median MoE | k-means Elbow | k-center Stability | k-median MoE | k-means Elbow | k-center Stability | k-median MoE | k-means Elbow | k-center Stability | k-median MoE | k-means Elbow | Eucl. k-means | SCAR | Ward | AMD-DBSCAN | DPC |
| **Density-based 2D-Data** | | | | | | | | | | | | | | | | | | | | |
| Boxes | 86.4 | 99.2 | 99.2 | 94.9 | 99.2 | 98.3 | 42.2±0.6 | 73.9±1.9 | 71.3±2.8 | 44.2±0.4 | 74.9±1.5 | 68.9±0.5 | 49.7 | 73.0 | 67.8 | 95.3±1.9 | 8.6±0.8 | 97.2 | 57.0 | 36.0 |
| D31 | 79.2 | 95.6 | 95.6 | 85.2 | 81.3 | 93.7 | 67.2±4.4 | 69.1±6.3 | 75.5±4.1 | 72.2±0.9 | 83.7±2.1 | 85.4±1.2 | 74.8 | 81.7 | 86.9 | 95.5±1.0 | 80.6±1.9 | 94.8 | 89.3 | 34.5±0.1 |
| 3-spiral | 70.5 | 96.7 | 96.7 | 97.6 | 96.7 | 96.7 | 13.1±1.5 | 12.2±8.2 | 15.5±6.2 | 28.8±3.8 | 37.4±7.4 | 42.8±2.9 | 25.0 | 15.5 | 25.0 | −0.5 | −0.4±0.1 | 0.0±0.1 | 28.9 | 94.9±15.3 |
| aggregation | 86.2 | 98.8 | 98.8 | 88.8 | 88.8 | 88.8 | 52.7±3.6 | 67.5±7.7 | 70.6±5.9 | 52.7±2.1 | 75.3±2.6 | 74.1±3.4 | 59.9 | 73.2 | 66.6 | 84.8±1.5 | 53.0±4.5 | 91.8±0.2 | 96.6 | 83.5 |
| chainlink | 100.0 | 100.0 | 100.0 | 100.0 | 100.0 | 100.0 | 20.1±8.6 | 23.0±10.5 | 24.9±11.8 | 29.5±1.3 | 38.6±2.7 | 34.8±0.5 | 30.7 | 27.1 | 33.0 | 7.0±0.3 | 11.3±2.9 | 36.5 | 31.8 | 17.3±0.2 |
| cluto-t4-8k | 1.0 | 83.6 | 87.1 | 80.4 | 80.9 | 76.3 | 36.7±0.7 | 51.2±2.8 | 53.3±2.5 | 37.2±0.5 | 56.2±1.0 | 55.7±1.6 | 44.0 | 56.1 | 52.8 | 58.5±0.4 | 9.8±1.2 | 62.8 | 74.9 | 33.2 |
| cluto-t5-8k | 0.7 | 79.9 | 82.1 | 76.1 | 75.3 | 74.2 | 36.8±1.1 | 65.1±3.2 | 63.7±3.8 | 40.7±0.6 | 63.5±3.0 | 58.9±2.5 | 43.3 | 59.2 | 56.4 | 79.6±0.2 | 10.7±0.8 | 79.7 | 74.4 | 49.4±6.0 |
| cluto-t7-10k | 1.8 | 80.4 | 80.4 | 82.4 | 82.4 | 69.6 | 37.3±0.6 | 49.7±3.0 | 51.7±1.3 | 39.9±0.7 | 53.5±1.4 | 54.8±0.7 | 44.4 | 53.5 | 53.2 | 57.2±0.7 | 11.3±0.9 | 64.1 | 81.1 | 21.0 |
| cluto-t8-8k | 75.9 | 85.7 | 85.6 | 78.4 | 86.0 | 80.2 | 37.9±0.7 | 50.7±4.4 | 51.8±3.4 | 42.0±1.0 | 53.7±1.6 | 55.2±0.6 | 44.5 | 57.4 | 53.9 | 56.4±1.7 | 12.5±1.2 | 55.2 | 65.7±0.1 | 19.5 |
| complex8 | 93.8 | 94.4 | 98.8 | 91.0 | 70.4 | 67.5 | 43.1±1.7 | 50.3±8.3 | 55.0±1.0 | 47.4±1.6 | 57.9±1.1 | 58.7±1.5 | 52.9 | 58.9 | 61.0 | 59.1±1.1 | 28.1±2.6 | 57.9 | 61.9±0.2 | 22.5±0.1 |
| complex9 | 96.8 | 93.0 | 93.0 | 96.9 | 96.9 | 85.3 | 44.1±1.6 | 56.2±3.3 | 56.2±3.3 | 46.3±1.7 | 61.4±2.7 | 61.3±1.5 | 52.6 | 54.7 | 55.8 | 63.3±1.8 | 23.0±2.2 | 70.4 | 89.4±0.3 | 13.2 |
| compound | 71.7 | 83.7 | 83.7 | 86.2 | 80.4 | 91.1 | 46.3±5.5 | 57.7±7.5 | 58.7±4.9 | 57.3±4.4 | 67.6±2.2 | 67.7±2.4 | 59.6 | 69.5 | 64.0 | 71.3±0.2 | 50.6±6.4 | 72.8 | 71.8 | 72.7 |
| dartboard1 | 100.0 | 100.0 | 100.0 | 100.0 | 100.0 | 100.0 | 17.8±17.9 | 16.1±16.9 | 19.9±20.0 | 45.9±1.4 | 41.7±6.3 | 51.8±1.7 | 40.7 | 34.4 | 36.9 | 3.2±10.7 | 36.8±5.8 | 5.3±1.9 | 0.0 | 2.3±0.4 |
| diamond9 | 86.4 | 99.6 | 99.6 | 81.8 | 85.5 | 76.2 | 53.8±2.8 | 63.8±8.5 | 67.4±8.8 | 57.4±0.9 | 79.9±2.7 | 75.6±1.2 | 57.1 | 69.9 | 67.7 | 99.0±3.1 | 36.7±3.7 | 99.6 | 96.2 | 23.2±0.1 |
| jain | 0.7 | 100.0 | 100.0 | 78.1 | 69.6 | 63.7 | 24.1±3.1 | 32.0±6.9 | 31.8±4.5 | 26.5±3.4 | 38.4±3.0 | 38.3±2.4 | 31.0 | 37.3 | 35.9 | 36.2±0.7 | 7.1±5.2 | 50.4 | 42.4 | 5.9±0.3 |
| pathbased | 59.3 | 65.7 | 65.7 | 53.4 | 65.7 | 66.1 | 32.6±5.2 | 38.8±7.1 | 45.4±6.0 | 35.1±5.5 | 50.5±4.4 | 48.8±5.1 | 38.4 | 42.8 | 44.9 | 54.3±1.7 | 36.9±9.1 | 56.3 | 32.7±0.8 | 35.3 |
| smile1 | 100.0 | 100.0 | 100.0 | 100.0 | 100.0 | 100.0 | 23.0±23.1 | 32.4±32.4 | 33.9±34.1 | 44.3±2.0 | 72.6±1.1 | 73.2±1.0 | 50.9 | 66.4 | 69.3 | 60.6 | 26.1±2.8 | 61.1±0.1 | 42.9 | 49.3±4.2 |
| **Tabular Data** | | | | | | | | | | | | | | | | | | | | |
| Synth_low | 85.3 | 89.1 | 89.1 | 86.3 | 59.4 | 63.2 | 59.3±10.4 | 72.5±10.5 | 72.6±10.0 | 51.7±1.9 | 74.4±4.2 | 79.1±1.9 | 57.2 | 76.0 | 80.1 | 77.4±7.0 | 23.5±1.3 | 80.6 | 43.4±0.6 | 0.0 |
| Synth_high | 85.3 | 87.0 | 87.0 | 86.3 | 43.1 | 81.0 | 57.4±8.7 | 63.0±14.1 | 69.4±4.3 | 57.0±0.3 | 63.5±0.7 | 74.1±0.8 | 55.7 | 69.3 | 68.4 | 71.3±5.1 | 20.9±1.1 | 84.0 | 0.0 | 0.0 |
| Mice | 7.6 | 42.3 | 38.7 | 2.4 | 25.3 | 28.7 | 20.5±2.2 | 18.5±2.8 | 21.7±1.9 | 24.3±2.9 | 33.9±3.4 | 40.7±2.2 | 28.8 | 26.3 | 29.6 | 24.9±1.3 | 28.4±2.2 | 27.1 | 0.0 | 3.8±0.1 |
| airway | 46.2 | 68.2 | 68.4 | 52.2 | 70.6 | 68.2 | 25.6±0.4 | 56.3±4.3 | 52.7±3.9 | 27.2±0.3 | 53.4±1.2 | 48.6±0.7 | 31.0 | 52.0 | 46.9 | 63.6±1.0 | 6.4±1.0 | 63.9 | 28.1 | 52.8 |
| lactate | 11.6 | 62.2 | 62.5 | 51.4 | 53.5 | 63.1 | 25.5±6.8 | 38.1±1.6 | 34.8±1.5 | 19.5±0.2 | 37.5±0.9 | 33.2±0.7 | 23.1 | 37.9 | 34.9 | 52.9±1.6 | 3.4±1.0 | 52.0 | 26.8 | 0.0 |
| HAR | 1.4 | 65.2 | 63.9 | 34.4 | 65.9 | 63.7 | 33.1±1.8 | 51.9±5.3 | 52.0±4.3 | 28.3±1.5 | 45.3±1.5 | 43.4±0.9 | 25.9 | 28.4 | 27.4 | 60.2±2.2 | 24.6±6.3 | 62.9 | 0.0 | 55.5 |
| letterrec. | −0.0 | 42.5 | 43.6 | 34.2 | 52.9 | 53.6 | 29.6±0.8 | 31.8±3.4 | 34.4±3.5 | 39.4±0.4 | 47.3±0.6 | 47.6±0.2 | 37.8 | 37.4 | 38.5 | 34.9±0.5 | 18.8±1.7 | 40.0±0.9 | 30.1 | −0.0 |
| PenDigits | 2.6 | 77.4 | 78.6 | 65.7 | 81.7 | 82.8 | 40.8±1.4 | 57.3±2.7 | 58.8±1.7 | 42.5±1.1 | 56.8±0.4 | 54.5±0.4 | 45.8 | 48.5 | 47.1 | 67.9±1.0 | 18.2±1.8 | 72.6 | 47.8 | 21.7±1.4 |
| **Image Data** | | | | | | | | | | | | | | | | | | | | |
| COIL20 | 76.1 | 86.2 | 87.4 | 87.4 | 86.2 | 86.0 | 57.2±1.8 | 64.8±7.5 | 67.9±2.5 | 61.6±3.1 | 71.7±1.0 | 71.1±1.3 | 42.2 | 42.0 | 42.3 | 76.8±1.9 | 62.8±0.8 | 81.8 | 43.5 | 33.1±0.1 |
| COIL100 | 71.2 | 87.6 | 87.7 | 86.8 | 87.0 | 87.7 | 59.4±1.2 | 69.2±6.1 | 72.7±2.3 | 64.4±0.6 | 74.2±0.8 | 75.2±0.6 | 45.1 | 45.8 | 46.3 | 80.0±0.7 | 68.1±0.4 | 83.6 | 49.4 | 0.6 |
| emnl_faces | 61.9 | 82.0 | 83.5 | 67.5 | 79.8 | 85.5 | 26.4±6.7 | 40.6±14.7 | 48.4±21.8 | 30.8±5.3 | 68.1±2.3 | 66.1±1.3 | 27.3 | 30.0 | 30.1 | 72.6±2.9 | 60.5±1.9 | 79.1 | 6.5 | 2.1±0.1 |
| OptDigits | 20.3 | 83.0 | 83.0 | 50.5 | 84.4 | 84.4 | 57.3±1.5 | 61.3±2.9 | 60.9±1.9 | 59.4±1.7 | 57.0±1.0 | 55.3±1.0 | 32.2 | 33.2 | 33.3 | 72.2±3.5 | 49.9±4.4 | 83.3±1.2 | 58.5 | 0.0 |
| USPS | 2.2 | 67.5 | 67.2 | 20.2 | 52.5 | 52.9 | 37.7±0.7 | 46.9±1.8 | 47.0±1.8 | 36.6±2.0 | 47.6±0.6 | 47.7±0.6 | 31.6 | 32.0 | 32.2 | 60.7±1.0 | 31.0±1.9 | 76.0 | 0.0 | 7.5 |
| MNIST | 0.5 | 61.6 | 62.9 | 9.8 | 58.3 | 62.5 | 35.3±1.5 | 39.0±0.5 | 39.1±0.6 | 35.9±1.4 | 40.8±0.4 | 40.9±0.4 | 21.7 | 21.1 | 21.2 | 49.1±0.7 | 19.2±2.7 | 68.2 | 0.0 | - |

Table 9: Correlation coefficient values for various combinations within our clustering framework SHiP over ten runs, compared with standard clustering methods. Noise points were handled as singleton clusters. This shows that the results yield high-quality clusterings when using the DC tree as an ultrametric.

| | DC tree | | | | | | HST-DPO | | | Cover tree | | | KD tree | | | competitors | | | | |
| Dataset | k-center GT | k-median GT | k-means GT | k-center Stability | k-median MoE | k-means Elbow | k-center Stability | k-median MoE | k-means Elbow | k-center Stability | k-median MoE | k-means Elbow | k-center Stability | k-median MoE | k-means Elbow | Eucl. k-means | SCAR | Ward | AMD-DBSCAN | DPC |
|---|---|---|---|---|---|---|---|---|---|---|---|---|---|---|---|---|---|---|---|---|
| **Density-based 2D-Data** | | | | | | | | | | | | | | | | | | | | |
| Boxes | 70.2 | 99.3 | 99.3 | 90.3 | 99.3 | 97.9 | 12.0±0.2 | 51.0±5.3 | 44.9±6.7 | 11.4±0.1 | 50.0±3.8 | 36.5±1.2 | 10.7 | 45.5 | 34.3 | 93.5±4.3 | 0.5±0.3 | 95.8 | 68.5 | 28.1 |
| D31 | 65.9 | 93.7 | 93.7 | 80.2 | 51.9 | 83.5 | 47.9±5.9 | 41.0±8.3 | 48.9±6.7 | 53.4±1.4 | 63.2±4.6 | 67.8±3.1 | 46.7 | 60.9 | 75.1 | 92.0±2.6 | 45.9±4.6 | 92.0 | 86.8 | 19.3 |
| 3-spiral | 60.9 | 98.1 | 98.1 | 98.6 | 98.1 | 98.1 | 8.6±1.8 | 6.5±4.7 | 7.1±3.8 | 31.2±1.7 | 21.6±4.4 | 25.0±3.4 | 13.6 | 7.4 | 13.6 | -0.6 | -0.3±0.1 | 0.0±0.1 | 42.5 | 97.6±7.1 |
| aggregation | 80.5 | 99.2 | 99.2 | 82.4 | 82.4 | 82.4 | 36.1±2.3 | 53.9±12.7 | 55.6±11.2 | 54.1±3.8 | 54.1±3.8 | 52.2±5.9 | 30.7 | 51.8 | 40.1 | 75.8±1.6 | 20.6±6.2 | 81.8±0.8 | 97.8 | 76.1 |
| chainlink | 100.0 | 100.0 | 100.0 | 100.0 | 100.0 | 100.0 | 13.7±5.6 | 16.4±7.7 | 16.7±8.0 | 17.7±0.6 | 25.2±3.2 | 20.7±0.6 | 16.0 | 15.9 | 18.8 | 9.5±0.4 | 3.4±1.7 | 28.7 | 25.7±0.1 | 20.4±0.1 |
| cluto-t4-8k | 3.0 | 79.8 | 87.3 | 87.3 | 75.8 | 63.5 | 12.6±0.5 | 34.3±4.4 | 32.4±5.3 | 11.5±0.3 | 31.8±1.2 | 29.9±3.3 | 12.0 | 32.3 | 24.7 | 45.5±1.9 | 1.8±1.4 | 49.8 | 81.4 | 44.9 |
| cluto-t5-8k | 0.7 | 74.8 | 78.2 | 87.0 | 66.9 | 65.5 | 13.3±0.6 | 51.2±5.9 | 48.2±7.7 | 14.2±0.3 | 46.1±5.9 | 36.7±5.1 | 11.8 | 41.1 | 31.3 | 74.3±0.4 | 1.0±0.5 | 74.0 | 87.6 | 47.0±3.3 |
| cluto-t7-10k | 5.3 | 69.3 | 69.2 | 88.9 | 72.2 | 44.5 | 11.3±0.3 | 27.2±3.2 | 24.6±2.3 | 11.6±0.3 | 25.3±1.2 | 24.1±0.7 | 10.2 | 24.9 | 21.2 | 34.6±1.5 | -0.3±2.5 | 42.2 | 88.3 | 34.8 |
| cluto-t8-8k | 67.3 | 80.0 | 80.3 | 67.8 | 75.2 | 68.1 | 13.3±0.5 | 29.3±4.2 | 27.2±4.0 | 13.7±0.4 | 28.3±2.6 | 26.5±0.9 | 11.8 | 31.7 | 24.2 | 35.9±1.7 | -0.2±0.6 | 34.0 | 78.4±0.1 | 37.4 |
| complex8 | 91.2 | 90.2 | 99.0 | 91.6 | 44.2 | 40.4 | 21.0±1.5 | 31.2±7.1 | 32.2±3.1 | 21.9±1.1 | 33.6±1.7 | 31.5±2.2 | 20.7 | 34.0 | 34.3 | 37.7±1.6 | 6.4±2.5 | 37.1 | 53.6 | 37.4±0.1 |
| complex9 | 93.3 | 83.4 | 83.4 | 93.4 | 93.4 | 66.6 | 19.1±1.2 | 28.6±6.4 | 28.9±3.9 | 19.3±1.0 | 32.0±3.1 | 29.3±2.5 | 19.0 | 27.0 | 25.1 | 39.0±2.2 | -2.5±3.0 | 43.8 | 82.3±0.3 | 25.7 |
| compound | 79.2 | 65.2 | 65.2 | 82.3 | 76.7 | 86.2 | 37.3±5.1 | 45.5±9.2 | 44.8±6.9 | 40.2±2.9 | 48.8±4.4 | 49.4±4.1 | 36.2 | 54.3 | 43.6 | 54.4±1.1 | 27.5±7.3 | 56.0±0.1 | 91.4 | 64.7 |
| dartboard1 | 100.0 | 100.0 | 100.0 | 100.0 | 100.0 | 100.0 | 10.2±10.3 | 8.6±9.2 | 10.6±11.0 | 25.0±0.5 | 62.3±6.2 | 29.7±1.5 | 18.1 | 17.5 | 20.2 | 2.8±1.5 | 20.4±4.4 | 2.8±1.5 | 0.0 | 6.0±0.6 |
| diamond9 | 71.1 | 99.7 | 99.7 | 84.7 | 73.6 | 53.9 | 29.5±2.7 | 43.2±12.3 | 46.1±13.8 | 27.9±0.6 | 63.2±6.2 | 53.8±3.2 | 23.0 | 47.9 | 40.0 | 98.1±5.6 | 8.8±1.8 | 99.6 | 96.7 | 17.8 |
| jain | 5.4 | 100.0 | 100.0 | 92.8 | 58.3 | 54.9 | 19.8±4.0 | 24.6±11.3 | 23.5±5.8 | 19.0±3.2 | 26.9±3.3 | 28.0±2.4 | 19.5 | 28.6 | 24.5 | 32.3±1.2 | -5.3±5.8 | 52.4 | 84.5 | 12.7±0.4 |
| pathbased | 54.3 | 55.4 | 55.4 | 61.1 | 55.4 | 58.3 | 29.9±5.2 | 31.1±7.9 | 37.2±7.4 | 29.5±5.8 | 38.1±6.5 | 35.4±9.4 | 24.8 | 32.0 | 32.8 | 46.7±0.1 | 27.9±9.4 | 48.8 | 45.7±0.8 | 38.8 |
| smile1 | 100.0 | 100.0 | 100.0 | 100.0 | 100.0 | 100.0 | 19.1±20.2 | 29.0±29.1 | 31.5±31.7 | 28.9±1.2 | 69.9±1.5 | 71.5±0.9 | 28.4 | 64.7 | 69.8 | 54.7 | 9.6±2.1 | 55.6±0.1 | 36.5 | 40.2±2.3 |
| **Tabular Data** | | | | | | | | | | | | | | | | | | | | |
| Synth_low | 94.3 | 84.1 | 84.1 | 95.5 | 45.8 | 46.8 | 44.8±12.8 | 60.3±12.4 | 60.0±11.1 | 26.5±2.5 | 61.5±6.6 | 70.6±5.3 | 22.8 | 56.2 | 80.0 | 61.6±11.2 | 3.1±2.2 | 66.0 | 41.3±0.1 | 0.0 |
| Synth_high | 94.3 | 84.8 | 84.8 | 95.5 | 35.2 | 72.6 | 34.7±15.4 | 43.7±12.0 | 56.0±9.1 | 26.4±0.4 | 37.4±1.3 | 69.7±2.5 | 20.2 | 44.8 | 48.4 | 55.3±6.0 | 6.9±1.6 | 72.8 | 0.0 | 0.0 |
| Mice | 2.5 | 27.8 | 22.8 | 0.7 | 16.4 | 15.3 | 10.9±1.7 | 10.8±1.9 | 11.0±1.4 | 11.9±2.1 | 14.7±2.1 | 16.5±1.6 | 11.7 | 12.0 | 12.0 | 14.6±1.1 | 13.2±2.9 | 16.6 | 0.0 | 5.5±0.1 |
| airway | 35.5 | 61.2 | 61.8 | 45.5 | 66.9 | 61.8 | 6.5±0.2 | 38.0±9.4 | 31.9±7.0 | 6.1±0.1 | 29.5±2.0 | 24.2±1.5 | 5.6 | 28.6 | 20.3 | 46.5±1.8 | -2.0±1.1 | 49.2 | 43.2 | |
| lactate | 15.6 | 65.5 | 65.8 | 50.5 | 50.7 | 69.6 | 7.4±4.9 | 15.2±2.0 | 11.9±2.0 | 2.6 | 13.0±1.1 | 8.6±0.4 | 2.5 | 13.1 | 10.2 | 35.1±1.6 | 5.3±3.5 | 33.8 | 72.6 | 65.1 |
| HAR | 1.9 | 51.4 | 50.7 | 33.5 | 52.1 | 53.0 | 22.0±1.4 | 38.0±6.9 | 32.8±6.2 | 18.3±5.3 | 20.8±3.1 | 17.6±1.8 | 6.1 | 7.4 | 6.4 | 46.1±4.7 | 8.4±3.8 | 49.4 | 0.0 | 44.6 |
| letterrec. | -0.0 | 12.6 | 12.1 | 20.4 | 16.9 | 17.9 | 5.3±0.3 | 8.2±0.9 | 8.3±0.9 | 10.4±1.5 | 12.7±0.5 | 13.0±0.3 | 8.6 | 9.3 | 9.4 | 13.0±0.6 | 1.2±0.3 | 14.8±0.9 | 10.1 | -0.0 |
| PenDigits | 0.6 | 68.3 | 70.3 | 68.8 | 73.2 | 75.8 | 19.9±1.8 | 36.9±4.9 | 35.0±4.5 | 19.6±1.1 | 23.8±0.7 | 20.7±0.6 | 13.8 | 16.2 | 14.1 | 55.5±3.0 | 2.5±0.7 | 55.8 | 60.2 | 30.4±1.1 |
| **Image Data** | | | | | | | | | | | | | | | | | | | | |
| COIL20 | 59.5 | 73.3 | 75.9 | 81.2 | 73.3 | 72.8 | 40.2±2.5 | 40.9±6.1 | 45.1±4.6 | 47.0±4.7 | 49.7±1.8 | 51.8±1.9 | 24.6 | 20.8 | 23.2 | 58.5±2.8 | 35.2±1.7 | 68.6 | 39.3 | 40.0±0.1 |
| COIL100 | 28.1 | 70.7 | 70.5 | 80.6 | 67.8 | 70.5 | 41.7±5.4 | 37.0±9.2 | 44.1±5.1 | 45.1±3.8 | 48.4±1.3 | 51.1±1.3 | 25.0 | 25.5 | 28.0 | 56.5±1.4 | 22.4±0.7 | 61.7 | 19.6 | 2.8 |
| emnl_faces | 29.2 | 61.6 | 63.2 | 61.4 | 59.6 | 67.3 | 13.9±4.4 | 21.9±9.2 | 30.3±17.4 | 16.0±3.3 | 46.2±2.6 | 44.6±1.4 | 11.4 | 12.8 | 13.0 | 53.3±4.7 | 38.7±3.0 | 61.7 | 2.7 | 4.2±0.3 |
| OptDigits | 11.5 | 75.1 | 75.1 | 59.5 | 77.1 | 77.1 | 44.0±4.3 | 41.5±6.4 | 41.8±5.0 | 47.5±3.4 | 30.9±2.1 | 28.4±2.3 | 9.7 | 8.9 | 9.3 | 61.5±6.5 | 19.0±3.9 | 74.7±2.3 | 65.1 | 0.0 |
| USPS | 4.1 | 56.8 | 55.2 | 42.7 | 38.1 | 38.1 | 20.3±3.6 | 28.5±3.6 | 26.5±3.4 | 18.9±2.1 | 18.8±1.2 | 22.2±1.6 | 9.1 | 9.3 | 12.4 | 52.3±1.7 | 4.7±1.1 | 63.9 | 0.0 | 28.2 |
| MNIST | 1.0 | 50.6 | 52.5 | 33.0 | 44.6 | 49.8 | 14.9±2.7 | 13.7±1.6 | 14.2±1.4 | 14.8±1.5 | 13.1±1.1 | 13.3±1.1 | 4.2 | 2.1 | 2.3 | 36.9±1.0 | 3.2±0.8 | 52.8 | 0.0 | - |

