# OpenReview forum: "Ultrametric Cluster Hierarchies: I Want ‘em All!"
_NeurIPS.cc/2025/Conference — NeurIPS 2025 poster_

### Official Review · Reviewer_7zeU · 2025-06-04

**Clarity:** 3
**Significance:** 4
**Originality:** 3
**Rating:** 6
**Confidence:** 4

**Summary:**

This paper presents a significant theoretical and practical advance in the realm of hierarchical clustering in ultrametric spaces. The core contribution is a general theorem (Theorem 4.2) establishing that, given any relaxed ultrametric represented as an LCA-tree, one can compute optimal clusterings for all $k$ ($1 \leq k \leq n$) in $\text{Sort}(n)$ time. This result generalizes beyond the known $k$-center or $k$-median special cases in specific ultrametrics and applies to all standard center-based clustering objectives in any ultrametric setting. The authors complement this theoretical breakthrough with practical experiments demonstrating the competitive accuracy and extremely low runtime overhead of their approach, once the ultrametric ($\text{dc-dist}$) has been computed.

**Questions:**

Do the runtime and accuracy results hold similarly if the ultrametric is not dc-dist but one of these HST-based approximations?
This would give a broader sense of how sensitive the algorithm is to the choice of ultrametric.

**Ethical Concerns:**

["NO or VERY MINOR ethics concerns only"]

**Final Justification:**

I maintain my strong support of the paper.

**Limitations:**

yes

**Quality:**

4

**Strengths And Weaknesses:**

Strengths:

(*) Strong Theoretical Generalization:
The $\text{Sort}(n)$ runtime for all $k$ and all center-based objectives in any ultrametric is a powerful unifying result. The paper goes beyond previous specialized treatments (e.g., $k$-center in $\text{dc-dist}$ or $k$-median in 2-HST) by showing that the same principle holds in full generality.
I would have already accepted the paper with just that.

(*) Clear and Constructive Proofs:
The proofs, particularly for Theorem 4.2 and its generalizations in Appendix A.4, are clear, constructive, and mathematically rigorous. The reliance on the LCA-tree’s hierarchical properties and the strong triangle inequality in ultrametrics is well-argued and elegant.

(*) Practical Impact:
While the ultrametric fitting (e.g., $\text{dc-dist}$ computation) remains $O(n^2)$ in the dense case, the paper correctly separates this from the subsequent $\text{Sort}(n)$ clustering steps. This is a valuable insight for practitioners seeking to re-use a single ultrametric representation for multiple clustering tasks.

(*) Empirical Evaluation:
The experiments (Figure 3, Figure 4, and associated tables) convincingly demonstrate that their framework can produce clusterings with quality competitive to widely-used methods (DBSCAN, HDBSCAN) while being orders of magnitude faster once the ultrametric is precomputed.

(*) Clarity of Writing and Visuals:
The paper is clearly written overall. Figures 4 and 8 provide helpful visual intuition on how different partitioning methods ($k$-center, $k$-median, $k$-means) behave on the $\text{dc-dist}$ ultrametric.

(*) Well-crafted Figures and Thoughtful Acronym:
I really appreciate the craft that went into the figures and I really appreciate shipping a nice acronym.



Weaknesses:

(*) Ambiguity in Linking Visuals to Algorithm:
While Figure 4b showcases the DC-tree-based results that belong to the authors’ algorithm, this connection is not immediately obvious from the figure’s caption alone. Explicitly labeling these as “ours” or clarifying in the caption that they are the authors’ algorithmic contribution would improve clarity.

(*) Preprocessing Assumptions:
The authors could make it even clearer in the introduction and algorithm sections that their $\text{Sort}(n)$ guarantee assumes the LCA-tree is already computed, to avoid possible confusion for readers new to the area.

---

> ### Author Rebuttal · Authors · 2025-07-30
>
> We thank the reviewer for their valuable feedback and are happy that they like our work.
>
> ---
>
> ### Addressing weaknesses:
>
> Following your advice, we’ll emphasize that the runtime guarantees refer to the step after computing the ultrametric.
>
> However, we would like to note that in Figure 4, the clustering results for using different HSTs *and* the dc-dist results are all “ours”. I.e., our algorithm shows how to perform center-based clustering in ultrametrics optimally. Thus, Figure 4a shows “our” clustering results in various HST ultrametrics, while Figure 4b shows “our” clustering results on the dc-dist ultrametric.
>
> ---
>
> ### Addressing questions:
>
> The runtime results (building a new hierarchy and extracting the clustering) are independent of the chosen (precomputed) ultrametric.
>
> In the Appendix in Table 5, we further compare all ultrametric construction times: KD trees can be constructed the fastest, followed by Cover trees. Depending on the datasets, the HST-DPO tree construction is sometimes faster than building the DC tree, but for very high dimensions, building the DC tree is always faster.
>
> Furthermore, the accuracy depends on the dataset and type of clusters contained in it: as shown in the Appendix in Tables 6 (ARI), 7 (NMI), 8 (AMI), and 9 (CC), density-based clusters can best be found with the density-connectivity distance. In contrast, linearly separable clusters can be detected by using HST ultrametrics. In short: if the ultrametric fits the underlying assumptions on the data, then our techniques provide fast and consistent results.

---

### Official Review · Reviewer_X4Az · 2025-07-01

**Clarity:** 3
**Significance:** 4
**Originality:** 3
**Rating:** 5
**Confidence:** 4

**Summary:**

This paper shows that given an ultrametric (or equivalently a hierarchical clustering), we can compute all $(k,z)$-clusterings over it very quickly. In other words, this paper shows that once an ultrametric is fit to a given dataset, a host of center-based clusterings (like k-means, k-medians, k-center) can be obtained (essentially) for free for this ultrametric. The paper's main theoretical results are also well augmented with experiments showcasing the potential practical utility of their algorithms for exploratory data analysis of real world data.

**Questions:**

No significant questions

**Ethical Concerns:**

["NO or VERY MINOR ethics concerns only"]

**Final Justification:**

As I did not have any major questions/comments during the original review, I do not feel there is much need for further discussion from my end. The authors have agreed to improve some minor formatting in the paper which I felt would improve the paper's readability. I would like to maintain my score even after reading the other reviews.

**Limitations:**

Yes

**Quality:**

4

**Strengths And Weaknesses:**

Quality: The paper is technically sound with claims that are well supported by both theoretical analysis and also experimental results.

Clarity: The paper is overall clearly-written and was easy to read. One suggestion is that Figure 1 in section 1 is not at all helpful for readers at section 1. The paper makes references to Figure 1 in later sections (like section 3, immediately after Theorem 3.3) and it is very inconvenient to go back and compare the notes in section 3 with Figure 1 on the first page. My suggestion is to split up the 3 sub-figures in Figure 1 and place them in the individual related sections of the paper.

Significance: The paper's results are significant to the ML community. Clustering is a topic that is very much of interest to readers of NeurIPS with several papers related to clustering being published every year at NeurIPS and other related conferences. In particular, the paper's results are related to very popular clustering algorithms like DBSCAN and k-means clustering.

Originality: Surprisingly, this is the first paper (or that's what the authors claim) to compute center-based clusterings on ultrametrics in a unified manner. This would be very natural to do, and I imagine some enterprising practitioners would have tried off-the-shelf k-means implementations on computed ultrametrics like dc-dist. But it would have been of a more ad-hoc nature...

---

> ### Author Rebuttal · Authors · 2025-07-30
>
> We thank the reviewer for their thoughtful comments. We share the reviewer’s surprise that this had not been done before – both because it seems natural to try and because the algorithm turns out to be fairly straightfroward in retrospect (since the problem reduces to sorting).
>
> Regarding the placement of Figure 1: We agree and will split up Figure 1 into the three subfigures which will be shown in the respective sections. We had originally kept it all in one figure in the introductory section due to space constraints, but we agree in retrospect that the figure could be more intuitive.

---

> > ### Comment · Reviewer_X4Az · 2025-08-05
> >
> > As I did not have any major questions/comments during the original review, I do not feel there is much need for further discussion from my end. The authors have agreed to improve some minor formatting in the paper which I felt would improve the paper's readability. I would like to maintain my score even after reading the other reviews.

---

### Official Review · Reviewer_VtwA · 2025-07-04

**Clarity:** 2
**Significance:** 3
**Originality:** 2
**Rating:** 5
**Confidence:** 4

**Summary:**

The paper discusses clustering on ultrametrics. They should that the three main center-based clustering problems - k-means, k-median and k-center, can all be solved optimally and quickly in these spaces. Further, the solution space is itself formed by nested clusters. One reason ultrametrics are interesting is their hierarchical structure, making them an example of hierarchical clustering.

**Questions:**

I refer the authors to my above comments.

**Ethical Concerns:**

["NO or VERY MINOR ethics concerns only"]

**Final Justification:**

As detailed in the interaction with the authors.

**Limitations:**

Yes

**Quality:**

3

**Strengths And Weaknesses:**

The major strength of the paper is in its usefulness. Hierarchical clustering is very popular, and it's natural to want to cluster on top of such a construction. The fact that the authors show optimally fast run-times for multiple clustering problems using a well-known algorithm is another strength. I plan on using similar techniques in my own research.

There are however multiple issues that should be addressed to strengthen the paper, and these all revolve around the central motivation and its relationship to previous known results.

- It is known that k-median and k-center can be solves optimally on trees, and I suspect the same holds for k-means due to the same considerations. Surprisingly, this line of research is not mentioned at all in the current paper. The properties of the algorithm should be compared to those for trees, including run-time and hierarchical structure.

- Ultrametrics are a very limited model. The motivation for studying them is typically due to with the general technique of embedding spaces into ultrametics in order to solve some problem in the new space - for example the solution to the k-server problem build on Bartal trees. Such an embedding is generally computationally intense, and also induces very large distortion. This makes the fast and exact solution of the clustering problem on the HST less interesting, since there is already a run-time and distortion overhead induced in the initial step.

- The level of novelty in the current solution is not clear to me. The claim on page 3 that the authors prove "an elegant unknown property of ultrametrics: all standard center-based clustering tasks... can be solved optimally in any ultrametric" seems to be an exaggeration. That the problems can be solved on ultrametrics were known, as it evidenced by the literature on clustering on trees, and k-center and k-median have already been studied specifically for ultrametrics. Indeed, while the authors compare themselves to the second algorithm in Cohen-Addad et al., that algorithm was a modification to achieve parallel computation, and is almost as fast as the one presented in the current paper. It would be interesting to know if the algorithm presented there implies similar ones for k-center and k-means - I suspect that it does.

---

> ### Author Rebuttal · Authors · 2025-07-30
>
> We thank the reviewer for the detailed questions and for helping us improve the paper. Below, we address the three weaknesses listed by the reviewer:
>
> ### **Related work**
>
> Regarding the discussion of tree clustering literature: we will add this important context in revision, specifically references [1], [2], [3] and [4]. Please let us know if there are also others we should include. However, we believe there are a few important distinctions to be made between our setting and those in the related work:
> - In our setting, the centers can only be placed on leaf nodes
> - Our setting’s distances must be ultrametric, while distances over edge-weighted trees may not be ultrametric in nature.
>
> As a result, the setting of k-center or (k,z)-clustering over trees is a generalization of our conditions and, consequently, none of our reductions to sorting will necessarily work. We will provide a detailed analysis of the relationships to clustering on trees in the updated text.
>
> ### **Ultrametric limitations**
>
> We disagree that ultrametrics are a limited model. To be clear, we agree with the reviewer that HSTs (e.g., Bartal trees) induce distortion and that their upfront computation may be unappealing if the goal is just to perform clustering using our techniques. However, this is not the use case we are suggesting. Instead, we are arguing that if one has *already* computed an ultrametric (as is common in many settings), then they get many additional clusterings essentially for free. Throughout the literature, there are many settings in which one computes ultrametrics implicitly:
>   - Many standard hierarchical clustering algorithms are ultrametric in nature. For example, single-linkage, complete-linkage, ward-linkage, and min-max linkage all produce ultrametric cluster trees. Our results immediately apply to these settings.
>   - If a solution to the Dasgupta objective is an ultrametric cluster-tree, then our algorithms can be used to obtain new solutions to the objective. We refer to the response to reviewer vbSb for more details.
>   - The DBSCAN* and HDBSCAN algorithms implicitly operate over ultrametrics. These are extremely popular clustering algorithms in practice and our results expand their versatility considerably.
>   - Many HSTs (e.g., Cover trees, Bartal trees, KD trees) are already frequently used to accelerate machine learning and computer graphics tasks.
>
> Our algorithms can be applied "for free" in any of the above settings.
>
> ### **Novelty**
>
> We agree that the algorithm in Cohen-Addad et al. [5] is related to our proposed method. Indeed, it was the starting point from which we began considering the problem: our (k, z)-clustering algorithm is structurally very similar to their Algorithm 2. We will make this clearer in the revision. We will also rephrase the sentence so it’s clear there was prior work on optimal center-based clustering in ultrametrics. However, there are several points which we believe support our claim that we show an “elegant, previously unknown property of ultrametrics”:
> - Although it was known that one can perform center-based clustering in trees and ultrametrics, the treatment had been fractured across the literature. The runtimes on trees differ between k-median and k-center, and differ again when we transition to ultrametrics. Moreover, the algorithmic techniques vary as well. Instead, our results show that, when the inputs are ultrametric, the same algorithm (sorting) can be used to solve any center-based clustering task. We believe this to be an elegant result due to its simplicity and its unifying nature.
> - The algorithm by Cohen-Addad et al. does indeed work for k-means on an HST. However, it is not clear that it extends to k-center. Specifically, their Algorithm 1 does, but runs in $O(n^2)$ time (this is essentially the algorithm employed by Beer et al. [6]). It is then not immediately obvious how to adjust Cohen-Addad et al.’s Algorithm 2 to k-center, as their “Benf” function does not apply in the k-center case. Furthermore, their algorithm explicitly relies on the 2-HST structure (which is why they obtain a $\Delta$ term in their runtime).
>
> Thus, our theoretical results generalize (and simplify!) the algorithm from Cohen-Addad et al. – we show that *all* of these tasks can be performed in *any* ultrametric in the *same* runtime using the *same* underlying algorithm.
>
> We also note that our experimental analysis is significantly more expansive than any which we have found in the literature.
>
> ---
>
> ### References
>
> [1]: Shah, R. (2003). Faster algorithms for k-median problem on trees with smaller heights.
>
> [2]: Benkoczi, R., & Bhattacharya, B. (2005, October). A new template for solving p-median problems for trees in sub-quadratic time. In European Symposium on Algorithms (pp. 271-282). Berlin, Heidelberg: Springer Berlin Heidelberg.
>
> [3]: Bhattacharya, B., Das, S., & Dev, S. R. (2022). The weighted k-center problem in trees for fixed k. Theoretical Computer Science, 906, 64-75.
>
> [4]: Wang, H., & Zhang, J. (2021). An O(n\logn)-Time Algorithm for the k-Center Problem in Trees. SIAM Journal on Computing, 50(2), 602-635.
>
> [5]: Cohen-Addad, V., Lattanzi, S., Norouzi-Fard, A., Sohler, C., & Svensson, O. (2021). Parallel and efficient hierarchical k-median clustering. Advances in Neural Information Processing Systems, 34, 20333-20345.
>
> [6]: Beer, A., Draganov, A., Hohma, E., Jahn, P., Frey, C. M., & Assent, I. (2023, August). Connecting the Dots--Density-Connectivity Distance unifies DBSCAN, k-Center and Spectral Clustering. In Proceedings of the 29th ACM SIGKDD conference on knowledge discovery and data mining (pp. 80-92).

---

> > ### Author Response · Authors · 2025-08-04
> >
> > Dear Reviewer, as the discussion period is coming to a close soon, we wanted to follow up once more to check if we have addressed your concerns regarding our work's novelty and relationship to the existing literature. We would be keen to use the remaining time to discuss improvements to our paper.

---

> > > ### Comment · Reviewer_VtwA · 2025-08-06
> > >
> > > I thank the authors' for their thoughtful replies. I raise my score accordingly.

---

### Official Review · Reviewer_vbSb · 2025-07-06

**Clarity:** 2
**Significance:** 2
**Originality:** 2
**Rating:** 4
**Confidence:** 3

**Summary:**

The authors introduce a new clustering framework based on relaxed ultrametric spaces and their associated lowest-common-ancestor (LCA) trees. They establish connections to existing problems and objectives by providing theoretical algorithms tailored to these input spaces. The authors also illustrate the effectiveness of the proposed methods on some real-world datasets.

**Questions:**

* Could the authors elaborate the context of introducing relaxed ultrametric in Section 3? For example, what kind of issues or counterexamples occur if we decide to stick with using the usual ultrametric definition?
* I found the formatting of the algorithms somewhat difficult to follow. Also, it seems a bit inconsistent (say for example, Algorithm 7).
* Do you think one might obtain a provable result on using DC tree (or other ultrametric fitting algorithms) in this framework?

**Ethical Concerns:**

["NO or VERY MINOR ethics concerns only"]

**Final Justification:**

I maintain my positive score. The results look interesting and promising.

**Limitations:**

Yes; I am especially interested if similar works on Dasgupta objective can be done.

**Quality:**

3

**Strengths And Weaknesses:**

Strengths:
* The authors have made a substantial effort to establish a solid foundation for the problems they address.
* The authors made clear theoretical contributions and demonstrated how it could be beneficial in practice.

Weaknesses:
* It is hard to get ideas how the algorithms or proofs work.
* While the authors provide a framing for problems considered, this may not be compelling for everyone.
* The paper itself is not well self-contained. One should read Appendix carefully to understand the material.

---

> ### Author Rebuttal · Authors · 2025-07-30
>
> We thank the reviewer for their helpful comments. Regarding the weaknesses, we will consider ways to present the algorithms in the paper so that they are easier to interpret and will try to minimize the necessity of referring to the Appendix.
>
> ### Addressing Questions
>
> Q1) There are three primary reasons for introducing the results over relaxed ultrametrics rather than standard ones:
> - Interestingly, our reduction from (k, z)-clustering to k-center clustering in an ultrametric requires the definition of relaxed ultrametrics in order to go through. Specifically, (k, z)-clustering in a *standard* ultrametric reduces to k-center clustering in a *relaxed* one.
>
>   - The technical reason for this is that, to solve the (k, z)-clustering task in an ultrametric, we evaluate how much the cost would decrease if we placed a center at a specific node. These cost decreases turn out to be a *relaxed* ultrametric, even if the original setting is a regular ultrametric. To see this, consider the cluster tree of two leaf nodes with one parent node. Let the parent node have value 1. Consequently, the k-means solution for $k=1$ has cost 1. If we now place a second center on the open leaf node, our cost *decreases* by 1. As a result, the cost-decrease associated with a leaf node is *strictly positive*.
>
> - It turns out that none of the proofs—either in the k-center or the (k,z)-clustering settings—require that the distance from a point to itself be 0. As a result, we may as well introduce our results in their most general format.
> - This generality proves useful in practice as well. Specifically, the dc-dist is most naturally expressed as a relaxed ultrametric.
>
> Q2) Thank you for the feedback - we will standardize and simplify the presentation of our algorithms. We will also provide context regarding the algorithms in the Appendix in more detail.
>
> Q3) We are not entirely sure what the reviewer means here. Every proof / claim we make for relaxed ultrametrics holds for the dc-distance (which is a relaxed ultrametric) as well as any other (relaxed or standard) ultrametric. Thus, our results show optimal center-based clustering in density-connectivity settings as well.
>
> ### Addressing Limitation
>
> We agree that the connection to trees based on Dasgupta’s objective is an interesting research direction and warrants further investigation for multiple reasons:
>
> First, we note that the generalization of Dasgupta’s objective in [1] defines *admissible* cost functions as those whose ground truth solution corresponds to an ultrametric tree (Definition 3.1 in [1]).
>
> Second, solutions to Dasgupta’s objective are necessarily cluster trees. Thus, to turn this into an LCA-tree, one must find non-decreasing values to assign to the nodes. Often, there is a natural mechanism by which to do this (such as in single-linkage or complete-linkage clustering). As a result, for many solutions to Dasgupta’s objective, our results show that one can subsequently obtain *new* solutions to the objective by running k-median or k-means. This only takes $\texttt{Sort}(n)$ time. It remains to be seen whether these produce better guarantees, but we suspect that they might in some settings.
>
> ---
>
> ### References
>
> [1]: Cohen-Addad, Vincent, Varun Kanade, Frederik Mallmann-Trenn, and Claire Mathieu. "Hierarchical clustering: Objective functions and algorithms." Journal of the ACM (JACM) 66, no. 4 (2019): 1-42.

---

> > ### Author Response · Authors · 2025-08-04
> >
> > Dear Reviewer, as the discussion period is coming to a close soon, we wanted to follow up once more to check if we have addressed your concerns, specifically regarding the definition of the relaxed ultrametric and the application to the Dasgupta objective. We would be keen to use the remaining time to discuss improvements to our paper.

---

> > ### Comment · Reviewer_vbSb · 2025-08-06
> >
> > I appreciate the authors' responses. In particular, the detailed explanation regarding the use of relaxed ultrametrics and the motivation behind Dasgupta's objective was helpful in understanding the context. For Q3, I was referring to lines 343--345, as there may be several **theoretical** ultrametric fitting algorithms (may or may not be) relevant in this setting. I do not have any further major questions or concerns.

---

> ### Author Response · Authors · 2025-08-07
>
> Thank you for the reply and for elaborating on the question. In reference to lines 343-345, we agree that there are likely theoretical statements and guarantees that can be made for clustering with the dc-dist on, for example, Euclidean data. In essence, it seems that robust single-link and dc-dist relationships should reliably emphasize dissimilarities between points in Euclidean space.
>
> Specifically, assume there are two dense clusters in Euclidean space, $C^1$ and $C^2$, which are only separated by a small distance. Then some intra-cluster Euclidean distances might be larger than some inter-cluster ones. I.e., there might exist points $c_i^1, c_j^1 \in C^1$ and $c_i^2, c_j^2 \in C^2$ such that $d(c_i^1, c_j^1) = d(c_i^2, c_j^2) \gg d(c_j^1, c_i^2)$. However, under the dc-dist, this would be resolved -- points in the same cluster are treated as mutually close and we get that all intra-cluster distances are smaller than all inter-cluster distances.
>
> We believe that such guarantees for the dc-dist would be interesting avenues for future research.

---

> > ### Comment · Reviewer_vbSb · 2025-08-08
> >
> > Thank you for your clarification! I am satisfied with the authors' additional explanation and have no further points to raise.

---

### Decision · Program_Chairs · 2025-09-17

**Decision:**

Accept (poster)

**Comment:**

This paper presents a unifying theoretical framework for clustering in ultrametric spaces. It shows that all center-based clustering objectives (k-means, k-median, k-center for all k) can be solved optimally by constructing and partitioning hierarchy in the ultrametric space (SHiP). The resulting algorithm is remarkably efficient as these center-based clustering tasks can be reduced to sorting. Experiments on synthetic and real-world datasets, particularly with density-connectivity ultrametrics, demonstrate that the framework is competitive with typical clustering algorithms at reduced computational cost. The reviewers received the submissions positively, acknowledging the work’s soundness and potential impact; yet raised concerns about: novelty positioning (some overlap with prior work on clustering in ultrametrics/trees), need for non-standard relaxed ultrametrics, clarity (dense exposition, heavy reliance on appendices), and limited evaluation scope beyond dc-dist. We thank the reviewers and authors for engaging during rebuttal period towards making the paper better. The author response helped resolve most of reviewer concerns leading to some reviewers even raising scores. Overall, despite clarity and novelty caveats, the paper should accepted as it is technically solid, unifies important concepts, and provides a promising practical framework.